# On the Target-kernel Alignment: a Unified Analysis with Kernel Complexity

**Chao Wang**[†], **Xin He**[†♯,∗] **Yuwen Wang**[‡,∗] **Junhui Wang** [‡]

[†] School of Statistics and Management, Shanghai University of Finance and Economics
[♯] Key Laboratory of Mathematical Economics (SUFE), Ministry of Education, Shanghai
[‡] Chinese University of Hong Kong
wang.chao@stu.sufe.edu.cn, he.xin17@mail.shufe.edu.cn
wangyw@link.cuhk.edu.hk, junhuiwang@cuhk.edu.hk

## Abstract

This paper investigates the impact of alignment between the target function of interest and the kernel matrix on a variety of kernel-based methods based on a general loss belonging to a rich loss function family, which covers many commonly used methods in regression and classification problems. We consider the truncated kernel-based method (TKM) which is estimated within a reduced function space constructed by using the spectral truncation of the kernel matrix and compare its theoretical behavior to that of the standard kernel-based method (KM) under various settings. By using the kernel complexity function that quantifies the complexity of the induced function space, we derive the upper bounds for both TKM and KM, and further reveal their dependencies on the degree of target-kernel alignment. Specifically, for the alignment with polynomial decay, the established results indicate that under the just-aligned and weakly-aligned regimes, TKM and KM share the same learning rate. Yet, under the strongly-aligned regime, KM suffers the saturation effect, while TKM can be continuously improved as the alignment becomes stronger. This further implies that TKM has a strong ability to capture the strong alignment and provide a theoretically guaranteed solution to eliminate the phenomena of saturation effect. The minimax lower bound is also established for the squared loss to confirm the optimality of TKM. Extensive numerical experiments further support our theoretical findings. The Python code for reproducing the numerical experiments is available on Github.

## 1 Introduction

Kernel-based methods have attracted increasing attention in recent years (Belkin et al., 2018; Liang & Rakhlin, 2020; Ghorbani et al., 2020; Li et al., 2023), due to its close connection with some cutting-edge research topics, including the understanding of over-parameterized neural network through the neural tangent kernel (Jacot et al., 2018; Chizat et al., 2019) and large-scale kernel learning with gradient descent (Lin & Zhou, 2018; Xu et al., 2021). It is of fundamental importance to provide theoretical explanations of their behaviors under these research topics.

In literature, some recent works show that the learning rate of kernel-based methods is actually affected by both the model complexity of the considered reproducing kernel Hilbert space (RKHS) and the target-kernel alignment, a measure to quantify the similarity between the considered RKHS (or kernel matrix from the empirical point of view) and the target function, which is also known as the smoothness of a target function in the RKHS (Caponnetto & De Vito, 2007; Smale & Zhou,

---

[∗]Xin He and Yuwen Wang are the corresponding author.

38th Conference on Neural Information Processing Systems (NeurIPS 2024).

2007). Particularly, the existing learning rate for the kernel ridge regression (KRR) is proved to be $\mathcal{O}(n^{-\frac{2\alpha\gamma}{2\alpha\gamma+1}})$ for $\frac{1}{2} \leq \gamma \leq 1$, where $\gamma$ is known as the source condition parameter (Cui et al., 2021) and can be treated as a measure of target-kernel alignment at the population level, and $\alpha$ controls the model complexity of the RKHS. This rate aligns with the intuitive sense that strong alignment and small model complexity contribute to a faster learning rate. Yet, with the increasing in $\gamma$ such that $\gamma > 1$ which corresponds to a smoother target function, the best learning rate of KRR plateaus at $\mathcal{O}(n^{-\frac{2\alpha}{2\alpha+1}})$ (Caponnetto & De Vito, 2007). This phenomenon is known as the **saturation effect** (Bauer et al., 2007; Li et al., 2022) and is widely observed in many applications (Bauer et al., 2007; Gerfo et al., 2008). It has been conjectured for decades that no matter what carefully chosen tuning parameter, the learning rate of KRR plateaus when the smoothness of the target function exceeds certain levels. Most recently, Li et al. (2022) establishes the rigorous saturation lower bound of KRR that confirms practical conjecture. Amini (2021); Amini et al. (2022) propose a truncated KRR based on the spectral-truncated kernel matrix, and further prove that as the alignment becomes stronger, the truncated KRR can be consistently improved in terms of the expected mean squared error and eventually tends to the parametric rate. Clearly, this improvement effectively tackles the saturation effect for the KRR where the squared loss is specified. Yet, it is still unclear whether the saturation effect can be solved for the general kernel-based methods where the specified loss function belongs to a rich loss function family.

In this paper, motivated by this theoretical gap, we investigate the impact of target-kernel alignment from the kernel complexity perspective for various kernel-based methods by considering a general loss function which belongs to a rich loss function family. Our established results shed light on the statistical benefits of the truncated estimator and are also verified by extensive numerical experiments. We want to emphasize that in contrast to the existing works that focus on the KRR benefit from the analytical solution and thus their theoretical analysis heavily relies on the closed form of the solution to establish some critical results (Cui et al., 2021; Amini et al., 2022), the explicit solution does not exist anymore in this paper, which requires different technical treatments to conduct the theoretical analysis. Specifically, our theoretical analysis crucially relies on the kernel complexity which quantifies the complexity of the RKHS (Bousquet & Herrmann, 2002) and some empirical process techniques. The established results successfully capture the trade-off between the complexity of the truncated RKHS and approximation bias as presented in Theorem 4.2. A simpler bound when considering the polynomial case in Corollary 4.3 further indicates that with a carefully chosen truncated space, the truncated method can efficiently eliminate the saturation effect. More importantly, we establish the minimax lower bound when the squared loss is specified, and thus rigorously confirm the conjecture in Amini et al. (2022), stating that the truncated KRR attains minimax optimality.

## 1.1 Contributions

The main contribution of this paper is to offer a unified analysis and a comprehensive understanding of the impact of target-kernel alignment, and provide a theoretically guaranteed solution to eliminate the phenomena of saturation effect. Some of its contributions are listed as follows.

(i) By leveraging the kernel complexity function, we establish the upper bounds for both the standard kernel-based estimator and the truncated estimator for a general loss function belonging to a family of Lipschitz loss functions. The established bounds indicate that with the variation of the alignment level, the learning rates for these two estimators exhibit distinct trajectories. Specifically, under the polynomial decay assumption, when alignment is at a lower level, the standard kernel-based estimator and the truncated estimator share the same learning efficiency and improve with the rise in alignment level. Yet, when the alignment level surpasses a threshold ($\gamma = 1$ in Assumption 3.2), the learning rate of the kernel-based estimator plateaus with no improvement as $\gamma$ increases — a phenomenon known as the **saturation effect** in literature. As opposed, the learning rate of the truncated estimator exhibits continuous improvement with the increasing alignment level, thus eliminating the saturation effect. This indicates a significant improvement in the truncated estimator over the standard kernel-based estimator.

(ii) By employing the standard Fano method, we establish minimax lower bound when the squared loss is specified, indicating that for both the just-aligned and weakly-aligned regimes, both the standard kernel-based estimator and the truncated estimator achieve minimax optimality. Furthermore, for the strong-aligned regime, we demonstrate that the standard kernel-based estimator can only attain sub-optimality, while the truncated estimator is also minimax-optimal. Our minimax analysis significantly

extends the existing results presented in Yang et al. (2017), offering a unified perspective for realistic scenarios where the true target is assumed to reside in the RKHS.

(iii) Various numerical experiments are conducted in the context of various regression and classification problems to demonstrate the advantages of the truncated estimator, to support the established theory substantially. More interestingly, we also empirically verify the existence of a trade-off arising from the model complexity of the RKHS and target-kernel alignment.

## 1.2 Related Work

Kernel-based methods have been widely studied for the past few decades, and are known as the time-proven efficient tools for statistical analysis. The theoretical behaviors of the kernel-based estimator have been established in Caponnetto & De Vito (2007); Li et al. (2007); Smale & Zhou (2007); Patle & Chouhan (2013). The concept of target-kernel alignment has been introduced in Cristianini et al. (2001), where a classification algorithm is developed adapting to the target-kernel alignment with a significant improvement in classification accuracy. Follow-up works have expanded the concept of target-kernel alignment to some other learning tasks, including regression (Kandola et al., 2002) and multi-class classification (Guermeur et al., 2004). The implications of target-kernel alignment have also inspired some applications to spectral kernel learning (Hoi et al., 2006), and feature selection (Wong & Burkowski, 2011).

Most recently, many works have attempted to provide a theoretical understanding of the kernel-based method by considering the target-kernel alignment. Specifically, Canatar et al. (2021) investigates the generalization error of KRR and derives an analytical framework for the generalization error that captures the effect of the target-kernel alignment. Amini (2021) considers a spectrally truncated KRR and demonstrates that with a carefully chosen truncation, the truncated KRR outperforms the standard KRR. Li et al. (2022) verifies the saturation effect observed behind the KRR estimator by establishing a lower bound that $\mathcal{O}(n^{-\frac{2\alpha}{2+\alpha}})$ whatever the smoothness degree of the target function is. Motivated by these works, Amini et al. (2022) further demonstrates the non-monotonicity of the regularization curve for the bandlimited alignment setting and further reveals that the learning rate of the truncated KRR can be consistently enhanced as the degree of target-kernel alignment increases.

## 2 Preliminaries

**Background on RKHS.** Let $\mathcal{H}_K$ denote the reproducing kernel Hilbert space (RKHS) induced by a positive semi-definite kernel function $K(\cdot, \cdot) : \mathcal{X} \times \mathcal{X} \to \mathcal{R}^+$, where $\mathcal{X} \subset \mathcal{R}^p$. The inner product equipped with $\mathcal{H}_K$ is denoted as $\langle \cdot, \cdot \rangle_K$ with the endowed norm $\| \cdot \|_K^2 = \langle \cdot, \cdot \rangle_K$. For each $\mathbf{x} \in \mathcal{X}$, it is well-known that $K_\mathbf{x} := K(\mathbf{x}, \cdot) \in \mathcal{H}_K$ and the reproducing property holds that $\langle f, K_\mathbf{x} \rangle_K = f(\mathbf{x})$ for all $f \in \mathcal{H}_K$. We assume that $\sup_{\mathbf{x}, \mathbf{x}' \in \mathcal{X}} K(\mathbf{x}, \mathbf{x}') \leq \kappa^2$ for some positive constant $\kappa$. This condition is commonly assumed in literature and various popularly used kernel functions satisfy this condition, including the Gaussian kernel and Laplacian kernel.

**Problem setup.** We consider a collection of pairs $\{(\mathbf{x}_i, Y_i)\}_{i=1}^n$ where $\{\mathbf{x}_i\}_{i=1}^n$ is a collection of covariates and the response $Y_i$ is independently drawn from a conditional distribution $\mathbb{P}_{Y|\mathbf{x}_i}$ on $\mathcal{Y} \subset \mathcal{R}$. Throughout this paper, we focus on the fixed design setting, where $\{\mathbf{x}_i\}_{i=1}^n$ are fixed, otherwise we treat all the random quantities as conditioning on $\{\mathbf{x}_i\}_{i=1}^n$. A similar treatment also appears in Yang et al. (2017); Wei et al. (2017); Amini et al. (2022).

In the literature of machine learning, the learning task is often defined with some pre-specified loss function. Specifically, we consider a loss function $L(\cdot, \cdot) : \mathcal{R} \times \mathcal{R} \to \mathcal{R}^+$, where $L(y, f(\mathbf{x}))$ quantifies the inaccuracy for predictor $f(\mathbf{x})$ when $y$ is the true response. Then, the population risk function can be defined as

$$\mathcal{E}(f) := \mathbb{E}_{Y^n}\left[\frac{1}{n}\sum_{i=1}^n L\big(Y_i, f(\mathbf{x}_i)\big)\right],$$

where $\mathbb{E}_{Y^n}$ denotes the expectation taken over $Y_1, ..., Y_n$. In literature, the target function of interest in the learning task is typically defined as the minimizer of the population risk $f^* = \operatorname{argmin}_f \mathcal{E}(f)$. In this paper, we assume $f^* \in \mathcal{H}_K$ and consider a family of loss functions that $L$ is assumed to be convex and locally Lipschitz continuous in the second argument (Wainwright, 2019; Dasgupta et al., 2019). Here, locally Lipschitz continuity is specified as that for any $b > 0$, there exists some

positive constant $M_{L,b}$[2] such that for any $y \in \mathcal{Y}$ and $\mathbf{x} \in \mathcal{X}$, and for any $f, f' \in \mathcal{H}_K$ satisfying $\max\{\|f\|_K, \|f'\|_K\} \leq b$, the following inequality holds:

$$|L(y, f(\mathbf{x})) - L(y, f'(\mathbf{x}))| \leq M_{L,b}|f(\mathbf{x}) - f'(\mathbf{x})|.$$

It is worth pointing out that the local Lipschitz continuity is satisfied for a variety of commonly used loss functions, and some of them are listed in Table 1.

Table 1: Different losses with corresponding Lipschitz constant $M_{L,b}$

| **Loss** | Squared | Exponential | Check | Hinge | Huber | Logistic |
|---|---|---|---|---|---|---|
| $\mathbf{M}_{L,b}$ | $2U + 2b\kappa$[3] | 1 | 1 | 1 | $\tau$[4] | 1 |

Note that the choice of the loss function is task-specific based on the problem of interest and prior knowledge on the data. For instance, under the regression setting, the squared loss can be specified for mean regression and the check loss can be specified for quantile regression. Under the classification setting, the hinge loss can be specified for margin-based classification and the logistic loss can be specified for estimating the conditional probability.

## 3 Standard Kernel-based Method

In the rest of this paper, we use lowercase letters $\{y_i\}_{i=1}^n$ to denote the observations of $\{Y_i\}_{i=1}^n$, and denote the empirical measure of $\{\mathbf{x}_1, ..., \mathbf{x}_n\}$ by $\mathbb{P}_n$. Given an estimator $\widehat{f}$, its accuracy can be evaluated by the $\mathcal{L}(\mathbb{P}_n)$-error which is defined as $\|\widehat{f} - f^*\|_n^2 = \frac{1}{n}\sum_{i=1}^n (\widehat{f}(\mathbf{x}_i) - f^*(\mathbf{x}_i))^2$. We also use the excess risk that $\mathcal{E}(\widehat{f}) - \mathcal{E}(f^*)$ as an evaluation measure. To estimate the underlying target function $f^*$, we consider the following penalized empirical risk minimization problem that

$$\widehat{f}_\lambda = \underset{f \in \mathcal{H}_K}{\mathrm{argmin}} \left\{ \widehat{\mathcal{E}}(f) + \lambda \|f\|_K^2 \right\}, \tag{1}$$

where $\widehat{\mathcal{E}}(f) = \frac{1}{n}\sum_{i=1}^n L(y_i, f(\mathbf{x}_i))$ and $\lambda$ is regularization parameter. We define a sample operator $S_\mathbf{x} : \mathcal{H}_K \to \mathcal{R}^n$ as $S_\mathbf{x}(f) := \frac{1}{\sqrt{n}}(f(\mathbf{x}_1), ..., f(\mathbf{x}_n))^\top$, and its adjoint operator $S_\mathbf{x}^\top : \mathcal{R}^n \to \mathcal{H}_K$ is defined as

$$S_\mathbf{x}^\top(\boldsymbol{\alpha}) := \frac{1}{\sqrt{n}} \sum_{j=1}^n \alpha_j K(\cdot, \mathbf{x}_j), \quad \boldsymbol{\alpha} = (\alpha_1, ..., \alpha_n)^\top \in \mathcal{R}^n.$$

Then, by the representer theorem (Kimeldorf & Wahba, 1971), the minimizer of the learning task (1) must have a finite form that $\widehat{f}_\lambda = S_\mathbf{x}^\top(\widehat{\boldsymbol{\alpha}})$ where $\widehat{\boldsymbol{\alpha}} \in \mathcal{R}^n$ is the solution to the following optimization task

$$\widehat{\boldsymbol{\alpha}} = \underset{\boldsymbol{\alpha} \in \mathcal{R}^n}{\mathrm{argmin}} \left\{ \widehat{\mathcal{E}}(S_\mathbf{x}^\top(\boldsymbol{\alpha})) + \lambda \boldsymbol{\alpha}^\top \mathbf{K} \boldsymbol{\alpha} \right\}.$$

Let $\mathbf{K} = \left\{ \frac{1}{n} K(\mathbf{x}_i, \mathbf{x}_j) \right\}_{i,j=1}^n$ be the empirical kernel matrix where the scaling is for analytical simplicity. In the subsequent analysis, we further assume that $\mathbf{K}$ is positive which is also required in literature (Liang & Rakhlin, 2020; Amini et al., 2022). Then, the kernel matrix $\mathbf{K}$ admits an eigen-decomposition that $\mathbf{K} = \mathbf{U} \mathbf{D} \mathbf{U}^\top$, where $\mathbf{U} = (\mathbf{u}_1, ..., \mathbf{u}_n) \in \mathcal{R}^{n \times n}$ is an orthonormal matrix and $\mathbf{D} \in \mathcal{R}^{n \times n}$ is a diagonal matrix with positive elements $\mu_1, ..., \mu_n$ arranging in a descending ordering. Without of loss generality, we further require that $\mu_j \to 0$ as $j \to \infty$.

Let $\xi^* = \mathbf{U}^\top S_\mathbf{x}(f^*)$. The elements of the vector $\xi^*$ are referred to as target alignment (TA) scores (Amini et al., 2022), which quantify the agreement level between $f^*$ and $\mathbf{K}$. Intuitively, a more

---

[2] $M_{L,b}$ is a constant with possible dependence on $b$.

[3] For squared loss, we assume that $\mathcal{Y} \subset [-U, U]$, which is commonly adopted in literature of learning theory (Bartlett et al., 2005; Smale & Zhou, 2005, 2007; Wei et al., 2017).

[4] $\tau$ is the threshold parameter for Huber loss.

favorable learning rate can be achieved if the target and kernel are strongly aligned corresponding to fast decay TA scores. For example, the scenario that $S_{\mathbf{x}}(f^*)$ is predominantly situated in the space generated by the eigenvectors corresponding to the first several eigenvalues of $\mathbf{K}$ indicates a strong alignment. In other words, $\xi_j^*$ is expected to be as large as possible for small $j$ and as small as possible for large $j$. An ideal scenario is that $S_{\mathbf{x}}(f^*)$ exactly matches the eigenvector $\mathbf{u}_1$, leading to $\xi^* = (1, 0, ..., 0)^\top$ with proper scaling such that $\|f^*\|_n^2 = 1$. In this paper, we are devoted to providing an analytic framework for the impact of alignment on the performance of the kernel-based methods based on the kernel complexity of $\mathbf{K}$.

## 3.1 Technical Assumptions

The following necessary assumption is needed in our theoretical analysis.

**Assumption 3.1.** There exist two constants $0 < c_0 \le c_0'$ such that

$$c_0 \|f - f^*\|_n^2 \le \mathcal{E}(f) - \mathcal{E}(f^*) \le c_0' \|f - f^*\|_n^2,$$

for any $f \in \mathcal{H}_K$ satisfying $\|f - f^*\|_n^2 \le b$ with some sufficiently small constant $b > 0$.

The first inequality in Assumption 3.1 is a $c_0$-locally strong convexity condition, and the second inequality is a $c_0'$-local smooth condition of the loss function with respect to $\| \cdot \|_n$. Assumption 3.1 is commonly assumed in literature (Steinwart & Christmann, 2008; Wei et al., 2017; Li et al., 2019; Farrell et al., 2021). Due to space limit, some brief discussions are provided below. For the squared loss, Assumption 3.1 is satisfied with $c_0 = c_0' = 1$. For the check loss, the $c_0$-locally strong convexity condition is slightly more relaxing than the similar assumption in the literature (Lian, 2022) that requires the conditional density function of the noise term to be uniformly bounded away from zero. And, the $c_0'$-local smoothness condition holds if the conditional density function is uniformly bounded. Other widely used loss functions including Huber loss, Logistic loss, Hinge loss, and exponential loss also satisfy Assumption 3.1 under some mild conditions as discussed in Appendix F.

**Assumption 3.2.** There exist some constants $\gamma \ge \frac{1}{2}$ and $u \ge 2$ such that $\sum_{j=1}^n \xi_j^{*2} \mu_j^{-2\gamma} \le u^2$ for any $n$.

Assumption 3.2 imposes the regularization on the TA scores $\xi^*$ concerning $\mathbf{K}$. Note that the parameter $\gamma$ reflects the degree of target-kernel alignment, where a larger $\gamma$ indicates stronger alignment between $\mathbf{K}$ and $f^*$. Moreover, the parameter $\gamma$ in Assumption 3.2 can be considered analogous to the source condition parameter under the random design setting (Caponnetto & De Vito, 2007; Cui et al., 2021; Li et al., 2023). Further discussions on the extension to the random setting are deferred to Appendix B. In our subsequent analysis, we consider the following three cases that

- Just-aligned regime: $\gamma = \frac{1}{2}$ where we only assume $f^* \in \mathcal{H}_K$.
- Weakly-aligned regime: $\frac{1}{2} < \gamma \le 1$ where $f^* \in \mathcal{H}_K$ and is more aligned with $\mathbf{K}$.
- Over-aligned regime: $\gamma > 1$ where $f^* \in \mathcal{H}_K$ and has a strong alignment with $\mathbf{K}$.

## 3.2 The Upper Bound for Standard Kernel-based Method

In the rest of this paper, we use $c, C$ to denote some constants independent of $n, \gamma, \alpha$, which may hide the constants such as $u, c_0, c_0'$ and whose values may vary from line to line. In literature, the empirical kernel complexity function is defined as $R(\delta) := \sqrt{\frac{1}{n} \sum_{j=1}^n \min\{\delta^2, \mu_j\}}$ (Bartlett et al., 2005). $R(\delta)$ serves as a measure of complexity of $\mathcal{H}_K$ and is closely connected to local Rademacher complexity (Bartlett et al., 2005; Steinwart et al., 2009). It plays a crucial role in establishing our theoretical results via the critical radius $\delta_n$, defined as the smallest positive value $\delta$ such that

$$C \log \iota^{-1} R(\delta) \le \frac{c_0}{2} \delta^{2\eta+1}, \tag{2}$$

where $\eta = \min\{\gamma, 1\}$ and $\iota$ is specified in the subsequent theorems and corollaries. The learning rate of the kernel-based estimator defined in (1) highly depends on $\delta_n$, and the existence and uniqueness of $\delta_n$ are verified in Appendix B.1. As pointed out in Yang et al. (2017), the statistical dimension is defined as the first index for which the associated eigenvalue $\mu_j$ drops below $\delta^2$ that $d(\delta) := \min\{j \in [n] : \mu_j \le \delta^2\}$, where $[n] = \{1, 2, .., n\}$ and $d(\delta) := n$ if $\{j \in [n] : \mu_j \le \delta^2\} = \emptyset$.

Note that the statistical dimension serves as a measure of the intrinsic dimension of the kernel matrix $\mathbf{K}$. Moreover, a kernel is regular if the tail sum of its eigenvalue sequence can be well bounded as the form $\sum_{d(\delta)+1}^{n} \mu_j \lesssim d(\delta)\delta^2$ (Yang et al., 2017). Note that kernels in the kernel class with the polynomial or exponential decay in their eigenvalues are regular. Then, the kernel complexity can be well approximated by $\sqrt{d(\delta)\delta^2/n}$. The close connection between $d(\delta)$ and $R(\delta)$ enables us to find the explicit formulation of $\delta_n$ in our theoretical analysis.

The following theorem provides theoretical guarantees of $\widehat{f}_\lambda$ defined in (1) in terms of $\mathcal{L}(\mathbb{P}_n)$-error and the excess risk which hold with high probability.

**Theorem 3.3.** *Suppose that Assumptions 3.1 and 3.2 are satisfied and $\delta_n^2 \leq \lambda \leq 1$[5]. Let $\eta = \min\{\gamma, 1\}$. Then, for any $\iota \in (0,1)$, with probability at least $1 - \iota$, there holds*

$$\max\left\{\|\widehat{f}_\lambda - f^*\|_n^2, \ \mathcal{E}(\widehat{f}_\lambda) - \mathcal{E}(f^*)\right\} \leq C(\delta_n^{4\eta} + \lambda^{2\eta}).$$

The proof of Theorem 3.3 is provided in Appendix C. To complete the proof of Theorem 3.3, we only need to require the first inequality in Assumption 3.1. Note that the established bound for $\widehat{f}_\lambda$ consists of two terms that are related to the critical radius $\delta_n$ and the parameter $\lambda$. Compared to the existing works (Yang et al., 2017; Amini et al., 2022) under the fixed setting where only the squared loss is considered, Theorem 3.3 provides a comprehensive theoretical analysis on various kernel-based estimators by considering a general loss function with the help of kernel complexity and also considers the effect of the target-kernel alignment on the estimation performance under different aligned regimes. Moreover, we notice that with the choice of $\lambda$ satisfying $\lambda \asymp \delta_n^2$, an optimal rate can be achieved that

$$\mathcal{E}(\widehat{f}_\lambda) - \mathcal{E}(f^*) \asymp \|\widehat{f}_\lambda - f^*\|_n^2 \asymp \delta_n^{4\eta}.$$

Note that Amini et al. (2022) provides some valuable insights into the learning rate of the kernel-based estimator under the squared loss in terms of the expected $\mathcal{L}(\mathbb{P}_n)$-error where the following polynomial decay condition is required.

**Assumption 3.4.** There exist some constants $\alpha > 1$ and $\gamma \geq \frac{1}{2}$ such that the eigenvalues of $\mathbf{K}$ and the TA scores exhibit polynomial decay rate that

$$\mu_j \asymp j^{-\alpha} \quad \text{and} \quad \xi_j^{*2} \asymp j^{-2\gamma\alpha-1}.$$

In Assumption 3.4, the parameter $\alpha$ controls the complexity of $\mathcal{H}_K$ in the sense that a decreasing $\alpha$ results in the increasing compacity of the RKHS $\mathcal{H}_K$ (Cui et al., 2021; Amini et al., 2022). Various widely used kernels, including the Sobolev kernel and the Laplacian kernel, belong to this class. Note that with slight modification by setting $\xi_j^{*2} \asymp j^{-2\gamma\alpha-1}(\log j)^{-2}$, it can be verified that Assumption 3.4 directly leads to Assumption 3.2 if we ignore the logarithmic term.

By Assumption 3.4, it is clear that $d(\delta) \asymp \delta^{-2/\alpha}$, and consequently $\delta_n^2 \asymp \left(\frac{(\log \iota^{-1})^2}{n}\right)^{\frac{\alpha}{2\eta\alpha+1}}$. To better understand the established results in Theorem 3.3, the following corollary is also provided.

**Corollary 3.5.** *Suppose that Assumptions 3.1, 3.2 and 3.4 are satisfied. Let $\eta = \min\{\gamma, 1\}$. For any $\iota \in (0,1)$, if we choose $\lambda \asymp \left(\frac{(\log \iota^{-1})^2}{n}\right)^{\frac{\alpha}{2\eta\alpha+1}}$, then with probability at least $1 - \iota$, there holds*

$$\mathcal{E}(\widehat{f}_\lambda) - \mathcal{E}(f^*) \asymp \|\widehat{f}_\lambda - f^*\|_n^2 \leq C\left(\frac{(\log \iota^{-1})^2}{n}\right)^{\frac{2\eta\alpha}{2\eta\alpha+1}}.$$

Under the just-aligned regime that $\gamma = \frac{1}{2}$, the learning rate turns to be $\left((\log \iota^{-1})^2/n\right)^{\frac{\alpha}{\alpha+1}}$, which is consistent with that in literature (Wei et al., 2017) where merely assumes $f^* \in \mathcal{H}_K$. Yet, under the weakly-aligned regime that $\frac{1}{2} < \gamma \leq 1$, the learning rate exceeds $\left((\log \iota^{-1})^2/n\right)^{\frac{\alpha}{\alpha+1}}$ due to the stronger target-kernel alignment. More interestingly, under the over-aligned regime that $\gamma > 1$, the learning rate plateaus with no improvement as $\gamma$ increases, which indicates a saturation effect for the standard kernel-based method. It is also interesting to notice that no matter how the choice of $\lambda$, the learning rate is always lower bounded by $\mathcal{O}(n^{-\frac{2\alpha}{2\alpha+1}})$ for $\gamma \geq 1$ (Li et al., 2022). In the next section, we will show that a careful choice of truncation allows us to construct an estimator based on a reduced RKHS that achieves the best rate and mitigates the saturation effect for $\gamma > 1$.

---

[5]Note that we assume $\lambda \leq 1$ as the theoretical choice of $\lambda$ typically depends on $n$ and is close to zero for sufficiently large $n$.

# 4 Truncated Kernel-based Method

To construct the reduced RKHS, we introduce a collection of functions $\{\psi_k\}_{k \in [n]} \subset \mathcal{H}_K$, defined as $\psi_k := \operatorname{argmin} \{ \|\psi\|_K : \psi \in \mathcal{H}_K, S_{\mathbf{x}}(\psi) = \mathbf{u}_k \}$. It can be verified that $\{\psi_k\}_{k \in [n]}$ is unique and by the orthogonality of $\mathbf{u}_1, ..., \mathbf{u}_n$, we also have $\langle \psi_i, \psi_j \rangle_n = 1$ for $i = j$ and $0$ otherwise (Amini et al., 2022). Then, for a given $r$, we define a function space as

$$\mathcal{H}_{K_r} := \Big\{ \sum_{k=1}^{r} \alpha_k \psi_k : \boldsymbol{\alpha} = (\alpha_1, ..., \alpha_r)^\top \in \mathcal{R}^r \Big\}.$$

Let $\mathcal{H}_{K_r}$ be equipped with the norm $\|f\|_{K_r}^2 = \langle f, f \rangle_{K_r}$, where the inner product is defined as $\langle f, g \rangle_{K_r} := \sum_{k=1}^{r} \alpha_k \beta_k / \mu_k$ for $f = \sum_{i=1}^{r} \alpha_k \psi_k$ and $g = \sum_{i=1}^{r} \beta_k \psi_k$. The following lemma from Amini et al. (2022) indicates that $\mathcal{H}_{K_r}$ is also an RKHS associated with a different kernel function.

**Lemma 4.1.** $\mathcal{H}_{K_r} \subset \mathcal{H}_K$ is an $r$-dimensional RKHS with reproducing kernel $K_r(\mathbf{x}, \mathbf{x}') = \sum_{k=1}^{r} \mu_k \psi_k(\mathbf{x}) \psi_k(\mathbf{x}')$.

Clearly, $\mathcal{H}_{K_r}$ can be treated as a relatively smaller function space compared to the full RKHS $\mathcal{H}_K$. Based on $\mathcal{H}_{K_r}$, we can find a truncated kernel-based estimator by solving

$$\widehat{f}_{\lambda,r} = \operatorname*{argmin}_{f \in \mathcal{H}_{K_r}} \big\{ \widehat{\mathcal{E}}(f) + \lambda \|f\|_{K_r}^2 \big\}.$$

For the truncated RKHS $\mathcal{H}_{K_r}$, we also define the operator $S_{\mathbf{x},r}^\top : \mathcal{R}^n \to \mathcal{H}_{K_r}$ as

$$S_{\mathbf{x},r}^\top(\boldsymbol{\alpha}) := \frac{1}{\sqrt{n}} \sum_{j=1}^{n} \alpha_j K_r(\cdot, \mathbf{x}_j), \quad \boldsymbol{\alpha} = (\alpha_1, ..., \alpha_n)^\top \in \mathcal{R}^n.$$

Then, by the representer theorem (Kimeldorf & Wahba, 1971) again, $\widehat{f}_{\lambda,r}$ also has a finite solution that $\widehat{f}_{\lambda,r} = S_{\mathbf{x},r}^\top(\widehat{\boldsymbol{\alpha}}_r)$ and $\widehat{\boldsymbol{\alpha}}_r$ can be obtained by solving the following optimization task

$$\widehat{\boldsymbol{\alpha}}_r = \operatorname*{argmin}_{\boldsymbol{\alpha} \in \mathcal{R}^n} \big\{ \widehat{\mathcal{E}}(S_{\mathbf{x},r}^\top(\boldsymbol{\alpha})) + \lambda \boldsymbol{\alpha}^\top \mathbf{K}_r \boldsymbol{\alpha} \big\}.$$

where $\mathbf{K}_r = \big\{ \frac{1}{n} K_r(\mathbf{x}_i, \mathbf{x}_j) \big\}_{i,j=1}^{n}$ is the empirical kernel matrix w.r.t. $K_r$. Note that the truncated method does not impose an additional computational cost compared to the standard kernel method, and its detailed discussion will be provided in Appendix B.5. By the construction of $\{\psi_i\}_{i \in [n]}$, it is easy to verify that $\mathbf{K}_r = \mathbf{U} \mathbf{D}_r \mathbf{U}^\top$, where $\mathbf{D}_r$ is diagonal matrix with elements $\mu_1, ..., \mu_r, 0, ..., 0$, detailed proof can be seen in Appendix B.2. This further implies that $\mathbf{K}_r = \mathbf{K}$ when $r = n$, and thus leads to $\widehat{f}_\lambda(\mathbf{x}_i) = \widehat{f}_{\lambda,n}(\mathbf{x}_i)$ for each $i \in [n]$.

## 4.1 The Upper Bound for Truncated Kernel-based Method

Given the truncated RKHS $\mathcal{H}_{K_r}$, our theoretical results below rely on the truncated kernel complexity function, defined as $R_r(\delta) := \sqrt{\frac{1}{n} \sum_{j=1}^{r} \min\{\delta^2, \mu_j\}}$. Moreover, the critical radius $\delta_{n,r}$ can be defined as the smallest positive value $\delta$ such that

$$C \log \iota^{-1} R_r(\delta) \leq \frac{c_0}{2} \delta^{2\eta+1}. \tag{3}$$

The existence and uniqueness of $\delta_{n,r}$ are verified in Appendix B.1. It can be verified that $R_r(\delta) \leq R(\delta)$ and thus leads to $\delta_{n,r} \leq \delta_n$. This observation indicates a potential improvement of the truncated estimator $\widehat{f}_{\lambda,r}$ and is the core of our theoretical analysis. The following theorem shows that $\widehat{f}_{\lambda,r}$ converges to the underlying target in terms of the $\mathcal{L}(\mathbb{P}_n)$-error and the excess risk with high probability.

**Theorem 4.2.** *Suppose that Assumptions 3.1 and 3.2 are satisfied and* $\max\{\delta_{n,r}^2, \sum_{j=r+1}^{n} \xi_j^{*2}\} \leq \lambda \leq 1$. *Let* $\eta = \min\{\gamma, 1\}$. *Then, for any* $\iota \in (0, 1)$, *with probability at least* $1 - \iota$, *there holds*

$$\max \big\{ \|\widehat{f}_{\lambda,r} - f^*\|_n^2, \ \mathcal{E}(\widehat{f}_{\lambda,r}) - \mathcal{E}(f^*) \big\} \leq C \big( \delta_{n,r}^{4\eta} + \lambda^{2\eta} + \sum_{j=r+1}^{n} \xi_j^{*2} \big).$$

The proof of Theorem 4.2 is provided in Appendix D. The established upper bound of $\widehat{f}_{\lambda,r}$ first decomposes the total error into two components: estimation error (the first two terms), which is controlled by complexity of the reduced RKHS $\mathcal{H}_{K_r}$, and approximation bias (the last term), which results from the dissimilarity between the truncated RKHS $\mathcal{H}_{K_r}$ and the full RKHS $\mathcal{H}_K$ where $f^*$ belongs to. Specifically, a smaller value of $r$ reduces the complexity of $\mathcal{H}_{K_r}$, possibly leading to a more favorable estimation error. Yet, it amplifies the gap between $\mathcal{H}_{K_r}$ and $\mathcal{H}_K$ and may lead to an extra approximation bias which may be significantly large. Clearly, $r$ can be regarded as a trade-off parameter that balances the approximation bias $\sum_{j=r+1}^{n} \xi_j^{*2}$ and the estimation error $\delta_{n,r}^{4\eta} + \lambda^{2\eta}$. It is clear that if $r = n$, the approximation bias is zero and the upper bound of $\widehat{f}_{\lambda,n}$ recovers that of $\widehat{f}_\lambda$. This implies that with careful choice of $r$, the truncated method at least performs as well as the standard estimator. If Assumption 3.4 also holds, we can conclude that $\sum_{j=r+1}^{n} \xi_j^{*2} \lesssim r^{-2\gamma\alpha} \mathrm{I}_{\{r<n\}}$. Then, the best choice of $r$ and $\lambda$ to achieve an optimal rate is given by $r \asymp \delta_{n,r}^{-2/(\gamma\alpha)}$ if $\gamma > 1$; $r = n$ if $\frac{1}{2} \leq \gamma \leq 1$, and $\lambda \asymp \delta_{n,r}^2$. Accordingly, there holds

$$\mathcal{E}(\widehat{f}_{\lambda,r}) - \mathcal{E}(f^*) \asymp \|\widehat{f}_{\lambda,r} - f^*\|_n^2 \asymp \delta_{n,r}^{4\eta}.$$

For the comparison of Corollary 3.5, we also establish the following corollary for $\widehat{f}_{\lambda,r}$.

**Corollary 4.3.** *Suppose that Assumptions 3.1, 3.2 and 3.4 are satisfied. For any $\iota \in (0,1)$, if we choose $\lambda \asymp \left(\frac{(\log \iota^{-1})^2}{n}\right)^{\frac{\max\{\gamma,1\}\alpha}{2\gamma\alpha+1}}$ and $r \asymp \left(\frac{n}{(\log \iota^{-1})^2}\right)^{\frac{1}{2\gamma\alpha+1}} I_{\{\gamma>1\}} + n I_{\{\frac{1}{2}\leq\gamma\leq1\}}$, then with probability at least $1 - \iota$, there holds*

$$\mathcal{E}(\widehat{f}_{\lambda,r}) - \mathcal{E}(f^*) \asymp \|\widehat{f}_{\lambda,r} - f^*\|_n^2 \leq C\left(\frac{(\log \iota^{-1})^2}{n}\right)^{\frac{2\gamma\alpha}{2\gamma\alpha+1}}.$$

Clearly, under the over-aligned regime that $\gamma > 1$, the truncated estimator $\widehat{f}_{\lambda,r}$ can achieve a faster learning rate compared to $\widehat{f}_\lambda$; for $\frac{1}{2} \leq \gamma \leq 1$, the trivial choice of $r = n$ is optimal and $\widehat{f}_{\lambda,r}$ shares the same learning rate as $\widehat{f}_\lambda$. More impressively, the learning rate of $\widehat{f}_{\lambda,r}$ can be continuously increased with the enhancement of the target-kernel alignment, thus eliminating the saturation effect. Note that as $\gamma \to \infty$, the learning rate of $\widehat{f}_{\lambda,r}$ approaches $\frac{1}{n}$, meaning that the truncated estimator can successfully capture the strong alignment to attain a comparable rate to the parametric rate.

**The connection between $r$ and $d(\delta)$.** Recall that for the regular kernel class, we have $R(\delta) \asymp \sqrt{n^{-1}d(\delta)\delta^2}$. It can also be shown by simple algebra that $R_r(\delta) \asymp \sqrt{n^{-1}\min\{r,d(\delta)\}\delta^2}$ (See Appendix D.3 for details). Particularly, for the kernel class with polynomial decay, we have $d(\delta) \asymp \delta^{-2/\alpha}$. Once the critical radius $\delta_{n,r}$ is determined for specified kernel matrix, we denote $d_n = d(\delta_{n,r}) \asymp \delta_{n,r}^{-2/\alpha}$ and take $r \asymp \delta_{n,r}^{-2\eta/(\gamma\alpha)}$ to balance $\delta_{n,r}^{4\eta}$ and $r^{-2\gamma\alpha}$. Consequently, we obtain $r \asymp d_n^{\eta/\gamma}$. Such a relation between $r$ and $d_n$ is very reflective and provides theoretical insight into why the truncated estimator is more efficient under a more aligned situation. Specifically, for the case $\gamma > 1$, it is clear that $r \asymp d_n^{1/\gamma} < d_n$, and we have

$$R_r(\delta_{n,r}) \asymp \sqrt{n^{-1}r\delta_{n,r}^2} < \sqrt{n^{-1}d_n\delta_{n,r}^2} \asymp R(\delta_{n,r}).$$

As a result, the truncated kernel complexity is substantially reduced compared to $R(\delta_{n,r})$, leading to an improved learning rate. On the contrary, for the case that $\frac{1}{2} \leq \gamma \leq 1$, we have $r \asymp d_n$ and $R_r(\delta_{n,r}) \asymp \sqrt{n^{-1}d_n\delta_{n,r}^2} \asymp R(\delta_{n,r})$, which indicates the truncated kernel complexity remains invariant as $r$ decreases. To avoid introducing additional approximation bias, the best choice of truncation level turns out to be $r = n$.

## 4.2 Minimax Lower Bound

In this section, we establish the minimax lower bound under squared loss based on the Fano method (see Chapter 15 in Wainwright (2019) for more details). For $\gamma \geq \frac{1}{2}$, we consider the space within a ball as $\mathcal{H}_K^b = \left\{f \in \mathcal{H}_K : \sum_{j=1}^{n} \xi_j^2 \mu_j^{-2\gamma} \leq u^2\right\}$, where $\xi_j$'s are the TA scores associated with $f$.

**Theorem 4.4.** *Suppose that the RKHS is induced by the regular kernel, and $\widetilde{f}$ is any estimator based on the data $\{(\mathbf{x}_i, y_i)\}_{i=1}^n$. If $\frac{1}{2} \leq \gamma \leq 1$, we have*

$$\inf_{\widetilde{f}} \sup_{f^* \in \mathcal{H}_K^b} \mathbb{P}\big(\|\widetilde{f} - f^*\|_n^2 \geq c\delta_n^{4\gamma}\big) \geq \frac{1}{2}.$$

*If $\gamma > 1$, with the choice of $r$ satisfying $r \asymp d(\delta_{n,r}^{1/\gamma}) \leq d(\delta_{n,r})$, we have*

$$\inf_{\widetilde{f}} \sup_{f^* \in \mathcal{H}_K^b} \mathbb{P}\big(\|\widetilde{f} - f^*\|_n^2 \geq c\delta_{n,r}^4\big) \geq \frac{1}{2}.$$

The proof of Theorem 4.4 is provided in Appendix E. For $\gamma > 1$, the condition $d(\delta_{n,r}^{1/\gamma}) \leq d(\delta_{n,r})$ can be easily verified for the most popular polynomial decay case that $\mu_j \asymp j^{-\alpha}$. Specifically, for the kernel class with polynomial decay, we have $d(\delta) \asymp \delta^{-2/\alpha}$, which leads to

$$d(\delta_{n,r}^{1/\gamma}) \asymp \delta_{n,r}^{-2/\alpha} \leq \delta_{n,r}^{-2/\gamma\alpha} \asymp d(\delta_{n,r}).$$

Moreover, it can be seen that $r \asymp \delta_{n,r}^{-2/\gamma\alpha}$ is the optimal choice, aligning with the optimal choice in the upper bound. Note that it is common to establish the upper bound for the other loss function and compare it to the lower bound established under the squared loss to check the optimality (Wei et al., 2017; Lv et al., 2018; Li et al., 2019). By comparing the lower bounds in Theorem 4.4 with the achievable rates from Theorems 3.3 and 4.2, we can conclude that under the case that $\frac{1}{2} \leq \gamma \leq 1$, both the standard kernel-based estimator $\widehat{f}_\lambda$ and the truncated estimator $\widehat{f}_{\lambda,r}$ is minimax-optimal. More importantly, under the more challenging case that $\gamma > 1$, $\widehat{f}_\lambda$ can only achieve a sub-optimal rate, whereas $\widehat{f}_{\lambda,r}$ can attain the minimax rate as long as $r \asymp d(\delta_{n,r}^{1/\gamma}) \leq d(\delta_{n,r})$, suggesting that the truncated kernel-based method can be treated as optimal tackling. It is also worthy pointing out that under the just-aligned regime that $\gamma = \frac{1}{2}$, Yang et al. (2017) derives the minimax lower bound by considering the regular kernel class, and Theorem 4.4 extends it to the more general setting by allowing $\gamma \geq \frac{1}{2}$.

## 5 Numerical Verification

Our established results indicate that a larger $\alpha$ corresponding to a lower model complexity of the RKHS leads to a better rate. As opposed, a smaller model with lower complexity simultaneously may result in a potential mismatch between the model space and the target. This may weaken the target-kernel alignment which undermines the learning efficiency. Consequently, a trade-off exists between model capacity $\alpha$ and target-kernel alignment $\gamma$, with a preference for relatively lower model complexity and stronger target-kernel alignment.

To illustrate this, we conduct some numerical experiments to study how the RKHS with varying model complexities affect the numerical performance of KM and TKM. Specifically, we use the spline kernel with order $\alpha$ that $K_\alpha(\mathbf{x}, \mathbf{x}') = \sum_{k=-\infty}^{\infty} e^{2\pi i k \mathbf{x}} e^{-2\pi i k \mathbf{x}'} |k|^{-\alpha}$ (Wahba, 1990), where $\alpha$ controls the model complexity of the induced RKHS at the population level. Moreover, we consider the nonparametric quantile regression that

$$Y_i = f^*(\mathbf{x}_i) + \sigma(\epsilon_i - \Phi^{-1}(\tau)), \ i = 1, ..., n,$$

where $f^*(\mathbf{x}) = K_{3.5}(\mathbf{x}, 0)\sin(\mathbf{x})$, $\sigma = 2$, $\epsilon_i \sim N(0, 1)$, $\{\mathbf{x}_i\}_{i=1}^n$ are independently sampled from the uniform distribution on $(0, 1)$ and $\Phi$ denotes CDF function of standard normal distribution. We conduct the numerical experiments by varying $\tau \in \{0.3, 0.5, 0.7\}$ and $\alpha \in \{2, 4, 6, 8, 10\}$ with fixed $n = 300$. The data generating scheme is repeated for 50 times and all the tuning parameters are tuned to the best for both methods. The obtained results are presented in Figure 1.

From Figure 1, we can conclude that the smaller $\alpha$, corresponding to richer RKHS and potentially stronger alignment, results in significant improvement in TKM over KM. Conversely, the larger $\alpha$, corresponding to a smaller RKHS and potentially weaker alignment, results in a comparable performance for these two methods. This observation precisely aligns with our theoretical results. Clearly, based on our theoretical findings, the experiment results verify the existence of a trade-off between the model complexity and target-kernel alignment, indicating that a carefully data-driven

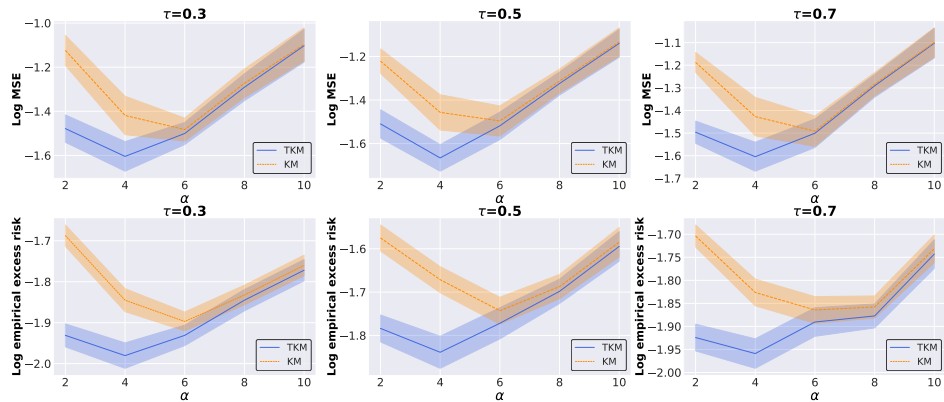

Figure 1: Averaged log MSE and log empirical excess risk for KM and TKM versus $\alpha$ for different $\tau$.

choice of the kernel may be necessary to achieve better learning efficiency. We defer the deeper exploration of data-driven selection of an appropriate kernel to future research endeavors.

The real data analysis is deferred to Appendix A. Furthermore, a variety of additional experiments are presented in Appendix H. The obtained results are discussed in detail, which further supports our theoretical findings.

# 6 Discussions and Conclusion

## 6.1 Comparison and Discussions

Amini et al. (2022) studied how the target-kernel alignment affects both the standard KRR and the truncated KRR. Although our work is motivated by Amini et al. (2022), especially for the methodological aspect, there exist significant differences between our established results and those in Amini et al. (2022), and some are summarized as follows: (a) Amini et al. (2022) only focused on the upper bounds in terms of expected mean squared error, while our results provide more precise high-probability upper bounds. (b) Beyond the polynomial decay condition considered in Amini et al. (2022), we introduce a more general condition as stated in Assumption 3.2. This condition involves $\gamma$, reflecting the degree of target-kernel alignment. (c) In Amini et al. (2022), both the standard KRR and the truncated KRR have explicit solutions. This allows leveraging analytic solutions to establish critical results, without requiring more advanced techniques. In contrast, no explicit solutions exist in our case and our theoretical analysis adopts an alternative analytic treatment by utilizing kernel complexity and empirical process techniques. (d) Last but not least, we rigorously confirm the conjecture in Amini et al. (2022) asserting that the truncated KRR can achieve the minimax optimality for all $\gamma \geq \frac{1}{2}$.

## 6.2 Conclusion and Future Work

This paper provides a comprehensive theoretical understanding of the properties of the truncated kernel-based method for a broad family of loss functions. By using kernel complexity and empirical process techniques, the established results reveal some significant benefits from the truncated RKHS and indicate that a carefully chosen truncation allows for an optimal trade-off between the model complexity and approximation bias. Extensive numerical studies further justify our theoretical findings, demonstrating a consistent improvement of the truncated estimator over the standard kernel-based estimator. We also derived an algorithm-free minimax lower bound that matches the upper bound on the truncated estimator and therefore confirmed its optimality. To some extent, our results shed light on future research in statistical learning theory and real-world applications. This paper also leaves several interesting open questions for future investigation, including the theoretical explorations under the misspecified setting that $0 < \gamma < \frac{1}{2}$ and how to develop a data-driven algorithm for selecting a strongly-aligned kernel with lower model complexity.

## Acknowledgements

CW's research is supported by the Fundamental Research Funds for the Central Universities. XH's research is sponsored by Natural Science Foundation of Shanghai (24ZR1421400), NSFC-11901375, Shanghai Science and Technology Development Funds (23JC1402100), Program for Innovative Research Team of Shanghai University of Finance and Economics and Shanghai Research Center for Data Science and Decision Technology. JW's research is supported in part by HK RGC Grant GRF-14303424 and CUHK Startup Grant 4937091.

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

## Appendix

This appendix is organized as follows. In Section A, we conduct a real data analysis. Section B is devoted to providing more discussions and future directions. In Section C, we provide the proof for results in Section 3, including Theorem 3.3 and Corollary 3.5. In Section D, we provide the proof for results in Section 4 in part, including Theorem 4.2 and Corollary 4.3. In Section E, we complement the upper bounds by deriving the minimax lower bound. In Section G, we discuss the locally strong convexity condition and local smoothness condition presented in Assumption 3.2 for various loss functions in detail. In Section G, we list some useful lemmas utilized in our proofs, including concentration inequality, symmetrization inequality, and Gaussian contraction inequality. Section H provides additional experiments under various settings.

**Notation.** Denote the vector inner product $\langle \boldsymbol{\alpha}, \boldsymbol{\beta} \rangle_2 = \sum_{i=1}^n \alpha_i \beta_i$ and the norm $\|\boldsymbol{\alpha}\|_2 = \sqrt{\langle \boldsymbol{\alpha}, \boldsymbol{\alpha} \rangle_2}$ for $\boldsymbol{\alpha}, \boldsymbol{\beta} \in \mathcal{R}^n$. For any integer $m$, we use $[m]$ to represent the set $\{1, 2, ..., m\}$ for short.

## A    Real Data Analysis

We apply both TKM and KM with check loss to the wine quality dataset, which is available in the UCI Machine Learning Repository. Specifically, we first adopt the random forest method (Breiman, 2001) to rank the feature importance and select the first three influential features: 'Alcohol', 'Sulfates', and 'Volatile Acidity' for analysis. Then, we randomly select 500 samples for training and another 500 samples for testing. The above procedure is repeated 10 times, where the Laplacian kernel $K(\mathbf{x}, \mathbf{x}') = \exp(-\|\mathbf{x} - \mathbf{x}'\|_1)$ is adopted and the parameters $\gamma$ and $r$ are tuned by 5-fold cross-validation. The averaged MSE with different $\tau \in (0.3, 0.5, 0.7)$ is reported in Table 2.

Table 2: Averaged MSE for different methods

| $\tau$ | 0.3 | 0.5 | 0.7 |
|---|---|---|---|
| KM | $0.590 \pm 0.027$ | $0.483 \pm 0.035$ | $0.638 \pm 0.073$ |
| TKM | $0.548 \pm 0.026$ | $0.454 \pm 0.045$ | $0.530 \pm 0.046$ |

It is thus clear that the obtained results in the real application align with the results for synthetic data and our theoretical findings in the main text, which further demonstrates the benefits of TKM.

## B    More Discussions and Future Directions

### B.1    Verification of the Existence and Uniqueness of $\delta_n$ and $\delta_{n,r}$

In this section, we provide a detailed verification of the existence and uniqueness of $\delta_n$ and $\delta_{n,r}$, where $\delta_n$ and $\delta_{n,r}$ are defined as the smallest solutions to (2) and (3), respectively.

For any $\iota \in (0, 1)$ and $\gamma \geq \frac{1}{2}$, define

$$g(\delta) := \frac{2C \log \iota^{-1} R(\delta)}{c_0 \delta^{2\eta + 1}} \quad \text{on} \quad \delta \in (0, \infty),$$

where $\eta = \max\{\gamma, 1\}$. Here, we ignore the dependence of $g$ on $\iota$ and $\gamma$ for ease of presentation.

The verifying argument is based on Lemma 3.2 in Bartlett et al. (2005), which states that if $\psi : [0, \infty) \to [0, \infty)$ is a nontrivial sub-root function[6], then it is continuous on $[0, \infty)$, and the equation $\psi(r) = r$ has a unique positive solution.

Recall that we assume $\mu_1, ..., \mu_n > 0$. From the definition of $R(\delta)$, it can be easily seen that $R(\sqrt{\delta})$ is a nontrivial sub-root function. Then from Lemma 3.2 in Bartlett et al. (2005), $R(\sqrt{\delta})$ is continuous on $[0, \infty)$, and thus $g(\delta)$ is also continuous on $(0, \infty)$. Note that $g(\delta)\delta^{2\eta}$ is nonincreaing on $(0, \infty)$, then $g(\delta)$ must be strictly decreasing $(0, \infty)$.

---

[6]A function $\psi$ is called nontrivial sub-root if $\psi \not\equiv 0$, and it is nonnegative, nondecreasing and if $\psi(r)/\sqrt{r}$ is nonincreasing for $r > 0$.

If $g(\delta)$ is always larger than 1 on $(0, \infty)$, we have

$$\lim_{\delta \to \infty} g(\delta)\delta^{2\eta} = \infty,$$

which is impossible since $g(\delta)\delta^{2\eta}$ is nonincreasing on $(0, \infty)$. On the other hand, if $g(\delta)$ is always smaller than 1 on $(0, \infty)$, then

$$\lim_{\delta \to 0} g(\delta)\delta^{2\eta} = 0,$$

implying $g(\delta)\delta^{2\eta} \equiv 0$ since $g(\delta)\delta^{2\eta}$ is nonincreasing. Since $R(\delta)$ is not trivial, this is also impossible.

Therefore, by the continuity of $g$ on $(0, 1)$, the equation $g(\delta) = 1$ has a positive solution $\delta_n$ that is unique by the strict monotonicity of $g$. Note that by the strict monotonicity of $g$, $\delta_n$ can be equivalently defined as the smallest solution to $g(\delta) \leq 1$. According to the definition of $g$, we conclude the existence and uniqueness of $\delta_n$. For $\delta_{n,r}$, repeat a similar argument, we can also verify its existence and uniqueness.

### B.2 Decomposition of the Reduced Kernel Matrix

Recall that $\mathbf{K}_r = \left\{ \frac{1}{n} K_r(\mathbf{x}_i, \mathbf{x}_j) \right\}_{i,j=1}^n$ is the empirical kernel matrix w.r.t. $K_r$, and $\mathbf{K} = \mathbf{U}\mathbf{D}\mathbf{U}^\top$, where $\mathbf{U} = (\mathbf{u}_1, ..., \mathbf{u}_n) \in \mathcal{R}^{n \times n}$ is an orthonormal matrix and $\mathbf{D} \in \mathcal{R}^{n \times n}$ is a diagonal matrix with positive elements $\mu_1, ..., \mu_n$ arranging in a descending ordering. Denote $\mathbf{u}_k = (u_{k1}, ..., u_{kn})^\top$. By the definitions of $K_r$ and $\{\psi_k\}_{i \in [n]}$, we have

$$(\mathbf{K}_r)_{ij} = \frac{1}{n} \sum_{k=1}^r \mu_k \psi_k(\mathbf{x}_i) \psi_k(\mathbf{x}_j) = \sum_{k=1}^r \mu_k u_{ki} u_{kj}.$$

On the other hand, recall that $\mathbf{D}_r$ is diagonal matrix with elements $\mu_1, ..., \mu_r, 0, ..., 0$, we have

$$(\mathbf{U}\mathbf{D}_r\mathbf{U}^\top)_{ij} = \sum_{k=1}^n \mathbf{D}_{kk} u_{ki} u_{kj} = \sum_{k=1}^r \mu_k u_{ki} u_{kj} = (\mathbf{K}_r)_{ij}.$$

Therefore, we conclude that $\mathbf{K}_r = \mathbf{U}\mathbf{D}_r\mathbf{U}^\top$.

### B.3 Extension to Random Design Setting

In this work, we investigate the effect of target-kernel alignment and provide the best solution to overcome the well-known saturation effect. It is also worthy pointing out that although the obtained results are derived under the fixed design setting, it may be possible to extend them to the random design setting and we leave it to future exploration. Some key steps of the possible extensions are discussed below.

Under the random design setting, we consider a random variable $X \sim \rho$, where $\rho$ is a probability measure supported on $\mathcal{X} \subset \mathcal{R}^p$. Let the covariates $\{\mathbf{x}_i\}_{i=1}^n$ be independently sampled from $\rho$. Denote the space of square-integrable functions $f : \mathcal{X} \to \mathcal{R}$ with respect to $\rho$ as $\mathcal{L}(\mathcal{X}, \rho)$, where $\mathcal{X} \subset \mathcal{R}^p$.

Recall that $\mathcal{H}_K$ is an RKHS induced by a positive semi-definite kernel function $K$. By Mercer's theorem (see, for instance, Theorem 12.20 in Wainwright (2019)), if $\mathcal{X}$ is compact and $K$ is a continuous, combined with our bounded assumption that $\sup_{\mathbf{x}, \mathbf{x}' \in \mathcal{X}} K(\mathbf{x}, \mathbf{x}') \leq \kappa^2$, the kernel function admits an expansion of form

$$K(\mathbf{x}, \mathbf{x}') = \sum_{j=1}^\infty \tilde{\mu}_j \phi_j(\mathbf{x}) \phi_j(\mathbf{x}'),$$

where $\tilde{\mu}_j$'s are the non-negative eigenvalues in descending ordering and $\phi_j$'s are the corresponding eigenfunctions in $\mathcal{L}(\mathcal{X}, \rho)$. Given this expansion of $K$, the RKHS $\mathcal{H}_K$ can be written as

$$\mathcal{H}_K = \left\{ f = \sum_{j=1}^\infty \alpha_j \phi_j : \sum_{j=1}^\infty \frac{\alpha_j^2}{\tilde{\mu}_j} < \infty \right\},$$

equipped with inner product $\langle f, g \rangle_K = \sum_{j=1}^\infty \frac{\alpha_j \beta_j}{\tilde{\mu}_j}$ for $f = \sum_{j=1}^\infty \alpha_j \phi_j$ and $g = \sum_{j=1}^\infty \beta_j \phi_j$.

Under the random design setting, the population risk function is defined as

$$\mathcal{E}(f) := \mathbb{E}\big[L\big(Y, f(X)\big)\big],$$

where $Y|X = \mathbf{x} \sim \mathbb{P}_{Y|\mathbf{x}}$. Then, the target function $f^*$ is defined as

$$f^* := \operatorname{argmin}_f \mathcal{E}(f).$$

If we assume $f^* \in \mathcal{H}_K$, then $f^*$ can be expanded as $f^* = \sum_{j=1}^{\infty} \alpha_j^* \phi_j$ with $\boldsymbol{\alpha}^* = (\alpha_1^*, \alpha_2^*, ...)^\top$ satisfying $\sum_{j=1}^{n} \frac{\alpha_j^{*2}}{\tilde{\mu}_j} < \infty$.

An assumption analogous to Assumption 3.2 under the random design setting is required that $\sum_{j=1}^{\infty} \alpha_j^{*2} \tilde{\mu}_j^{-2\gamma} \leq u^2$ for some constants $\gamma \geq \frac{1}{2}$ and $u$. Here, $\gamma$ measures the target-kernel alignment at the population level, and it is equivalent to smooth (or source) parameter in literature (Caponnetto & De Vito, 2007; Cui et al., 2021; Li et al., 2023). Note that if $\gamma = \frac{1}{2}$, we merely assume that the target function $f^*$ belongs to the RKHS $\mathcal{H}_K$, and as $\gamma$ increases, the target function becomes smoother w.r.t. the RKHS $\mathcal{H}_K$.

Furthermore, the polynomial decay assumption analogous to Assumption 3.4 under the random design setting can be made as

$$\tilde{\mu}_j \asymp j^{-\alpha} \quad \text{and} \quad \alpha_j^{*2} \asymp j^{-2\gamma\alpha - 1} \tag{4}$$

with constants $\alpha > 1$ and $\gamma \geq \frac{1}{2}$. Note that the assumption (4) is equivalent to condition (8) in Cui et al. (2021).

Grant these assumptions, the impact of the target-kernel alignment on both standard and truncated kernel-based methods under the random design setting can be analyzed by using the population kernel complexity, defined as

$$\widetilde{R}(\delta) := \sqrt{\frac{1}{n} \sum_{j=1}^{\infty} \min\{\delta^2, \tilde{\mu}_j\}}.$$

Since the theoretical derivation should be more deeply involved, we leave this promising topic to future investigation.

### B.4  Connection with Spectrally Transform Kernel Regression

The spectrally transformed kernel regression (SKRR, Zhai et al. (2024)) aims to use spectrally transformation for constructing a new kernel that can leverage the information contained in unlabeled data in an explicit way. Note that we have shown that the truncated kernel method can overcome the saturation effect thanks to the reduced kernel complexity. We also believe that SKRR may be able to overcome the saturation effect if the transformation function can be properly chosen. The possible routine for establishing the theoretical results is briefly discussed below.

Recall that by Mercer's theorem, we have

$$K(\mathbf{x}, \mathbf{x}') = \sum_{j=1}^{\infty} \tilde{\mu}_j \phi_j(\mathbf{x}) \phi_j(\mathbf{x}').$$

For SKRR, $K(\mathbf{x}, \mathbf{x}')$ is replaced with a new kernel that

$$K'(\mathbf{x}, \mathbf{x}') := \sum_{j=1}^{\infty} s(\tilde{\mu}_j) \phi_j(\mathbf{x}) \phi_j(\mathbf{x}'),$$

where $s(\cdot) : \mathcal{R}^+ \to \mathcal{R}^+$ is the general transformation function. Let $\widehat{f}_{\lambda,s}$ be the kernel-based estimator via the RKHS $\mathcal{H}_{K'}$ induced by $K'$. The primary goal is to study how the prediction error $\|\widehat{f}_{\lambda,s} - f^*\|_\rho^2$ depends on the choice of $s$.

The idea of deriving an upper bound on the prediction error is to separately bound the estimation error $\|\widehat{f}_{\lambda,s} - f_s^\sharp\|_\rho^2$ and approximation bias $\|f_s^\sharp - f^*\|_\rho^2$, where $\|\cdot\|_\rho$ denotes the norm equipped

with $\mathcal{L}(\mathcal{X}, \rho)$, and $f_s^\sharp := \sum_{j=1}^\infty s(\alpha_j^*)\phi_j$ is introduced as an immediate function belonging to $\mathcal{H}_{K'}$. Following a similar technical treatment in Section D, the upper bound on estimation error can be established. For the approximation bias, by writing $f^* = \sum_{j=1}^\infty \alpha_j^* \phi_j$, we find that

$$\left\| f_s^\sharp - f^* \right\|_\rho^2 = \sum_{j=1}^\infty \left( s(\alpha_j^*) - \alpha_j^* \right)^2.$$

Clearly, the selection of $s(\cdot)$ is crucial and it is favorable if $s(\cdot)$ is close to the identity function for small $j$ and decays extremely rapidly as $j$ tends to infinity, such as $s(\tilde{\mu}_j) = \tilde{\mu}_j \mathbf{I}_{\{j \leq r\}}$ corresponding to the truncated method. Then, SKRR with some proper choices of $s(\cdot)$ may achieve similar conclusions about the upper bound as we provided in the main text. We leave such a promising topic as potential future work.

### B.5 Computational Complexity of Truncated Kernel Method

For simplicity, we focus only on the mean regression task where the squared loss is specified. Note that the total computational complexity of the truncated KRR is composed of three parts. Specifically, in the first part, spectrally decomposing the kernel matrix $\mathbf{K}$ has the computational complexity of $\mathcal{O}(n^3)$. In the second part, the basis $\{\psi_k\}_{1 \leq k \leq r}$ can be simply calculated by $\psi_k(\mathbf{x}) = \mathbf{u}_k^\top \mathbf{K}^{-1} \mathbf{K}_\mathbf{x}$ with

$$\mathbf{K}_\mathbf{x} = \frac{1}{\sqrt{n}} \big( K(\mathbf{x}, \mathbf{x}_1), ..., K(\mathbf{x}, \mathbf{x}_n) \big)^\top$$

for each $k \in [n]$, which also has $\mathcal{O}(n^3)$ computational complexity. In the last part, computing the KRR via the $r$-dimensional RKHS $\mathcal{H}_{K_r}$ has computational complexity of $\mathcal{O}(nr^2)$. To sum up, the overall computational complexity of TKM is $\mathcal{O}(n^3)$. Solving the standard KRR also has computational complexity of $\mathcal{O}(n^3)$, and thus the truncated KRR does not impose an additional computational cost.

## C Proof of Results for Kernel-based Method

### C.1 Error Analysis

For ease of presentation, without loss of generality, we assume $\mu_j \leq 1$ for all $j \in [n]$ in the rest of this paper[7]. We start the error analysis by noting that for any $f = S_\mathbf{x}^\top(\boldsymbol{\alpha}) \in \mathcal{H}_K$ with $\boldsymbol{\alpha} \in \mathcal{R}^n$, we have

$$\|f\|_n = \|\mathbf{K}\boldsymbol{\alpha}\|_2 \quad \text{and} \quad \|f\|_K = \left\| \mathbf{K}^{1/2} \boldsymbol{\alpha} \right\|_2. \tag{5}$$

This transform from the $\mathcal{L}(\mathbb{P}_n)$-norm and the norm in RKHS to vector norm will be frequently applied in our proof.

Denote

$$\mathbf{q}^* := \sqrt{n} S_\mathbf{x}(f^*) = (f^*(\mathbf{x}_1), ..., f^*(\mathbf{x}_n))^\top.$$

Recall that the TA scores are the elements of the vector

$$\xi^* = \mathbf{U}^\top S_\mathbf{x}(f^*) = \frac{1}{\sqrt{n}} \mathbf{U}^\top \mathbf{q}^*.$$

Define an immediate function

$$f^\sharp := S_\mathbf{x}^\top(\boldsymbol{\alpha}^\sharp) \in \mathcal{H}_K \quad \text{with} \quad \boldsymbol{\alpha}^\sharp = \mathbf{U} \mathbf{D}^{-1} \xi^* \in \mathcal{R}^n.$$

$f^\sharp$ can be viewed as the best approximation of $f^*$ onto the $n$-dimensional function space $\mathcal{H}_n$, defined as

$$\mathcal{H}_n := \left\{ f = S_\mathbf{x}^\top(\boldsymbol{\alpha}) : \boldsymbol{\alpha} \in \mathcal{R}^n \right\}.$$

---

[7]Otherwise, by the assumption $\mu_j \to 0$, we always have $\mu_j \leq 1$ for $j$ exceeding some constant $j^*$.

To be more clear, by the orthogonality of $\mathbf{U}$, we have

$$
\begin{aligned}
\left\| f^* - f^\sharp \right\|_n^2 &= \frac{1}{n} \left\| \mathbf{q}^* - \sqrt{n}\, \mathbf{K}\, \boldsymbol{\alpha}^\sharp \right\|_2^2 \\
&= \left\| n^{-1/2} \mathbf{U}^\top \mathbf{q}^* - \mathbf{U}^\top \mathbf{U}\, \mathbf{D}\, \mathbf{U}^\top \boldsymbol{\alpha}^\sharp \right\|_2^2 \\
&= \left\| \mathbf{U}^\top S_{\mathbf{x}}(f^*) - \mathbf{U}^\top \mathbf{U}\, \mathbf{D}\, \mathbf{U}^\top \boldsymbol{\alpha}^\sharp \right\|_2^2 \\
&= \left\| \xi^* - \mathbf{D}\, \mathbf{U}^\top \boldsymbol{\alpha}^\sharp \right\|_2^2 = \left\| \xi^* - \mathbf{D}\, \mathbf{U}^\top \mathbf{U}\, \mathbf{D}^{-1} \xi^* \right\|_2^2 = 0,
\end{aligned}
\tag{6}
$$

implying

$$
f^*(\mathbf{x}_i) = f^\sharp(\mathbf{x}_i) \quad \text{for each} \quad i \in [n].
$$

Therefore, we obtain

$$
\left\| f - f^* \right\|_n = \left\| f - f^\sharp \right\|_n \quad \text{for all} \quad f \in \mathcal{H}_K
\tag{7}
$$

and

$$
\mathcal{E}(f) - \mathcal{E}(f^*) = \mathcal{E}(f) - \mathcal{E}(f^\sharp) \quad \text{for all} \quad f \in \mathcal{H}_K.
\tag{8}
$$

Furthermore, by applying (5), we have

$$
\left\| f^\sharp \right\|_K = \left\| \mathbf{K}^{1/2}\, \mathbf{U}\, \mathbf{D}^{-1} \xi^* \right\|_K = \left\| \mathbf{U}\, \mathbf{D}^{1/2} \mathbf{U}^\top \mathbf{U}\, \mathbf{D}^{-1} \xi^* \right\|_K = \left\| \mathbf{D}^{-1/2} \xi^* \right\|_K.
$$

By our assumption that $\mu_j \leq 1$ for all $j$, one has

$$
\left\| f^\sharp \right\|_K^2 = \left\| \mathbf{D}^{-1/2} \xi^* \right\|_2^2 = \sum_{j=1}^n \mu_j^{-1} \xi_j^{*2} \leq \sum_{j=1}^n \mu_j^{-2\gamma} \xi_j^{*2} \leq u^2,
\tag{9}
$$

where the last inequality holds by Assumption 3.2 with $\gamma \geq \frac{1}{2}$.

The construction of $f^\sharp$ plays a crucial role in our proofs. Intuitively, the kernel complexity function $R(\delta)$ is defined at an empirical level (depends on the fixed points $\mathbf{x}_1, ..., \mathbf{x}_n$) and serves as a complexity measure of finite space $\mathcal{H}_n$. This poses a technical challenge as the true function $f^*$ lies in an infinite-dimensional function space, creating a mismatch with the empirical kernel complexity $R(\delta)$. To solve this problem, we introduce the best approximation $f^\sharp$ of $f^*$ in finite-dimensional function space $\mathcal{H}_n$. Our proof will first focus on deriving the upper bound on $\|\widehat{f}_\lambda - f^\sharp\|_n^2$, and then move to $\|\widehat{f}_\lambda - f^*\|_n^2$ by using the relation (7).

Another advantage to consider the best approximation of $f^*$ instead of itself is that $\widehat{f}_\lambda$ lies in the same space $\mathcal{H}_n$ as $f^\sharp$, allowing us to express $\|\widehat{f}_\lambda - f^\sharp\|_2^2$ and $\|\widehat{f}_\lambda - f^\sharp\|_K^2$ in the term of the kernel matrix $\mathbf{K}$ according to (5), which is useful in the technical proof. To the best of our knowledge, this is a novel treatment to establish theoretical results for the kernel-based estimator. In the proof for the truncated estimator in Section D, we will adopt a similar proof strategy to construct the best approximation $f_r^\sharp$ of $f^*$ in the reduced space $\mathcal{H}_{K_r}$.

Based on the error analysis, we are ready to present the proof for Theorem 3.3.

## C.2 Proof of Theorem 3.3

Define the localized function class

$$
\mathcal{H}_{n,b} := \left\{ f : f \in \mathcal{H}_n, \ \|f - f^\sharp\|_K \leq b \right\}.
$$

Here, $b$ is a constant independent of $n, \gamma$, which will be specified in the proof. Without loss of generality, we assume $b > 1$.

For any $\iota \in (0,1)$ and $\delta > 0$, define the auxiliary event

$$
\mathcal{V}(\iota, \delta) := \left\{ \left| \widehat{\mathcal{E}}(f) - \widehat{\mathcal{E}}(f^\sharp) - [\mathcal{E}(f) - \mathcal{E}(f^\sharp)] \right| \leq C \log \iota^{-1} R(\delta) W(f, \delta) \ \text{ holds for any } f \in \mathcal{H}_{n,b} \right\},
$$

where $W(f, \delta) := \delta^{-1} \|f - f^\sharp\|_n + \|f - f^\sharp\|_K$ for $\delta > 0$ and $f \in \mathcal{H}_{n,b}$.

The following two lemmas are crucial for proving Theorem 3.3.

**Lemma C.1.** *Fix any $\iota \in (0,1)$ and $\delta > 0$. The event $\mathcal{V}(\iota, \delta)$ occurs with probability greater than $1 - \iota$, i.e.*

$$\mathbb{P}\big(\mathcal{V}(\iota, \delta)\big) \geq 1 - \iota.$$

As demonstrated in the proof of Lemma C.1, $b$ is incorporated into the constant $C$ of the upper bound $C \log \iota^{-1} R(\delta)$, meaning that $C$ depends on $b$.

Recall that $\delta_n$ is the critical radius defined as the smallest solution to (2).

**Lemma C.2.** *Let $\eta = \min\{\gamma, 1\}$. On the event $\mathcal{V}(\iota, \delta_n)$, with the choice of $\lambda$ satisfying $\delta_n^2 \leq \lambda \leq 1$, we have*

$$\big\| \widehat{f}_\lambda - f^\sharp \big\|_n^2 \leq C \left( \delta_n^{4\eta} + \lambda^{2\eta} \right),$$

*where $C$ is a constant independent of $n, \gamma$.*

*Proof of Theorem 3.3.* By applying Lemma C.1, we have $\mathbb{P}(\mathcal{V}(\iota, \delta_n)) \geq 1 - \iota$. Together with Lemma C.2 and the relation (6), it holds with probability at least $1 - \iota$ that

$$\big\| \widehat{f}_\lambda - f^* \big\|_n^2 = \big\| \widehat{f}_\lambda - f^\sharp \big\|_n^2 \leq C \left( \delta_n^{4\eta} + \lambda^{2\eta} \right),$$

which completes the proof for the $\mathcal{L}(\mathbb{P}_n)$-error.

For the excess risk, it immediately follows from Assumption 3.1. $\qquad\square$

Accordingly, it remains to prove Lemmas C.1 and C.2.

### C.2.1 Proof of Lemma C.1

Denote

$$\mathcal{D} := \widehat{\mathcal{E}}(f) - \widehat{\mathcal{E}}(f^\sharp) - \left[ \mathcal{E}(f) - \mathcal{E}(f^\sharp) \right].$$

Then, our goal is to prove that for all $f \in \mathcal{H}_{n,b}$

$$|\mathcal{D}| \leq C \log \iota^{-1} R(\delta) W(f, \delta).$$

If $W(f, \delta) = 0$, the above inequality is naturally satisfied. Therefore, without loss of generality, we assume $W(f, \delta) > 0$ for all $f \in \mathcal{H}_{n,b}$. It is equivalent to proving that

$$\mathcal{A} := \sup_{f \in \mathcal{H}_{n,b}} \frac{|\mathcal{D}|}{W(f, \delta)} \leq C \log \iota^{-1} R(\delta).$$

By applying the triangle inequality, together with (9), we find that for any $f \in \mathcal{H}_{n,b}$

$$\|f\|_K \leq \big\| f - f^\sharp \big\|_K + \|f^\sharp\|_K \leq u + b.$$

Let $\tilde{b} := u + b$. According to our assumption for the loss function in Section 2, $L(y, \cdot)$ satisfies the Lipschitz continuity over the function class $\mathcal{H}_{n,b}$ with Lipschitz constant $M_{L,\tilde{b}}$ in the sense that for any $y \in \mathcal{Y}$, $\mathbf{x} \in \mathcal{X}$, and $f, f' \in \mathcal{H}_{n,b}$, the following inequality holds:

$$\big| L(y, f(\mathbf{x})) - L(y, f'(\mathbf{x})) \big| \leq M_{L,\tilde{b}} \big| f(\mathbf{x}) - f'(\mathbf{x}) \big|.$$

For simplifying notation, we hide the dependence of the Lipschitz constant on $L(\cdot, \cdot), \tilde{b}$ by writing $M := M_{L,\tilde{b}}$.

The remaining proof follows a standard procedure: first bound the expectation of $\mathcal{A}$ and then bound the deviation of $\mathcal{A}$ from its expectation. Finally, we combine these two bounds to obtain the desired result.

**Bounding $\mathbb{E}[\mathcal{A}]$.** Let $w_1, ..., w_n \sim N(0, 1)$ denote the standard Gaussian variables, independent of the data. To bound $\mathbb{E}[\mathcal{A}]$, we employ the symmetrization technique in Lemma G.2. Specifically, we

have

$$\mathbb{E}[\mathcal{A}] = \mathbb{E}\Big[\sup_{f \in \mathcal{H}_{n,b}} \frac{|\mathcal{D}|}{W(f,\delta)}\Big]$$

$$\overset{(i)}{\leq} \frac{\sqrt{2\pi}}{n}\mathbb{E}\Big[\sup_{f \in \mathcal{H}_{n,b}} \frac{\big|\sum_{i=1}^{n} w_i\big(L(y_i, f(\mathbf{x}_i)) - L(y_i, f^\sharp(\mathbf{x}_i))\big)\big|}{W(f,\delta)}\Big]$$

$$\overset{(ii)}{\leq} \frac{2\sqrt{2\pi}M}{n}\mathbb{E}\Big[\sup_{f \in \mathcal{H}_{n,b}} \frac{\big|\sum_{i=1}^{n} w_i(f(\mathbf{x}_i) - f^\sharp(\mathbf{x}_i))\big|}{W(f,\delta)}\Big], \qquad (10)$$

where (i) follows from Lemma G.2, and (ii) follows from the fact that the loss function is $M$-Lipschitz continuous and the Gaussian contraction inequality in Lemma G.3. To further derive the upper bound for the RHS of (10), we consider the localized function class of form

$$\mathcal{F}(\delta) := \Big\{f = S_{\mathbf{x}}^\top(\boldsymbol{\alpha}) : \ \big\|f - f^\sharp\big\|_K \leq 1, \ \big\|f - f^\sharp\big\|_n \leq \delta, \ \boldsymbol{\alpha} \in \mathcal{R}^n\Big\}.$$

Recall that $f^\sharp = S_{\mathbf{x}}^\top(\boldsymbol{\alpha}^\sharp)$ and for any $f \in \mathcal{H}_{n,b}$, there exists $\boldsymbol{\alpha} \in \mathcal{R}^n$ such that $f = S_{\mathbf{x}}^\top(\boldsymbol{\alpha})$.

Define the vector $\boldsymbol{\beta} := \mathbf{D}\,\mathbf{U}^\top(\boldsymbol{\alpha} - \boldsymbol{\alpha}^\sharp)$. Then, by applying (5), $f \in \mathcal{F}(\delta)$ implies the constraints on $\boldsymbol{\beta}$ that

$$\big\|\mathbf{D}^{-1/2}\boldsymbol{\beta}\big\|_2 \leq 1 \quad \text{and} \quad \|\boldsymbol{\beta}\|_2 \leq \delta.$$

Further note that any vector satisfying these two constraints must belong to the ellipse class

$$\mathcal{E} := \Big\{\boldsymbol{\beta} = (\beta_1, \beta_2, \dots)^\top \in \mathcal{R}^n : \sum_{j=1}^{n} \frac{\beta_j^2}{\nu_j} \leq 2 \text{ with } \nu_j = \min\{\delta^2, \mu_j\}\Big\}.$$

Denote $\mathbf{w} = (w_1, ..., w_n)^\top$, we have

$$\mathbb{E}\Big[\sup_{f \in \mathcal{F}(\delta)} \Big|\sum_{i=1}^{n} w_i(f(\mathbf{x}_i) - f^\sharp(\mathbf{x}_i))\Big|\Big] \leq \mathbb{E}\Big[\sup_{\boldsymbol{\beta} \in \mathcal{E}} \sqrt{n}\big|\langle \mathbf{w}, \mathbf{U}\boldsymbol{\beta}\rangle\big|\Big]$$

$$\overset{(i)}{=} \mathbb{E}\Big[\sup_{\boldsymbol{\beta} \in \mathcal{E}} \sqrt{n}\big|\langle \mathbf{w}, \boldsymbol{\beta}\rangle\big|\Big]$$

$$\overset{(ii)}{\leq} \mathbb{E}\Big[\sup_{\boldsymbol{\beta} \in \mathcal{E}} \sqrt{n}\sqrt{\sum_{j=1}^{n} \frac{\beta_j^2}{\nu_j}}\sqrt{\sum_{j=1}^{n} \nu_j w_j^2}\Big]$$

$$\leq \sqrt{2n}\mathbb{E}\Big[\sqrt{\sum_{j=1}^{n} \nu_j w_j^2}\Big] \overset{(iii)}{\leq} \sqrt{2n}\sqrt{\frac{1}{n}\sum_{j=1}^{n} \nu_j} = \sqrt{2n}R(\delta),$$

$$(11)$$

where (i) follows $\langle \mathbf{w}, \mathbf{U}\boldsymbol{\beta}\rangle = \langle \mathbf{U}^\top \mathbf{w}, \boldsymbol{\beta}\rangle$ and $\mathbf{U}^\top \mathbf{w} \sim N(0, \mathbf{I}_n)$ since $\mathbf{U}^\top$ is an orthogonal matrix, (ii) follows from Cauchy-Schwarz inequality, and (iii) follows from Jensen's inequality.

Note that (11) holds when considering the supremum over $\mathcal{F}(\delta)$. For $f \in \mathcal{H}_{n,b}$, by defining the rescaled function

$$\widetilde{f} = \frac{f - f^\sharp}{W(f,\delta)} + f^\sharp,$$

we have

$$\big\|\widetilde{f} - f^\sharp\big\|_n = \frac{\|f - f^\sharp\|_n}{\delta^{-1}\|f - f^\sharp\|_n + \|f - f^\sharp\|_K} \leq \delta$$

and

$$\big\|\widetilde{f} - f^\sharp\big\|_K = \frac{\|f - f^\sharp\|_K}{\delta^{-1}\|f - f^\sharp\|_n + \|f - f^\sharp\|_K} \leq 1.$$

As a result, $\widetilde{f} \in \mathcal{F}(\delta)$. On the other hand,

$$\mathbb{E}\Big[\frac{\big|\sum_{i=1}^{n} w_i(f(\mathbf{x}_i) - f^\sharp(\mathbf{x}_i))\big|}{W(f, \delta)}\Big] = \mathbb{E}\Big[\big|\sum_{i=1}^{n} w_i(\widetilde{f}(\mathbf{x}_i) - f^\sharp(\mathbf{x}_i))\big|\Big],$$

which, combined with (10) and (11), implies

$$\begin{aligned}
\mathbb{E}[\mathcal{A}] &\leq \frac{2\sqrt{2\pi}M}{n} \mathbb{E}\Big[\sup_{f \in \mathcal{H}_{n,b}} \frac{\big|\sum_{i=1}^{n} w_i(f(\mathbf{x}_i) - f^\sharp(\mathbf{x}_i))\big|}{W(f, \delta)}\Big] \\
&\leq \frac{2\sqrt{2\pi}M}{n} \mathbb{E}\Big[\sup_{f \in \mathcal{F}(\delta)} \big|\sum_{i=1}^{n} w_i(f(\mathbf{x}_i) - f^\sharp(\mathbf{x}_i))\big|\Big] \leq 4\sqrt{\pi}MR(\delta).
\end{aligned} \tag{12}$$

**Bounding $\mathcal{A} - \mathbb{E}(\mathcal{A})$.** We use the concentration inequality in Lemma G.1 to bound $\mathcal{A} - \mathbb{E}(\mathcal{A})$. For each $i \in [n]$ and any $f \in \mathcal{F}(\delta)$, define $s_j = \text{sign}(f(\mathbf{x}_i) - f^\sharp(\mathbf{x}_i))$ if $j = i$ and $s_j = 0$ if $j \neq i$ and let $\mathbf{s} = (s_1, ..., s_n)^\top$, then we have

$$\begin{aligned}
|f(\mathbf{x}_i) - f^\sharp(\mathbf{x}_i)| &= \sum_{j=1}^{n} s_j(f(\mathbf{x}_i) - f^\sharp(\mathbf{x}_i)) \\
&= \sqrt{n}\langle \mathbf{s}, \mathbf{U}\boldsymbol{\beta} \rangle \overset{(i)}{\leq} \sqrt{n}\sqrt{\sum_{j=1}^{n} \frac{\beta_j^2}{\nu_j}} \sqrt{\sum_{j=1}^{n} \nu_j s_j^2} \leq \sqrt{n}\sqrt{\sum_{j=1}^{n} \frac{\beta_j^2}{\nu_j}} \sqrt{\sum_{j=1}^{n} \nu_j} = \sqrt{2}nR(\delta),
\end{aligned}$$

where (i) follows from the fact that $\mathbf{U}$ is orthogonal and Cauchy-Schwarz inequality. Consequently, for each $i \in [n]$, we have

$$\big|L(y_i, f(\mathbf{x}_i)) - L(y_i, f^\sharp(\mathbf{x}_i))\big| \leq M\big|f(\mathbf{x}_i) - f^\sharp(\mathbf{x}_i)\big| \leq M\sqrt{2}nR(\delta),$$

where the second inequality follows from that $L(y_i, \cdot)$ is $M$-Lipschitz continuous. In addition, for any $f \in \mathcal{F}(\delta)$, we have

$$\begin{aligned}
\frac{1}{n}\sum_{i=1}^{n} \mathbb{E}\Big[\big(L(y_i, f(\mathbf{x}_i)) - L(y_i, f^\sharp(\mathbf{x}_i))\big)^2\Big] &\leq \frac{M^2}{n}\sum_{i=1}^{n}(f(\mathbf{x}_i) - f^\sharp(\mathbf{x}_i))^2 \\
&= M^2\|f - f^\sharp\|_n^2 \\
&= M^2\langle \mathbf{U}\boldsymbol{\beta}, \mathbf{U}\boldsymbol{\beta} \rangle \\
&= M^2\langle \boldsymbol{\beta}, \boldsymbol{\beta} \rangle \\
&\leq M^2 \max_{i \in [n]} \nu_j \sum_{i=1}^{n} \frac{\beta_j^2}{\nu_j} \leq 2M^2 \sum_{i=1}^{n} \nu_j = 2M^2 nR^2(\delta).
\end{aligned}$$

By a similar rescaled method, we have

$$\begin{aligned}
\frac{\big|L(y_i, f(\mathbf{x}_i)) - L(y_i, f^\sharp(\mathbf{x}_i))\big|}{W(f, \delta)} &\leq \frac{M|f(\mathbf{x}_i) - f^\sharp(\mathbf{x}_i)|}{W(f, \delta)} \\
&= M|\widetilde{f}(\mathbf{x}_i) - f^\sharp(\mathbf{x}_i)| \\
&\leq \sqrt{2}MnR(\delta),
\end{aligned}$$

and

$$\begin{aligned}
\frac{1}{n}\sum_{i=1}^{n} \mathbb{E}\Big[\Big(\frac{L(y_i, f(\mathbf{x}_i)) - L(y_i, f^\sharp(\mathbf{x}_i))}{W(f, \delta)}\Big)^2\Big] &\leq \frac{M^2}{n}\sum_{i=1}^{n}\Big(\frac{f(\mathbf{x}_i) - f^\sharp(\mathbf{x}_i)}{W(f, \delta)}\Big)^2 \\
&= \frac{M_{L,\tilde{b}}^2}{n}\sum_{i=1}^{n}\big(\widetilde{f}(\mathbf{x}_i) - f^\sharp(\mathbf{x}_i)\big)^2 \\
&\leq 2M^2 nR^2(\delta).
\end{aligned}$$

Thus, the requirements in Lemma G.1 hold with $\eta = \sqrt{2}MnR(\delta)$ and $\zeta^2 = 2M^2nR^2(\delta)$. Then, by applying Lemma G.1, for any $\iota \in (0,1)$, let $t = \sqrt{\frac{\log \iota^{-1}}{n}}$, it holds at with probability at least $1 - \iota$ that

$$
\begin{aligned}
\mathcal{A} - \mathbb{E}(\mathcal{A}) &\leq \sqrt{\frac{\log \iota^{-1}}{n} \left( 4M^2nR^2(\delta) + 4\sqrt{2}MnR(\delta)\mathbb{E}(\mathcal{A}) \right)} + \frac{2\sqrt{2}M}{3} R(\delta) \log \iota^{-1} \\
&\overset{(i)}{\leq} \widetilde{C} \log \iota^{-1} R(\delta),
\end{aligned}
\tag{13}
$$

where $\widetilde{C} = \sqrt{(4M^2 + 16\sqrt{2\pi}M^2)} + \frac{2\sqrt{2}M}{3}$ and $(i)$ follows from (12).

Therefore, by combining (12) and (13), it holds with probability at least $1 - \iota$ that

$$
\mathcal{A} \leq C \log \iota^{-1} R(\delta)
$$

where $C = \sqrt{(4M^2 + 8\sqrt{2\pi}M^2)} + \frac{2\sqrt{2}M}{3} + 4\sqrt{\pi}M$. This completes the proof of Theorem 3.3. $\square$

### C.2.2  Proof of Lemma C.2

For short, we write

$$
\Delta = \frac{c_0}{4} \delta_n^{4\eta} + 4u^2 \lambda^{2\eta}.
$$

On the event $\mathcal{V}(\iota, \delta_n)$, we claim that

$$
\left\| \widehat{f}_\lambda - f^\sharp \right\|_n^2 \leq C\Delta.
$$

According to the optimality of $\widehat{f}_\lambda$ and the feasibility of $f^\sharp$, we obtain

$$
\widehat{\mathcal{E}}(\widehat{f}_\lambda) - \widehat{\mathcal{E}}(f^\sharp) + \lambda \|\widehat{f}_\lambda\|_K^2 - \lambda \|f^\sharp\|_K^2 \leq 0.
\tag{14}
$$

Then, proving $\|\widehat{f}_\lambda - f^\sharp\|_n^2 \leq C\Delta$ suffices to prove that if $\|\widehat{f}_\lambda - f^\sharp\|_n^2 > C\Delta$ or $\|\widehat{f}_\lambda - f^\sharp\|_K > b$, we have

$$
\widehat{\mathcal{E}}(\widehat{f}_\lambda) - \widehat{\mathcal{E}}(f^\sharp) + \lambda \|\widehat{f}_\lambda\|_K^2 - \lambda \|f^\sharp\|_K^2 > 0.
$$

Below we are devoted to verifying this fact. Define the function class

$$
\mathcal{G} := \left\{ f \in \mathcal{H}_n : \left\| f - f^\sharp \right\|_n^2 \leq C\Delta, \ \left\| f - f^\sharp \right\|_K \leq b \right\}.
$$

Suppose that $\widehat{f}_\lambda \notin \mathcal{G}$. Since both $\mathcal{G}$ and $\mathcal{H}_n$ are convex class by the convexity of $L(y, \cdot)$ and Jensen's inequality, there exists a function $\widetilde{f} = \alpha \widehat{f}_\lambda + (1 - \alpha)f^\sharp$ with $\alpha \in (0, 1]$ that sits on the boundary of $\mathcal{G}$ (Ma et al., 2023). If we can show that

$$
\widehat{\mathcal{E}}(\widetilde{f}) - \widehat{\mathcal{E}}(f^\sharp) + \lambda \|\widetilde{f}\|_K^2 - \lambda \|f^\sharp\|_K^2 > 0,
\tag{15}
$$

by the convexity of $L(y, \cdot)$ and Jensen's inequality, we must have

$$
\widehat{\mathcal{E}}(\widehat{f}_\lambda) - \widehat{\mathcal{E}}(f^\sharp) + \lambda \|\widehat{f}_\lambda\|_K^2 - \lambda \|f^\sharp\|_K^2 \geq \frac{1}{\alpha} \left( \widehat{\mathcal{E}}(\widetilde{f}) - \widehat{\mathcal{E}}(f^\sharp) + \lambda \|\widetilde{f}\|_K^2 - \lambda \|f^\sharp\|_K^2 \right) > 0.
$$

Then, let us focus on proving (15) on the event $\mathcal{V}(\iota, \delta_n)$.

Note that $\widetilde{f}$ belongs to the the boundary of $\mathcal{G}$, we can split the remaining proof into two cases: (i) $\|\widetilde{f} - f^\sharp\|_n^2 = C\Delta$ and $\|\widetilde{f} - f^\sharp\|_K \leq b$; and (ii) $\|\widetilde{f} - f^\sharp\|_n^2 \leq C\Delta$ and $\|\widetilde{f} - f^\sharp\|_K = b$.

*Case (i):* By applying (7) and (8), from Assumption 3.1, we have

$$
c_0 \left\| \widetilde{f} - f^\sharp \right\|_n^2 = c_0 \left\| \widetilde{f} - f^* \right\|_n^2 \leq \mathcal{E}(\widetilde{f}) - \mathcal{E}(f^*) = \mathcal{E}(\widetilde{f}) - \mathcal{E}(f^\sharp).
$$

so that the $c_0$-strong convexity also holds for $f^\sharp$.

On the event $\mathcal{V}(\iota, \delta_n)$, we have

$$
\begin{aligned}
&\widehat{\mathcal{E}}(f^\sharp) - \widehat{\mathcal{E}}(\widetilde{f}) \\
&\leq \mathcal{E}(f^\sharp) - \mathcal{E}(\widetilde{f}) + C \log \iota^{-1} R(\delta_n) W(\widetilde{f}, \delta_n) \\
&\leq -c_0 \big\| \widetilde{f} - f^\sharp \big\|_n^2 + C \log \iota^{-1} R(\delta_n) \left( \delta_n^{-1} \big\| \widetilde{f} - f^\sharp \big\|_n + \big\| \widetilde{f} - f^\sharp \big\|_K \right) \\
&\overset{(i)}{=} -c_0 C \Delta + C \log \iota^{-1} R(\delta_n) \left( \delta_n^{-1} \sqrt{C\Delta} + \big\| \widetilde{f} - f^\sharp \big\|_K \right) \\
&\overset{(ii)}{\leq} -c_0 C \Delta + C \delta_n^{-1} \log \iota^{-1} R(\delta_n) \sqrt{C\Delta} + C(\log \iota^{-1} R(\delta_n))^2 \lambda^{-1} + \frac{\lambda}{4} \big\| \widetilde{f} - f^\sharp \big\|_K^2 \\
&\overset{(iii)}{\leq} -c_0 C \Delta + \frac{c_0}{2} \delta_n^{2\eta} \sqrt{C\Delta} + \frac{c_0}{4} \delta_n^{4\eta+2} \lambda^{-1} + \frac{\lambda}{4} \big\| \widetilde{f} - f^\sharp \big\|_K^2,
\end{aligned}
$$

where (i) follows from $\| \widetilde{f} - f^\sharp \|_n^2 = C\Delta$, (ii) uses the elementary inequality that $2ab \leq a^2 + b^2$ and (iii) follows from the definition of $\delta_n$ satisfying

$$
C \log \iota^{-1} R(\delta_n) \leq \frac{c_0}{2} \delta_n^{2\eta+1}.
$$

Further with the choice of $\lambda \geq \delta_n^2$, we have

$$
\widehat{\mathcal{E}}(f^\sharp) - \widehat{\mathcal{E}}(\widetilde{f}) \leq -c_0 C \Delta + \frac{c_0}{2} \delta_n^{2\eta} \sqrt{C\Delta} + \frac{c_0}{4} \delta_n^{4\eta} + \frac{\lambda}{4} \big\| \widetilde{f} - f^\sharp \big\|_K^2.
$$

On the other hand, we notice that

$$
\lambda \big\| f^\sharp \big\|_K^2 - \lambda \big\| \widetilde{f} \big\|_K^2 = -2\lambda \langle f^\sharp, \widetilde{f} - f^\sharp \rangle_K - \lambda \big\| \widetilde{f} - f^\sharp \big\|_K^2. \tag{16}
$$

Note that $\langle f, g \rangle_K = \langle \mathbf{K}^{1/2} f, \mathbf{K}^{1/2} g \rangle_K$ for any $f, g \in \mathcal{H}_n$. $\widetilde{f} \in \mathcal{H}_n$ so that it can be written as

$$
\widetilde{f} = S_{\mathbf{x}}^\top(\widetilde{\boldsymbol{\alpha}}) \quad \text{for some} \quad \widetilde{\boldsymbol{\alpha}} \in \mathcal{R}^n.
$$

Therefore, for $\frac{1}{2} \leq \gamma \leq 1$, there holds

$$
\begin{aligned}
\left| \lambda \langle f^\sharp, \widetilde{f} - f^\sharp \rangle_K \right| &= \left| \lambda \langle \mathbf{K}^{1/2} \boldsymbol{\alpha}^\sharp, \mathbf{K}^{1/2}(\widetilde{\boldsymbol{\alpha}} - \boldsymbol{\alpha}^\sharp) \rangle_2 \right| \\
&= \left| \lambda \langle \mathbf{K}^{1-\gamma} \boldsymbol{\alpha}^\sharp, \mathbf{K}^\gamma(\widetilde{\boldsymbol{\alpha}} - \boldsymbol{\alpha}^\sharp) \rangle_2 \right| \\
&\leq \lambda \left\| \mathbf{K}^{1-\gamma} \boldsymbol{\alpha}^\sharp \right\|_2 \left\| \mathbf{K}^\gamma(\widetilde{\boldsymbol{\alpha}} - \boldsymbol{\alpha}^\sharp) \right\|_2 \\
&= \lambda \left\| \mathbf{D}^{1-\gamma} \mathbf{D}^{-1} \xi^* \right\|_2 \left\| \mathbf{K}^\gamma(\widetilde{\boldsymbol{\alpha}} - \boldsymbol{\alpha}^\sharp) \right\|_2 \\
&= \lambda \left\| \mathbf{D}^{-\gamma} \xi^* \right\|_2 \left\| \mathbf{K}^\gamma(\widetilde{\boldsymbol{\alpha}} - \boldsymbol{\alpha}^\sharp) \right\|_2 \\
&\overset{(i)}{\leq} u\lambda \left\| \mathbf{K}^\gamma(\widetilde{\boldsymbol{\alpha}} - \boldsymbol{\alpha}^\sharp) \right\|_2,
\end{aligned}
$$

where (i) follows from Assumption 3.2.

For $\frac{1}{2} \leq \gamma \leq 1$, using a similar treatment as that in Lian (2022), we apply Young's inequality $AB \leq \frac{A^p}{p} + \frac{B^q}{q}$ for any two positive operators $A$ and $B$ with $\frac{1}{p} + \frac{1}{q} = 1$ and $p, q \geq 1$ to obtain

$$
\begin{aligned}
&\lambda \left\| \mathbf{K}^\gamma(\widetilde{\boldsymbol{\alpha}} - \boldsymbol{\alpha}^\sharp) \right\|_2 \\
&= \lambda^\gamma \sqrt{\langle \lambda^{2-2\gamma} \mathbf{K}^{2\gamma-1} \mathbf{K}(\widetilde{\boldsymbol{\alpha}} - \boldsymbol{\alpha}^\sharp), \widetilde{\boldsymbol{\alpha}} - \boldsymbol{\alpha}^\sharp \rangle_2} \\
&\leq \lambda^\gamma \sqrt{\langle ((2-2\gamma)\lambda + (2\gamma-1)\mathbf{K}) \mathbf{K}(\widetilde{\boldsymbol{\alpha}} - \boldsymbol{\alpha}^\sharp), \widetilde{\boldsymbol{\alpha}} - \boldsymbol{\alpha}^\sharp \rangle_2} \\
&\overset{(i)}{\leq} \lambda^\gamma \max \left\{ \sqrt{\langle \lambda \mathbf{K}^{1/2}(\widetilde{\boldsymbol{\alpha}} - \boldsymbol{\alpha}^\sharp), \mathbf{K}^{1/2}(\widetilde{\boldsymbol{\alpha}} - \boldsymbol{\alpha}^\sharp) \rangle_2}, \sqrt{\langle \mathbf{K}(\widetilde{\boldsymbol{\alpha}} - \boldsymbol{\alpha}^\sharp), \mathbf{K}(\widetilde{\boldsymbol{\alpha}} - \boldsymbol{\alpha}^\sharp) \rangle_2} \right\} \\
&\overset{(ii)}{\leq} \lambda^{\gamma+\frac{1}{2}} \big\| \widetilde{f} - f^\sharp \big\|_K + \lambda^\gamma \big\| \widetilde{f} - f^\sharp \big\|_n, \tag{17}
\end{aligned}
$$

where (i) holds by taking $\gamma = \frac{1}{2}$ and $\gamma = 1$, and (ii) follows from $\max\{a, b\} \le a + b$.

For $\gamma > 1$, we claim that $\|\mathbf{D}^{-1}\xi^*\|_2 \le u$. To see this, we find that

$$\left\|\mathbf{D}^{-1}\xi^*\right\|_2^2 = \sum_{j=1}^{\infty} \mu_j^{-2}\xi_j^{*2} \overset{(i)}{\le} \sum_{j=1}^{\infty} \mu_j^{-2\gamma}\xi_j^{*2} \le u^2,$$

where (i) holds by our assumption $\mu_j \le 1$ for all $j$.

Then, similar to (17) with $\gamma = 1$, we have

$$\left|\lambda\langle f^\sharp, \widetilde{f} - f^\sharp\rangle_K\right| \le u\lambda^{\frac{3}{2}}\left\|\widetilde{f} - f^\sharp\right\|_K + u\lambda\left\|\widetilde{f} - f^\sharp\right\|_n.$$

Combine these two case to obtain that for $\gamma \ge \frac{1}{2}$

$$\begin{aligned}
\left|\lambda\langle f^\sharp, \widetilde{f} - f^\sharp\rangle_K\right| &\le u\lambda^{\eta+\frac{1}{2}}\left\|\widetilde{f} - f^\sharp\right\|_K + u\lambda^\eta\left\|\widetilde{f} - f^\sharp\right\|_n \\
&= u\lambda^{\eta+\frac{1}{2}}\left\|\widetilde{f} - f^\sharp\right\|_K + u\lambda^\eta\sqrt{C\Delta}.
\end{aligned} \tag{18}$$

Here, we recall that $\eta = \min\{\gamma, 1\}$.

Putting the pieces together, for all $\gamma \ge \frac{1}{2}$, we have

$$\begin{aligned}
&\widehat{\mathcal{E}}(f^\sharp) - \widehat{\mathcal{E}}(\widetilde{f}) + \lambda\|f^\sharp\|_K^2 - \lambda\|\widetilde{f}\|_K^2 \\
&\le -c_0 C\Delta + \left(\frac{c_0}{2}\delta_n^{2\eta} + 2u\lambda^\eta\right)\sqrt{C\Delta} + \frac{c_0}{4}\delta_n^{4\eta} + \frac{\lambda}{4}\left\|\widetilde{f} - f^\sharp\right\|_K^2 + 2u\lambda^{\eta+\frac{1}{2}}\left\|\widetilde{f} - f^\sharp\right\|_K - \lambda\left\|\widetilde{f} - f^\sharp\right\|_K^2 \\
&\overset{(i)}{\le} -c_0 C\Delta + \left(\frac{c_0}{2}\delta_n^{2\eta} + 2u\lambda^\eta\right)\sqrt{C\Delta} + \frac{c_0}{4}\delta_n^{4\eta} + \frac{\lambda}{2}\left\|\widetilde{f} - f^\sharp\right\|_K^2 + 4u^2\lambda^{2\eta} - \lambda\left\|\widetilde{f} - f^\sharp\right\|_K^2 \\
&\le -c_0 C\Delta + \left(\frac{c_0}{2}\delta_n^{2\eta} + 2u\lambda^\eta\right)\sqrt{C\Delta} + \frac{c_0}{4}\delta_n^{4\eta} + 4u^2\lambda^{2\eta},
\end{aligned} \tag{19}$$

where (i) uses the elementary inequality.

Below is devoted to proving that for a sufficiently large $C$, the RHS of (19) is less than $0$. Precisely, let

$$\varphi(x) = c_0 x^2 - \left(\frac{c_0}{2}\delta_n^{2\eta} + 2u\lambda^\eta\right)x - \frac{c_0}{4}\delta_n^{4\eta} - 4u^2\lambda^{2\eta}.$$

Let $x = \sqrt{C\Delta}$, note that

$$\begin{aligned}
\varphi(x) &= c_0 C\Delta - \left(\frac{c_0}{2}\delta_n^{2\eta} + 2u\lambda^\eta\right)\sqrt{C\Delta} - \frac{c_0}{4}\delta_n^{4\eta} - 4u^2\lambda^{2\eta} \\
&= (c_0 C - 1)\Delta - \left(\frac{c_0}{2}\delta_n^{2\eta} + 2u\lambda^\eta\right)\sqrt{C\Delta} \\
&= \sqrt{C\Delta}\left[\frac{c_0 C - 1}{\sqrt{C}}\sqrt{\Delta} - \left(\frac{c_0}{2}\delta_n^{2\eta} + 2u\lambda^\eta\right)\right] \\
&\overset{(i)}{\ge} \sqrt{C\Delta}\left[\frac{c_0 C - 1}{\sqrt{2C}}\left(\frac{\sqrt{c_0}}{2}\delta_n^{2\eta} + 2u\lambda^\eta\right) - \left(\frac{c_0}{2}\delta_n^{2\eta} + 2u\lambda^\eta\right)\right],
\end{aligned}$$

where (i) follows from the basic inequality $\frac{a+b}{2} \le \sqrt{\frac{a^2+b^2}{2}}$. Since $\frac{c_0 C - 1}{\sqrt{2C}}$ is increasing in $C$, we can select $C$ such that $\frac{c_0 C - 1}{\sqrt{2C}} \ge \max\{\sqrt{c_0}, 1\}$, which leads to $\varphi(x) > 0$.

In conclusion, for a sufficiently large $C$ in the definition of the function class $\mathcal{G}$, on the event $\mathcal{V}(\iota, \delta_n)$, for case (i), we have

$$\widehat{\mathcal{E}}(f^\sharp) - \widehat{\mathcal{E}}(\widetilde{f}) + \lambda\|f^\sharp\|_K^2 - \lambda\|\widetilde{f}\|_K^2 < 0.$$

*Case (ii)*: Repeat the similar argument as that in Case (i), on the event $\mathcal{V}(\iota, \delta_n)$ and by Assumption 3.1, we have

$$
\begin{aligned}
&\widehat{\mathcal{E}}(f^\sharp) - \widehat{\mathcal{E}}(\widetilde{f}) \\
&\leq -c_0 \big\| \widetilde{f} - f^\sharp \big\|_n^2 + C \log \iota^{-1} R(\delta_n) \left( \delta_n^{-1} \big\| \widetilde{f} - f^\sharp \big\|_n + \big\| \widetilde{f} - f^\sharp \big\|_K \right) \\
&\leq C \log \iota^{-1} R(\delta_n) \left( \delta_n^{-1} \big\| \widetilde{f} - f^\sharp \big\|_n + \big\| \widetilde{f} - f^\sharp \big\|_K \right) \\
&\leq \frac{c_0}{2} \delta_n^{2\eta} \sqrt{C\Delta} + \frac{c_0}{4} \delta_n^{4\eta+2} \lambda^{-1} + \frac{\lambda}{4} \big\| \widetilde{f} - f^\sharp \big\|_K^2 .
\end{aligned}
$$

Further with the choice of $\lambda \geq \delta_n^2$, we have

$$
\widehat{\mathcal{E}}(f^\sharp) - \widehat{\mathcal{E}}(\widetilde{f}) \leq \frac{c_0}{2} \delta_n^{2\eta} \sqrt{C\Delta} + \frac{c_0}{4} \delta_n^{4\eta} + \frac{\lambda}{4} \big\| \widetilde{f} - f^\sharp \big\|_K^2 .
$$

Combine with (16) and (18) to obtain

$$
\begin{aligned}
&\widehat{\mathcal{E}}(f^\sharp) - \widehat{\mathcal{E}}(\widetilde{f}) + \lambda \|f^\sharp\|_K^2 - \lambda \|\widetilde{f}\|_K^2 \\
&\leq \left( \frac{c_0}{2} \delta_n^{2\eta} + 2u\lambda^\eta \right) \sqrt{C\Delta} + \frac{c_0}{4} \delta_n^{4\eta} + \frac{\lambda}{4} \big\| \widetilde{f} - f^\sharp \big\|_K^2 + 2u\lambda^{\eta+\frac{1}{2}} \big\| \widetilde{f} - f^\sharp \big\|_K - \lambda \big\| \widetilde{f} - f^\sharp \big\|_K^2 \\
&\leq \left( \frac{c_0}{2} \delta_n^{2\eta} + 2u\lambda^\eta \right) \sqrt{C\Delta} + \frac{c_0}{4} \delta_n^{4\eta} + 4u^2\lambda^{2\eta} - \frac{\lambda}{2} \big\| \widetilde{f} - f^\sharp \big\|_K^2 .
\end{aligned}
$$

Note that $0 < \lambda \leq 1$ and $\lambda \geq \delta_n^2$ together implies

$$
\lambda \geq \lambda^{2\eta} \quad \text{and} \quad \lambda \geq \delta_n^{4\eta} .
$$

Moreover, in Case (ii), $\|\widetilde{f} - f^\sharp\|_K = b$. Therefore,

$$
\begin{aligned}
&\widehat{\mathcal{E}}(f^\sharp) - \widehat{\mathcal{E}}(\widetilde{f}) + \lambda \|f^\sharp\|_K^2 - \lambda \|\widetilde{f}\|_K^2 \\
&\leq \left( \frac{c_0}{2} \delta_n^{2\eta} + 2u\lambda^\eta \right) \sqrt{C\Delta} + \frac{c_0}{4} \delta_n^{4\eta} + 4u^2\lambda^{2\eta} - \frac{1}{2} b^2 \delta_n^{4\eta} - \frac{1}{2} b^2 \lambda^{2\eta} \\
&\leq \sqrt{\frac{c_0^2}{2} \delta_n^{4\eta} + 8u^2\lambda^{2\eta}} \sqrt{C\Delta} + \frac{c_0}{4} \delta_n^{4\eta} + 4u^2\lambda^{2\eta} - \frac{1}{2} b^2 \delta_n^{4\eta} - \frac{1}{2} b^2 \lambda^{2\eta} \\
&\leq \sqrt{C} \left( \frac{c_0}{4} \max\{2c_0, 1\} \delta_n^{4\eta} + 8u^2\lambda^{2\eta} \right) + \frac{c_0}{4} \delta_n^{4\eta} + 4u^2\lambda^{2\eta} - \frac{1}{2} b^2 \delta_n^{4\eta} - \frac{1}{2} b^2 \lambda^{2\eta} ,
\end{aligned}
$$

where the last line is less line is less than 0 for sufficiently large constant $b$.

At last, by combining Cases (i) and (ii), we prove (15). Therefore, on the event $\mathcal{V}(\iota, \delta_n)$, we have

$$
\big\| \widehat{f}_\lambda - f^\sharp \big\|_n^2 \leq C\Delta \lesssim \delta_n^{4\eta} + \lambda^{2\eta} ,
$$

which completes the proof. $\qquad\square$

## C.3 Proof of Corollary 3.5

Recall that the statistical dimension is defined as

$$
d(\delta) = \min \left\{ j \in [n] : \mu_j \leq \delta^2 \right\} .
$$

From the definition of $d(\delta)$, we have

$$
R(\delta) = \sqrt{\frac{1}{n} d(\delta)\delta^2 + \frac{1}{n} \sum_{d(\delta)+1}^{n} \mu_j} . \tag{20}
$$

Recalling that $\sum_{d(\delta)+1}^{n} \mu_j \lesssim d(\delta)\delta^2$ for regular kernel and kernel with the polynomial in its eigenvalues is regular (Yang et al., 2017), the kernel complexity function satisfies

$$
R(\delta) \asymp \sqrt{\frac{1}{n} d(\delta)\delta^2} .
$$

Therefore, the solution to the inequality (2) can be bounded from above by the solution to

$$C \log \iota^{-1} \sqrt{\frac{1}{n} d(\delta)\delta^2} \leq \frac{c_0}{2} \delta^{2\eta+1}. \tag{21}$$

Moreover, if the eigenvalues of $\mathbf{K}$ exhibit $\alpha$-polynomial decay that is $\mu_j \asymp j^{-\alpha}$, then we have $d(\delta) \asymp \delta^{-2/\alpha}$. Together with (21) leads to

$$\delta_n^2 \leq C\Big(\frac{(\log \iota^{-1})^2}{n}\Big)^{\frac{\alpha}{2\eta\alpha+1}}.$$

Then, with the choice of $\lambda \asymp \delta_n^2$, it holds with probability at least $1 - \iota$ that

$$\|\widehat{f}_\lambda - f^*\|_n^2 \leq C(\delta_n^{4\eta} + \lambda^{2\eta}) \leq C\Big(\frac{(\log \iota^{-1})^2}{n}\Big)^{\frac{2\eta\alpha}{2\eta\alpha+1}}.$$

We conclude the upper bound in Corollary 3.5. $\qquad\square$

# D  Proof of Results for Truncated Kernel-based Method

## D.1  Error Analysis

Recall that

$$\mathbf{q}^* = \sqrt{n} S_{\mathbf{x}}(f^*) = (f^*(\mathbf{x}_1), ..., f^*(\mathbf{x}_n))^\top$$

and

$$\xi^* = \mathbf{U}^\top S_{\mathbf{x}}(f^*) = \frac{1}{\sqrt{n}} \mathbf{U}^\top \mathbf{q}^*.$$

In the proof for the truncated kernel-based estimator, we partition $\xi^*$ into two sub-vectors as

$$\xi^{*\top} = (\xi_1^{*\top}, \xi_2^{*\top})$$

with $\xi_1^* \in \mathcal{R}^r$ and $\xi_2^* \in \mathcal{R}^{n-r}$.

Moreover, we partition $\mathbf{U}$ into two sub-matrixs

$$\mathbf{U} = (\mathbf{U}_1, \mathbf{U}_2)$$

with $\mathbf{U}_1 \in \mathcal{R}^{n\times r}$ and $\mathbf{U}_2 \in \mathcal{R}^{n\times(n-r)}$, and partition $\mathbf{D}$ into two blocks $\mathbf{D}_1$ and $\mathbf{D}_2$, that is

$$\mathbf{D} = \begin{pmatrix} \mathbf{D}_1 & \\ & \mathbf{D}_2 \end{pmatrix},$$

where $\mathbf{D}_1 \in \mathcal{R}^{r\times r}$ and $\mathbf{D}_2 \in \mathcal{R}^{(n-r)\times(n-r)}$. Since the last $n - r$ diagonal elements of $\mathbf{D}_r$ are all zero, for any $\boldsymbol{\alpha} \in \mathcal{R}^n$, the last $n - r$ elements of $\mathbf{D}_r \mathbf{U}^\top \boldsymbol{\alpha}$ are also all zero. Then, we have

$$\mathbf{D}_r \mathbf{U}^\top \boldsymbol{\alpha} = \big((\mathbf{D}_1 \mathbf{U}_1^\top \boldsymbol{\alpha})^\top, \mathbf{0}^\top\big)^\top \quad \text{for all} \quad \boldsymbol{\alpha} \in \mathcal{R}^n. \tag{22}$$

Define

$$\mathcal{H}_{n,r} := \big\{ f = S_{\mathbf{x},r}^\top(\boldsymbol{\alpha}) : \boldsymbol{\alpha} \in \mathcal{R}^n \big\}.$$

For any $f = S_{\mathbf{x},r}^\top(\boldsymbol{\alpha}) \in \mathcal{H}_{n,r}$, we have

$$\|f\|_n = \big\|\mathbf{K}_r \boldsymbol{\alpha}\big\|_2 = \big\|\mathbf{D}_r \mathbf{U}^\top \boldsymbol{\alpha}\big\|_2 = \big\|\mathbf{D}_1 \mathbf{U}_1^\top \boldsymbol{\alpha}\big\|_2,$$

where the last step holds by applying (22). We also observe

$$\|f\|_{K_r} = \big\|\mathbf{K}_r^{1/2} \boldsymbol{\alpha}\big\|_2 = \big\|\mathbf{D}_1^{1/2} \mathbf{U}_1^\top \boldsymbol{\alpha}\big\|_2. \tag{23}$$

Define an immediate function

$$f_r^\sharp := S_{\mathbf{x},r}^\top(\boldsymbol{\alpha}_r^\sharp) \in \mathcal{H}_{n,r} \quad \text{with} \quad \boldsymbol{\alpha}_r^\sharp = \mathbf{U}_1 \mathbf{D}_1^{-1} \xi_1^* \in \mathcal{R}^n. \tag{24}$$

From (23), we have

$$\left\|f_r^\sharp\right\|_{K_r}^2 = \left\|\mathbf{D}_1^{1/2}\mathbf{U}_1^\top\mathbf{U}_1\mathbf{D}_1^{-1}\xi_1^*\right\|_{K_r}^2$$
$$= \left\|\mathbf{D}_1^{-1/2}\xi_1^*\right\|_{K_r}^2 = \sum_{j=1}^r \mu_j^{-1}\xi_j^{*2} \overset{(i)}{\leq} \sum_{j=1}^r \mu_j^{-2\gamma}\xi_j^{*2} \overset{(ii)}{\leq} u^2, \tag{25}$$

where (i) holds by our assumption $\mu_j \leq 1$ for all $j$ and (ii) follows from Assumption 3.2.

The construction of $f_r^\sharp$ allows us to analyze the prediction error of the truncated kernel-based estimator from two sources: the estimation error depending on the complexity of the truncated RKHS $\mathcal{H}_{K_r}$, and the approximation error arising from the dissimilarity between the truncated RKHS $\mathcal{H}_{K_r}$ and the full RKHS $\mathcal{H}_K$. Specifically, we have the error decomposition as follows.

**Error decomposition.** By applying the elementary inequality that $(a+b)^2 \leq 2a^2 + 2b^2$, the total error $\|\widehat{f}_{\lambda,r} - f^*\|_n^2$ can be decomposed as

$$\left\|\widehat{f}_{\lambda,r} - f^*\right\|_n^2 \leq 2 \underbrace{\left\|\widehat{f}_{\lambda,r} - f_r^\sharp\right\|_n^2}_{\text{Estimation error}} + 2 \underbrace{\left\|f_r^\sharp - f^*\right\|_n^2}_{\text{Approximation bias}}. \tag{26}$$

Note that both $\widehat{f}_{\lambda,r}$ and $f_r^\sharp$ belong to $\mathcal{H}_{n,r}$, allowing us to analyze the estimation error based on the complexity of the reduced kernel matrix $\mathbf{K}_r$. This decomposition successfully captures two components of error: estimation error and approximation bias. The estimation error is controlled by the model richness of the truncated space $\mathcal{H}_{K_r}$, while approximation bias depends on the dissimilarity between the truncated RKHS $\mathcal{H}_{K_r}$ and the full RKHS $\mathcal{H}_K$ where the true target $f^*$ is sitting in. A larger $r$ amplifies the space $\mathcal{H}_{K_r}$, resulting in a larger estimation error. At the same time, it narrows the gap between $\mathcal{H}_{K_r}$ and $\mathcal{H}_K$, thereby decreasing the approximation bias. Consequently, a trade-off emerges, and an optimal choice of truncation $r$ aims to balance the estimation error and approximation bias.

## D.2  Proof of Theorem 4.2

We will separately bound each term from above appearing in the decomposition (26).

**Bounding the approximation bias.** Note that

$$\left\|f_r^\sharp - f^*\right\|_n^2 = \frac{1}{n}\left\|\mathbf{q}^* - \sqrt{n}\mathbf{K}_r\boldsymbol{\alpha}_r^\sharp\right\|_n^2$$
$$\overset{(i)}{=} \left\|\xi^* - \mathbf{D}_r\mathbf{U}^\top\boldsymbol{\alpha}_r^\sharp\right\|_2^2 \overset{(ii)}{=} \left\|\xi_1^* - \mathbf{D}_1\mathbf{U}_1^\top\boldsymbol{\alpha}_r^\sharp\right\|_2^2 + \left\|\xi_2^*\right\|_2^2,$$

where (i) follows from the eigen-expansion that $\mathbf{K}_r = \mathbf{U}\mathbf{K}_r\mathbf{U}^\top$ and (ii) follows from (22).

By the definition of $\boldsymbol{\alpha}_r^\sharp$, we have

$$\mathbf{D}_1\mathbf{U}_1^\top\boldsymbol{\alpha}_r^\sharp = \mathbf{D}_1\mathbf{U}_1^\top\mathbf{U}_1\mathbf{D}_1^{-1}\xi_1^* = \xi_1^*.$$

Therefore, we arrive at

$$\left\|f_r^\sharp - f^*\right\|_n^2 = \left\|\xi_2^*\right\|_2^2 = \sum_{j=r+1}^n \xi_j^{*2}. \tag{27}$$

**Bounding the estimation error.** Define the localized function class

$$\mathcal{H}_{n,r,b} := \left\{f : f \in \mathcal{H}_{n,r}, \ \|f - f_r^\sharp\|_K \leq b\right\},$$

where $b > 1$ is a constant independent of $n, \gamma$, which will be specified in the proof.

Recall $r$ from Section 4. For any given $\iota \in (0,1)$ and $\delta > 0$, define the auxiliary event

$$\mathcal{V}_r(\iota, \delta) := \left\{\left|\widehat{\mathcal{E}}(f) - \widehat{\mathcal{E}}(f_r^\sharp) - [\mathcal{E}(f) - \mathcal{E}(f_r^\sharp)]\right| \leq C \log \iota^{-1} R_r(\delta) W_r(f, \delta) \quad \text{holds for any } f \in \mathcal{H}_{n,r,b}\right\},$$

where $W_r(f, \delta) := \delta^{-1}\|f - f_r^\sharp\|_n + \|f - f_r^\sharp\|_{K_r}$ for $\delta > 0$ and $f \in \mathcal{H}_{n,r,b}$.

To establish the bound for the estimation error, we need the following two lemmas.

**Lemma D.1.** *Fix any $\iota \in (0, 1)$ and $\delta > 0$. The event $\mathcal{V}_r(\iota, \delta)$ occurs with probability greater than $1 - \iota$, i.e.*

$$\mathbb{P}\big(\mathcal{V}_r(\iota, \delta)\big) \geq 1 - \iota.$$

Recall that $\delta_{n,r}$ is the critical radius w.r.t. the truncated kernel complexity function defined as the smallest solution to (3).

**Lemma D.2.** *Let $\eta = \min\{\gamma, 1\}$. On the event $\mathcal{V}_r(\iota, \delta_{n,r})$, with the choice of $\lambda$ satisfying $\max\{\delta_{n,r}^2, \sum_{j=r+1}^n \xi_j^{*2}\} \leq \lambda \leq 1$, we have*

$$\big\|\widehat{f}_{\lambda,r} - f_r^\sharp\big\|_n^2 \leq C\Big(\delta_{n,r}^{4\eta} + \lambda^{2\eta} + \sum_{j=r+1}^n \xi_j^{*2}\Big),$$

*where $C$ is a constant independent of $n, \gamma$.*

*Proof of Theorem 4.2.* By applying Lemma D.1, we have

$$\mathbb{P}\big(\mathcal{V}_r(\iota, \delta_{n,r})\big) \geq 1 - \iota,$$

which, together with Lemma D.2, implies

$$\big\|\widehat{f}_{\lambda,r} - f_r^\sharp\big\|_n^2 \leq C\Big(\delta_{n,r}^{4\eta} + \lambda^{2\eta} + \sum_{j=r+1}^n \xi_j^{*2}\Big).$$

holds with probability at least $1 - \iota$.

Finally, by applying the error decomposition (26) and the equality (27), one has

$$\big\|\widehat{f}_{\lambda,r} - f^*\big\|_n^2 \leq 2\big\|\widehat{f}_{\lambda,r} - f_r^\sharp\big\|_n^2 + 2\big\|f_r^\sharp - f^*\big\|_n^2$$

$$\leq C\Big(\delta_{n,r}^{4\eta} + \lambda^{2\eta} + \sum_{j=r+1}^n \xi_j^{*2}\Big),$$

which completes the proof for the $\mathcal{L}(\mathbb{P}_n)$-error. For the excess risk, it immediately follows from Assumption 3.1. □

### D.2.1 Proof of Lemma D.1

Denote

$$\mathcal{D}_r = \widehat{\mathcal{E}}(f) - \widehat{\mathcal{E}}(f_r^\sharp) - \big[\mathcal{E}(f) - \mathcal{E}(f_r^\sharp)\big].$$

Similar to the proof of Lemma C.2, it is equivalent to proving that

$$\mathcal{A}_r := \sup_{f \in \mathcal{H}_{n,r,b}} \frac{|\mathcal{D}_r|}{W_r(f, \delta)} \leq C \log \iota^{-1} R_r(\delta).$$

From (25), observe that for any $f \in \mathcal{H}_{n,r,b}$

$$\|f\|_{K_r} \leq \big\|f - f_r^\sharp\big\|_{K_r} + \big\|f_r^\sharp\big\|_{K_r} \leq u + b.$$

Let $\tilde{b} := u + b$, and we have that $L(y, \cdot)$ satisfies the Lipschitz continuity over the function class $\mathcal{H}_{n,r,b}$ with Lipschitz constant $M_{L,\tilde{b}}$. Write $M := M_{L,\tilde{b}}$ for short.

**Bounding $\mathbb{E}[\mathcal{A}_r]$.** Following a similar treatment as that in (10), by using the Lemma G.2 and Lemma G.3, we have

$$\begin{aligned}
\mathbb{E}[\mathcal{A}_r] &= \mathbb{E}\Big[\sup_{f \in \mathcal{H}_{n,r,b}} \frac{|\mathcal{D}_r|}{W_r(f, \delta)}\Big] \\
&\leq \frac{2\sqrt{2\pi}M}{n}\mathbb{E}\Big[\sup_{f \in \mathcal{H}_{n,r,b}} \frac{\big|\sum_{i=1}^n w_i(f(\mathbf{x}_i) - f_r^\sharp(\mathbf{x}_i))\big|}{W_r(f, \delta)}\Big].
\end{aligned}$$

(28)

Let

$$\mathcal{F}_r(\delta) := \Big\{f = S_{\mathbf{x},r}^\top(\boldsymbol{\alpha}) : \big\|f - f_r^\sharp\big\|_{K_r} \leq 1, \big\|f - f_r^\sharp\big\|_n \leq \delta, \boldsymbol{\alpha} \in \mathcal{R}^n\Big\}.$$

For any $f \in \mathcal{H}_{n,r,b}$, there exists $\boldsymbol{\alpha} \in \mathcal{R}^n$ such that $f = S_{\mathbf{x},r}^\top(\boldsymbol{\alpha})$. Recall that $f_r^\sharp = S_{\mathbf{x},r}^\top(\boldsymbol{\alpha}_r^\sharp)$ from (24).

Define the vector $\boldsymbol{\beta} := \mathbf{D}_r \mathbf{U}^\top(\boldsymbol{\alpha} - \boldsymbol{\alpha}_r^\sharp)$, then $f \in \mathcal{F}(\delta)$ is equivalent to the constraints on $\boldsymbol{\beta}_r$ that

$$\left\|\mathbf{D}_r^{-1/2}\boldsymbol{\beta}\right\|_2 \leq 1 \quad \text{and} \quad \|\boldsymbol{\beta}\|_2 \leq \delta.$$

From (22), the last $n - r$ elements of $\boldsymbol{\beta}$ are all zero, then any vector satisfying these two constraints must belong to the ellipse class

$$\mathcal{E}_r := \left\{ \boldsymbol{\beta} = (\beta_1, \beta_2, \dots)^\top \in \mathcal{R}^n : \sum_{j=1}^r \frac{\beta_j^2}{\nu_j} \leq 2 \text{ with } \nu_j = \min\{\delta^2, \mu_j\}\right\}.$$

Denote $\mathbf{w} = (w_1, ..., w_n)^\top$, we have

$$\mathbb{E}\left[\sup_{f \in \mathcal{F}_r(\delta)} \left| \sum_{i=1}^n w_i(f(\mathbf{x}_i) - f_r^\sharp(\mathbf{x}_i))\right|\right] = \mathbb{E}\left[\sup_{\boldsymbol{\beta} \in \mathcal{E}_r} \sqrt{n}|\langle \mathbf{w}, \mathbf{U}\boldsymbol{\beta}\rangle|\right]$$

$$= \mathbb{E}\left[\sup_{\boldsymbol{\beta} \in \mathcal{E}_r} \sqrt{n}|\langle \mathbf{w}, \boldsymbol{\beta}\rangle|\right]$$

$$\overset{(i)}{\leq} \mathbb{E}\left[\sup_{\boldsymbol{\beta} \in \mathcal{E}_r} \sqrt{n}\sqrt{\sum_{j=1}^r \frac{\beta_j^2}{\nu_j}}\sqrt{\sum_{j=1}^r \nu_j w_j^2}\right]$$

$$\leq \sqrt{2n}\sqrt{\frac{1}{n}\sum_{j=1}^r \nu_j} = \sqrt{2n}R_r(\delta),$$

where (i) follows from Cauchy-Schwarz inequality and the fact that the last $n - r$ elements of $\boldsymbol{\beta}$ are all zero.

Similar to the argument in the proof for Lemma C.2, by appropriately scaling, we obtain

$$\mathbb{E}[\mathcal{A}_r] \leq \frac{2\sqrt{2\pi}M}{n}\mathbb{E}\left[\sup_{f \in \mathcal{F}_r(\delta)} \left| \sum_{i=1}^n w_i(f(\mathbf{x}_i) - f_r^\sharp(\mathbf{x}_i))\right|\right] \leq 4\sqrt{\pi}MR_r(\delta). \quad (29)$$

**Bounding** $\mathcal{A}_r - \mathbb{E}[\mathcal{A}_r]$. For each $i \in [n]$ and any $f \in \mathcal{F}_r(\delta)$, define $s_j = \text{sign}(f(\mathbf{x}_i) - f_r^\sharp(\mathbf{x}_i))$ if $j = i$ and $s_j = 0$ if $j \neq i$ and let $\mathbf{s} = (s_1, ..., s_n)^\top$, then we have

$$|f(\mathbf{x}_i) - f_r^\sharp(\mathbf{x}_i)| = \sum_{i=1}^n s_i(f(\mathbf{x}_i) - f_r^\sharp(\mathbf{x}_i))$$

$$= \sqrt{n}\langle \mathbf{s}, \mathbf{U}\boldsymbol{\beta}\rangle \leq \sqrt{n}\sqrt{\sum_{j=1}^r \frac{\beta_j^2}{\nu_j}}\sqrt{\sum_{j=1}^r \nu_j s_j^2} \leq \sqrt{n}\sqrt{\sum_{j=1}^r \frac{\beta_j^2}{\nu_j}}\sqrt{\sum_{j=1}^r \nu_j} = \sqrt{2n}R_r(\delta).$$

Consequently, for each $i \in [n]$, we have

$$\left|L(y_i, f(\mathbf{x}_i)) - L(y_i, f_r^\sharp(\mathbf{x}_i))\right| \leq M|f(\mathbf{x}_i) - f_r^\sharp(\mathbf{x}_i)| \leq M\sqrt{2n}R_r(\delta).$$

In addition, note that

$$\frac{1}{n}\sum_{i=1}^n \mathbb{E}\left[\left(L(y_i, f(\mathbf{x}_i)) - L(y_i, f_r^\sharp(\mathbf{x}_i))\right)^2\right]$$

$$\leq \frac{M^2}{n}\sum_{i=1}^n (f(\mathbf{x}_i) - f_r^\sharp(\mathbf{x}_i))^2 = M^2\|f - f_r^\sharp\|_n^2$$

$$= M^2\langle \mathbf{U}\boldsymbol{\beta}, \mathbf{U}\boldsymbol{\beta}\rangle = M^2\langle \boldsymbol{\beta}, \boldsymbol{\beta}\rangle \leq M^2 \max_{i \in [r]} \nu_j \sum_{i=1}^r \frac{\beta_j^2}{\nu_j} \leq 2M^2 \sum_{i=1}^r \nu_j = 2M^2 n R_r^2(\delta).$$

By a similar rescaled method, we have

$$\frac{\left|L(y_i, f(\mathbf{x}_i)) - L(y_i, f_r^\sharp(\mathbf{x}_i))\right|}{W_r(f, \delta)} \leq \frac{M|f(\mathbf{x}_i) - f_r^\sharp(\mathbf{x}_i)|}{W_r(f, \delta)}$$

$$= M\left|\widetilde{f}(\mathbf{x}_i) - f_r^\sharp(\mathbf{x}_i)\right| \leq M\sqrt{2}nR_r(\delta),$$

and

$$\frac{1}{n}\sum_{i=1}^n \mathbb{E}\left[\left(\frac{L(y_i, f(\mathbf{x}_i)) - L(y_i, f_r^\sharp(\mathbf{x}_i))}{W_r(f, \delta)}\right)^2\right] \leq \frac{M^2}{n}\sum_{i=1}^n \left(\frac{f(\mathbf{x}_i) - f_r^\sharp(\mathbf{x}_i)}{W_r(f, \delta)}\right)^2$$

$$= \frac{M^2}{n}\sum_{i=1}^n \left(\widetilde{f}(\mathbf{x}_i) - f_r^\sharp(\mathbf{x}_i)\right)^2$$

$$= 2M^2 nR_r^2(\delta).$$

Thus, the requirements in Lemma G.1 hold with $\eta = M\sqrt{2}nR_r(\delta)$ and $\zeta^2 = 2M^2 nR_r^2(\delta)$. Then, by applying Lemma G.1, for any $\iota \in (0, 1)$, let $t = \sqrt{\frac{1}{n}\log \iota^{-1}}$, it holds at with probability at least $1 - \iota$ that

$$\mathcal{A}_r - \mathbb{E}(\mathcal{A}_r) \leq \sqrt{\frac{\log \iota^{-1}}{n}\left(4M^2 nR_r^2(\delta) + 4M\sqrt{2}nR_r(\delta)\mathbb{E}(\mathcal{A}_r)\right)} + \frac{2\sqrt{2}M}{3}R_r(\delta)\log \iota^{-1} \tag{30}$$

$$\overset{(i)}{\leq} \widetilde{C}\log \iota^{-1} R_r(\delta),$$

where $\widetilde{C} = \sqrt{(4M^2 + 16\sqrt{2\pi}M^2)} + \frac{2\sqrt{2}M}{3}$ and $(i)$ follows from (29).

Therefore, by combining (29) and (30), it holds at with probability at least $1 - \iota$ that

$$\mathcal{A}_r \leq C\log \iota^{-1} R_r(\delta), \tag{31}$$

where $C = \sqrt{(4M^2 + 16\sqrt{2\pi}M^2)} + \frac{2\sqrt{2}M}{3} + 4\sqrt{\pi}M$.

This completes the proof of Theorem 4.2. $\qquad\square$

### D.2.2 Proof of Lemma D.2

Denote

$$\Delta_r = \frac{c_0}{4}\delta_{n,r}^{4\eta} + 4u^2\lambda^{2\eta} + (c_0' + c_0)\sum_{j=r+1}^n \xi_j^{*2}.$$

On the event $\mathcal{V}_r(\iota, \delta_{n,r})$, we claim that

$$\left\|\widehat{f}_{\lambda,r} - f_r^\sharp\right\|_n^2 \leq C\Delta_r.$$

By the optimality of $\widehat{f}_{\lambda,r}$ and the feasibility of $f_r^\sharp$, we have

$$\widehat{\mathcal{E}}(\widehat{f}_{\lambda,r}) - \widehat{\mathcal{E}}(f_r^\sharp) + \lambda\|\widehat{f}_{\lambda,r}\|_{K_r}^2 - \lambda\|f_r^\sharp\|_{K_r}^2 \leq 0.$$

Define the function class

$$\mathcal{G}_r := \left\{f \in \mathcal{H}_{n,r} : \left\|f - f_r^\sharp\right\|_n^2 \leq C\Delta_r, \ \left\|f - f_r^\sharp\right\|_K \leq b\right\}.$$

By following a similar argument as that in the proof of Lemma C.2, it suffices to prove that

$$\widehat{\mathcal{E}}(\widetilde{f}_r) - \widehat{\mathcal{E}}(f_r^\sharp) + \lambda\|\widetilde{f}_r\|_K^2 - \lambda\|f_r^\sharp\|_K^2 > 0, \tag{32}$$

where $\widetilde{f}_r$ is some function belonging to the boundary of $\mathcal{G}_r$.

It follows from Assumption 3.1 that

$$c_0\big\|\widetilde{f}_r - f^*\big\|_n^2 \leq \mathcal{E}(\widetilde{f}_r) - \mathcal{E}(f^*)$$

and

$$\mathcal{E}(f_r^\sharp) - \mathcal{E}(f^*) \leq c_0'\big\|f_r^\sharp - f^*\big\|_n^2.$$

Then, we have

$$
\begin{aligned}
c_0\big\|\widetilde{f}_r - f_r^\sharp\big\|_n^2 &- (2c_0' + 2c_0)\big\|f_r^\sharp - f^*\big\|_n^2 \\
&\overset{(i)}{\leq} 2c_0\big\|\widetilde{f}_r - f^*\big\|_n^2 - 2c_0'\big\|f_r^\sharp - f^*\big\|_n^2 \\
&\leq 2\mathcal{E}(\widetilde{f}_r) - \mathcal{E}(f^*) - 2(\mathcal{E}(f_r^\sharp) - \mathcal{E}(f^*)) \\
&= 2\big(\mathcal{E}(\widetilde{f}_r) - \mathcal{E}(f_r^\sharp)\big),
\end{aligned}
\tag{33}
$$

where (i) uses the elementary inequality.

Note that (33) establishes a connection between the excess risk and $\mathcal{L}(\mathbb{P}_n)$-norm at $f_r^\sharp$, which allows us to prove Lemma D.2 by using a similar argument as the proof for Lemma C.2.

Below we separately consider two cases: (i) $\|\widetilde{f}_r - f_r^\sharp\|_n^2 = C\Delta_r$ and $\|\widetilde{f}_r - f_r^\sharp\|_{K_r} \leq b$; and (ii) $\|\widetilde{f}_r - f_r^\sharp\|_n^2 \leq C\Delta_r$ and $\|\widetilde{f}_r - f_r^\sharp\|_{K_r} = b$.

*Case (i):* On the event $\mathcal{V}_r(\iota, \delta_{n,r})$, we have

$$
\begin{aligned}
\widehat{\mathcal{E}}(f_r^\sharp) &- \widehat{\mathcal{E}}(\widetilde{f}_r) \\
&\leq \mathcal{E}(f_r^\sharp) - \mathcal{E}(\widetilde{f}_r) + C\log\iota^{-1} R_r(\delta_{n,r}) W_r(\widetilde{f}_r, \delta_{n,r}) \\
&\overset{(i)}{\leq} -\frac{c_0}{2}\big\|\widetilde{f}_r - f_r^\sharp\big\|_n^2 + (c_0' + c_0)\big\|f_r^\sharp - f^*\big\|_n^2 + C\log\iota^{-1} R_r(\delta_{n,r})\left(\delta_{n,r}^{-1}\big\|\widetilde{f}_r - f_r^\sharp\big\|_n + \big\|\widetilde{f}_r - f_r^\sharp\big\|_{K_r}\right) \\
&\overset{(ii)}{=} -\frac{c_0}{2}C\Delta_r + (c_0' + c_0)\sum_{j=r+1}^{n}\xi_j^{*2} + C\log\iota^{-1} R_r(\delta_{n,r})\left(\delta_{n,r}^{-1}\sqrt{C\Delta_r} + \big\|\widetilde{f}_r - f_r^\sharp\big\|_{K_r}\right),
\end{aligned}
$$

where (i) follows from (33), and (ii) follows from $\|\widetilde{f}_r - f_r^\sharp\|_n^2 = C\Delta_r$ and (27).

Recall that $\delta_{n,r}$ satisfies

$$C\log\iota^{-1} R_r(\delta_{n,r}) \leq \frac{c_0}{2}\delta_{n,r}^{2\eta+1}$$

and we choose $\lambda$ satisfying $\lambda \geq \delta_{n,r}^2$. Following a similar argument as the proof for Lemma C.2, we have

$$\widehat{\mathcal{E}}(f_r^\sharp) - \widehat{\mathcal{E}}(\widetilde{f}_r) \leq -\frac{c_0}{2}C\Delta_r + (c_0' + c_0)\sum_{j=r+1}^{n}\xi_j^{*2} + \frac{c_0}{2}\delta_{n,r}^{2\eta}\sqrt{C\Delta_r} + \frac{c_0}{4}\delta_{n,r}^{4\eta} + \frac{\lambda}{4}\big\|\widetilde{f}_r - f_r^\sharp\big\|_{K_r}^2.$$

Since $\widetilde{f}_r \in \mathcal{H}_{n,r}$, there exists $\widetilde{\alpha}_r \in \mathcal{R}^n$ such that

$$\widetilde{f}_r = S_{\mathbf{x},r}^\top(\widetilde{\alpha}_r).$$

Note that

$$\lambda\big\|f_r^\sharp\big\|_{K_r}^2 - \lambda\big\|\widetilde{f}_r\big\|_{K_r}^2 = -2\lambda\langle f_r^\sharp, \widetilde{f}_r - f_r^\sharp\rangle_{K_r} - \lambda\big\|\widetilde{f}_r - f_r^\sharp\big\|_{K_r}^2. \tag{34}$$

Following the similar treatment as that in the proof of Lemma C.2 with $\mathbf{K}$ replaced by $\mathbf{K}_r$ and $\widetilde{\alpha}$ replaced by $\widetilde{\alpha}_r$, we have

$$
\begin{aligned}
\big|\lambda\langle f_r^\sharp, \widetilde{f}_r - f_r^\sharp\rangle_{K_r}\big| &\leq u\lambda^{\eta+\frac{1}{2}}\big\|\widetilde{f}_r - f_r^\sharp\big\|_{K_r} + u\lambda^\eta\big\|\widetilde{f}_r - f_r^\sharp\big\|_n \\
&= u\lambda^{\eta+\frac{1}{2}}\big\|\widetilde{f}_r - f_r^\sharp\big\|_{K_r} + u\lambda^\eta\sqrt{C\Delta_r}. \tag{35}
\end{aligned}
$$

Put the pieces together, and repeat the similar argument as that in the proof of Lemma C.2, for all $\gamma \geq \frac{1}{2}$, we obtain

$$\widehat{\mathcal{E}}(f_r^\sharp) - \widehat{\mathcal{E}}(\widetilde{f}_r) + \lambda\|f_r^\sharp\|_{K_r}^2 - \lambda\|\widetilde{f}_r\|_{K_r}^2$$

$$\leq -\frac{c_0}{2}C\Delta_r + (c_0' + c_0)\sum_{j=r+1}^n \xi_j^{*2} + \left(\frac{c_0}{2}\delta_{n,r}^{2\eta} + 2u\lambda^\eta\right)\sqrt{C\Delta_r} + \frac{c_0}{4}\delta_{n,r}^{4\eta} + 4u^2\lambda^{2\eta}$$

and for sufficiently large $C$, we have

$$\widehat{\mathcal{E}}(f_r^\sharp) - \widehat{\mathcal{E}}(\widetilde{f}_r) + \lambda\|f_r^\sharp\|_{K_r}^2 - \lambda\|\widetilde{f}_r\|_{K_r}^2 < 0.$$

*Case (ii)*: Repeat the similar argument as that in Case (i), on the event $\mathcal{V}_r(\iota, \delta_{n,r})$ and by Assumption 3.1, we have

$$\widehat{\mathcal{E}}(f_r^\sharp) - \widehat{\mathcal{E}}(\widetilde{f}_r)$$

$$\leq -\frac{c_0}{2}\|\widetilde{f}_r - f_r^\sharp\|_n^2 + (c_0' + c_0)\sum_{j=r+1}^n \xi_j^{*2} + C\log\iota^{-1}R_r(\delta_{n,r})\left(\delta_{n,r}^{-1}\|\widetilde{f}_r - f_r^\sharp\|_n + \|\widetilde{f}_r - f_r^\sharp\|_{K_r}\right)$$

$$\leq (c_0' + c_0)\sum_{j=r+1}^n \xi_j^{*2} + C\log\iota^{-1}R_r(\delta_{n,r})\left(\delta_{n,r}^{-1}\|\widetilde{f}_r - f_r^\sharp\|_n + \|\widetilde{f}_r - f_r^\sharp\|_{K_r}\right)$$

$$\leq (c_0' + c_0)\sum_{j=r+1}^n \xi_j^{*2} + \frac{c_0}{2}\delta_{n,r}^{2\eta}\sqrt{C\Delta_r} + \frac{c_0}{4}\delta_{n,r}^{4\eta+2}\lambda^{-1} + \frac{\lambda}{4}\|\widetilde{f}_r - f_r^\sharp\|_{K_r}^2$$

$$\leq (c_0' + c_0)\sum_{j=r+1}^n \xi_j^{*2} + \frac{c_0}{2}\delta_{n,r}^{2\eta}\sqrt{C\Delta_r} + \frac{c_0}{4}\delta_{n,r}^{4\eta} + \frac{\lambda}{4}\|\widetilde{f}_r - f_r^\sharp\|_{K_r}^2,$$

where the last step holds with the choice of $\lambda$ satisfying $\lambda \geq \delta_{n,r}^2$.

Combine with (34) and (35) and by applying the elementary inequality to obtain

$$\widehat{\mathcal{E}}(f_r^\sharp) - \widehat{\mathcal{E}}(\widetilde{f}_r) + \lambda\|f_r^\sharp\|_{K_r}^2 - \lambda\|\widetilde{f}_r\|_{K_r}^2$$

$$\leq (c_0' + c_0)\sum_{j=r+1}^n \xi_j^{*2} + \left(\frac{c_0}{2}\delta_{n,r}^{2\eta} + 2u\lambda^\eta\right)\sqrt{C\Delta_r} + \frac{c_0}{4}\delta_{n,r}^{4\eta} + 4u^2\lambda^{2\eta} - \frac{\lambda}{2}\|\widetilde{f}_r - f_r^\sharp\|_{K_r}^2.$$

By our choice that $0 < \lambda \leq 1$, $\lambda \geq \max\{\delta_{n,r}^2, \sum_{j=r+1}^n \xi_j^{*2}\}$, we have

$$\lambda \geq \lambda^{2\eta}, \quad \lambda \geq \delta_{n,r}^{4\eta}, \quad \text{and} \quad \lambda \geq \sum_{j=r+1}^n \xi_j^{*2}.$$

Note that in this case, $\|\widetilde{f}_r - f_r^\sharp\|_{K_r}^2 = b^2$. Therefore, repeat the similar argument as that in the proof of Lemma C.2, for a sufficiently large constant $b$, the RHS of the above inequality is less than 0.

By combining Cases (i) and (ii), on the event $\mathcal{V}_r(\iota, \delta_{n,r})$, we have

$$\left\|\widehat{f}_{\lambda,r} - f_r^\sharp\right\|_n^2 \leq C\Delta_r \lesssim \delta_{n,r}^{4\eta} + \lambda^{2\eta} + \sum_{j=r+1}^n \xi_j^{*2},$$

which completes the proof. $\qquad\square$

### D.3 Proof of Corollary 4.3

From (27)

$$\left\|f_r^\sharp - f^*\right\|_n^2 = \sum_{j=r+1}^n \xi_j^{*2}.$$

For the approximation bias, according to the polynomial assumption that $\xi_j^* \asymp j^{-2\gamma\alpha-1}$, we have

$$\|f_r^\sharp - f^*\|_n^2 = \sum_{j=r+1}^{n} \xi_j^{*2} \leq C \sum_{j=r+1}^{n} j^{-2\gamma\alpha-1} \leq C \int_r^\infty t^{-2\gamma\alpha-1} dt \leq C r^{-2\gamma\alpha}.$$

For the estimation error, we control the truncated kernel function first. Similar to the equality (20) for $R(\delta)$, if $r > d(\delta)$, we have

$$R_r(\delta) = \sqrt{\frac{1}{n} d(\delta)\delta^2 + \frac{1}{n} \sum_{d(\delta)+1}^{r} \mu_j}.$$

Moreover, for the regular kernel class, we have $R_r(\delta) \asymp \sqrt{d(\delta)\delta^2/n}$. If $r \leq d(\delta)$, we have $R_r(\delta) \asymp \sqrt{r\delta^2/n}$. Combining these two results, we have

$$R_r(\delta) \asymp \sqrt{\frac{1}{n} \min\{r, d(\delta)\}\delta^2}. \tag{36}$$

Next, we split the remaining proof by considering two cases: (i) $\frac{1}{2} \leq \gamma \leq 1$; and (ii) $\gamma > 1$.

*Case (i):* Recall that for the eigenvalues of $\mathbf{K}$ that satisfy $\mu_j \asymp j^{-\alpha}$, we have $d(\delta) \asymp \delta^{-2/\alpha}$. In addition, we notice that in this case, the best truncation level $r$ to balance $\delta^{4\eta}$ and $r^{-2\gamma\alpha}$ is $r \asymp d(\delta)$. This means that whatever $r$ is, we always have $R_r(\delta) \asymp \sqrt{\frac{1}{n} d(\delta)\delta^2}$. Hence, the kernel complexity remains the same, and to avoid introducing additional approximation bias, the best choice of truncation level turns out to be $r = n$. Then, following a similar argument as that in Section C.3, we have $\delta_{n,r}^2 \leq C\left(\frac{(\log \iota^{-1})^2}{n}\right)^{\frac{\alpha}{2\gamma\alpha+1}}$. Choosing the optimal parameter of $\lambda \asymp \delta_{n,r}^2$ yields

$$\mathcal{E}(\widehat{f}_{\lambda,r}) - \mathcal{E}(f^*) \asymp \|\widehat{f}_{\lambda,r} - f^*\|_n^2 \leq C\left(\frac{(\log \iota^{-1})^2}{n}\right)^{\frac{2\gamma\alpha}{2\gamma\alpha+1}}.$$

*Case (ii):* In this case, the best truncation level $r$ to balance $\delta^{4\eta}$ and $r^{-2\gamma\alpha}$ is $r \asymp \delta^{-2/(\gamma\alpha)}$, which implies $r \lesssim d(\delta)$ so that from (36), we have

$$R_r(\delta) \asymp \sqrt{\frac{1}{n} r\delta^2} \asymp \sqrt{\frac{1}{n} \delta^{\frac{2\alpha\gamma-2}{\alpha\gamma}}}.$$

Therefore, the solution to the inequality (3) can be upper bounded by the solution to

$$C \log \iota^{-1} \sqrt{\frac{1}{n} \delta^{\frac{2\alpha\gamma-2}{\alpha\gamma}}} \leq \frac{c_0}{2} \delta^3.$$

Solving this inequality yields

$$\delta_{n,r}^2 \leq C\left(\frac{(\log \iota^{-1})^2}{n}\right)^{\frac{\gamma\alpha}{2\gamma\alpha+1}},$$

and we can choose

$$r \asymp \left(\frac{n}{(\log \iota^{-1})^2}\right)^{\frac{1}{2\gamma\alpha+1}}.$$

The desired upper bound in the case $\gamma > 1$ follows by choosing $\lambda \asymp \delta_{n,r}^2 \asymp r^{-2\gamma\alpha}$. By combining these two cases, we complete the proof. $\qquad\square$

# E   Proof of Theorem 4.4

We consider the special case that the data is generated according to the mean regression model

$$Y_i = f^*(\mathbf{x}_i) + \varepsilon_i \quad \text{with} \quad \varepsilon_i \sim N(0,1)$$

for each $i \in [n]$. For this mean regression model, $f^*$ is the minimizer of the population risk $\mathcal{E}(f)$ with squared loss specified.

For any $\delta > 0$ and $\gamma \geq \frac{1}{2}$, define the ellipse class

$$\mathcal{E}_\gamma(\delta) := \left\{ \xi = (\xi_1, ..., \xi_n)^\top \in \mathcal{R}^n : \sum_{j=1}^n \frac{\xi_j^2}{(\min\{\delta^2, \mu_j\})^{2\gamma}} \leq u^2 \right\}.$$

For $\xi \in \mathcal{R}^n$, define the rescaled norm

$$\|\xi\|_{\mathcal{E}_\gamma}^2 := \sum_{j=1}^n \frac{\xi_j^2}{(\min\{\delta^2, \mu_j\})^{2\gamma}}.$$

Then, it is equivalent to write

$$\mathcal{E}_\gamma(\delta) = \left\{ \xi = (\xi_1, ..., \xi_n)^\top \in \mathcal{R}^n : \|\xi\|_{\mathcal{E}_\gamma}^2 \leq u^2 \right\}.$$

Recall that the statistical dimension is defined as

$$d(\delta) = \min\left\{ j \in [n] : \mu_j \leq \delta^2 \right\}.$$

Our main proof is based on the following lemma that states a result concerning metric entropy.

**Lemma E.1.** *For any $\delta > 0$ and $\gamma \geq \frac{1}{2}$, there is a collection of $\frac{1}{2}\delta^{2\gamma}$-separated points $\{\xi^1, ..., \xi^M\}$ in $\mathcal{E}_\gamma(\delta)$ such that $\log M \geq \frac{1}{32} d(\delta)$.*

By using Lemma E.1, there exists a $\frac{1}{2}\delta^{2\gamma}$-separated collection of points $\{\xi^1, ..., \xi^M\}$ in $\mathcal{E}_\gamma(\delta)$ such that $\log M \geq \frac{1}{32} d(\delta)$. Given $\{\xi^1, ..., \xi^M\}$, we construct $f^1, ..., f^M$ as $f^i = S_\mathbf{x}^\top(\mathbf{U}\mathbf{D}^{-1}\xi^i)$. Note that the TA scores corresponding to $f^i$ are given by

$$\mathbf{U}^\top S_\mathbf{x}(f^i) = \mathbf{U}^\top \mathbf{K} \mathbf{U} \mathbf{D}^{-1} \xi^i = \xi^i.$$

Hence, $\{\xi^1, ..., \xi^M\} \subset \mathcal{E}_\gamma(\delta)$ implies $f^i \in \mathcal{H}_K^b$ for each $i \in [M]$. Moreover, we have

$$\|f^i - f^j\|_n^2 = \|\mathbf{D}\mathbf{U}^\top(\mathbf{U}\mathbf{D}^{-1}\xi^i - \mathbf{U}\mathbf{D}^{-1}\xi^j)\|_2^2 = \|\xi^i - \xi^j\|_2^2 \geq \frac{\delta^{4\gamma}}{4},$$

which implies that $\{f^1, ..., f^M\}$ is $\frac{1}{2}\delta^{2\gamma}$-separated in $\mathcal{H}_K^b$.

Since $\xi^i \in \mathcal{E}_\gamma(\delta)$, we also have

$$\|f^i\|_n^2 = \|\xi^i\|_2^2 = \sum_{k=1}^n \xi_k^{i\,2} = \delta^{4\gamma} \sum_{k=1}^n \frac{\xi_k^{i\,2}}{\delta^{4\gamma}} \leq u^2 \delta^{4\gamma}.$$

Therefore, by using the triangle inequality, we have

$$\|f^i - f^j\|_n^2 \leq 2u^2\delta^{4\gamma} \quad \text{for each } i, j \in [M].$$

Let $\rho^k$ be the underlying distribution of the collected data $\{(\mathbf{x}_i, y_i)\}_{i=1}^n$ corresponding to $f^k$. Then, there holds

$$\mathrm{KL}(\rho^i\|\rho^j) \overset{(i)}{=} \sum_{i=1}^n \mathrm{KL}(N(f^i(\mathbf{x}_i), 1)\|N(f^j(\mathbf{x}_i), 1)) \overset{(ii)}{=} \frac{n}{2}\|f_i - f_j\|_n^2 \leq u^2 n \delta^{4\gamma}$$

where $\mathrm{KL}(\cdot\|\cdot)$ denotes the KL divergence between two distributions, (i) follows from the fact that $\mathrm{KL}(P_1 \otimes P_2\|Q_1 \otimes Q_2) = \mathrm{KL}(P_1\|Q_1) + \mathrm{KL}(P_2\|Q_2)$ and $\otimes$ denoting the product measure, and (ii) follows from the fact $\mathrm{KL}(N(\mu_1, \sigma^2)\|N(\mu_2, \sigma^2)) = \frac{(\mu_1 - \mu_2)^2}{2\sigma^2}$.

Below is devoted to establishing the minimax lower bound by applying the standard Fano's method (see, for instance, Proposition 15.12 in Wainwright (2019)). To be specific, for $\delta > 0$ and for any estimator $\widetilde{f}$ based on the data $\{(\mathbf{x}_i, y_i)\}_{i=1}^n$, we have

$$\inf_{\widetilde{f}} \sup_{f^* \in \mathcal{H}_K^b} P\left( \|\widetilde{f} - f^*\|_n^2 \geq \frac{\delta^{4\gamma}}{4} \right) \geq 1 - \frac{\max_{1 \leq i, j \leq M} \mathrm{KL}(\rho^i\|\rho^j) + \log 2}{\log M}$$

$$\geq 1 - \frac{u^2 n \delta^{4\gamma} + \log 2}{d(\delta)/32}. \tag{37}$$

Below we separately consider two cases: i) $\frac{1}{2} \leq \gamma \leq 1$; and ii) $\gamma > 1$.

*Case i:* For $\frac{1}{2} \leq \gamma \leq 1$, we take $\delta = (4c)^{\frac{1}{4\gamma}} \delta_n$, where $\delta_n$ is the critical radius defined as the smallest solution to (2) in Section 3. Plugging into (37) yields

$$
\begin{aligned}
\inf_{\widetilde{f}} \sup_{f^* \in \mathcal{H}_K^b} P\left(\|\widetilde{f} - f^*\|_n^2 \geq c\delta_n^{4\gamma}\right) &\geq 1 - \frac{4cu^2 n\delta_n^{4\gamma} + \log 2}{d((4c)^{\frac{1}{4\gamma}}\delta_n)/32} \\
&\geq 1 - \frac{4cu^2 n\delta_n^{4\gamma} + \log 2}{d_n/32},
\end{aligned}
\tag{38}
$$

where $d_n = d(\delta_n)$ and the last inequality holds since $d(\delta)$ is decreasing as $\delta$ grows and $(4c)^{\frac{1}{4\gamma}}\delta_n \leq \delta_n$ for sufficiently small $c$.

Recall that for $\frac{1}{2} \leq \gamma \leq 1$, $\delta_n$ is smallest solution to

$$
C \log \iota^{-1} R(\delta) \leq \frac{c_0}{2} \delta^{2\gamma+1}.
$$

Moreover, for the regular kernel, we have $R(\delta) \asymp \sqrt{\frac{1}{n}d(\delta)\delta^2}$, which implies

$$
\delta_n^{2\gamma+1} \lesssim \sqrt{\frac{1}{n}d(\delta_n)\delta_n^2} = \sqrt{\frac{1}{n}d_n\delta_n^2}.
$$

Then, $d_n \geq c_1 n\delta_n^{4\gamma}$ for some universal constant $c_1$. Plugging this inequality into (38) yields

$$
\inf_{\widetilde{f}} \sup_{f^* \in \mathcal{H}_K^b} P\left(\|\widetilde{f} - f^*\|_n^2 \geq c\delta_n^{4\gamma}\right) \geq 1 - \frac{4cu^2 n\delta_n^{4\gamma} + \log 2}{c_1 n\delta_n^{4\gamma}} \geq \frac{1}{2},
$$

where the last step holds for sufficiently small $c$.

*Case ii:* For $\gamma > 1$, we take $\delta = (4c)^{\frac{1}{4\gamma}} \delta_{n,r}^{1/\gamma}$, where $\delta_{n,r}$ is the critical radius defined as the smallest solution to (3) in Section 4. It follows from (37) that

$$
\begin{aligned}
\inf_{\widetilde{f}} \sup_{f^* \in \mathcal{H}_K^b} P\left(\|\widetilde{f} - f^*\|_n^2 \geq c\delta_{n,r}^4\right) &\geq 1 - \frac{4cu^2 n\delta_{n,r}^4 + \log 2}{d((4c)^{\frac{1}{4\gamma}}\delta_{n,r}^{1/\gamma})/32} \\
&\geq 1 - \frac{4cu^2 n\delta_{n,r}^4 + \log 2}{d(\delta_{n,r}^{1/\gamma})/32}.
\end{aligned}
$$

Recall that for $\gamma > 1$, $\delta_{n,r}$ is smallest solution to

$$
C \log \iota^{-1} R_r(\delta) \leq \frac{c_0}{2} \delta^3.
$$

According to (36), if $r \leq d(\delta)$, we have

$$
R_r(\delta) = \sqrt{\frac{1}{n}r\delta^2}.
$$

Hence, we have

$$
\delta_{n,r}^3 \lesssim \sqrt{\frac{1}{n}r\delta^2} = \sqrt{\frac{1}{n}r\delta^2},
$$

which leads to $r \geq c_2 n\delta_{n,r}^4$ for some universal constant $c_2$. Then, if we choose $r \asymp d(\delta_{n,r}^{1/\gamma})$ satisfying $d(\delta_{n,r}^{1/\gamma}) \leq d(\delta_{n,r})$, we have

$$
\inf_{\widetilde{f}} \sup_{f^* \in \mathcal{H}_K^b} P\left(\|\widetilde{f} - f^*\|_n^2 \geq c\delta_{n,r}^4\right) \geq 1 - \frac{4u^2 cn\delta_{n,r}^4 + \log 2}{Cc_2 n\delta_{n,r}^4/32} \geq \frac{1}{2},
$$

where the last step holds for sufficiently small $c$. This completes the proof of Theorem 4.4. $\qquad\square$

*Remark* E.2. In the proof for the lower bound, we consider the Gaussian noise case. However, for the upper bound, we require $\mathcal{Y} \subset [-U, U]$ when the squared loss is specified. The bounded range assumption essentially requires the random noise to be uniformly bounded. Nevertheless, the upper bound established in this paper can be also extended to the sub-Gaussian noise case with a slight order sacrifice of some $\log$ factors in the upper bound. Specifically, suppose that $\{\varepsilon_i\}_{i=1}^n$ are i.i.d. sub-Gaussian variables: that is, there exist positive constants $c, \sigma^2$ such that $P(|\varepsilon_i| > t) \leq c \exp(-\sigma^2 t^2)$ for all $t \geq 0$. Then, by the union bound, we have

$$P\big( \max_{i=1,\dots,n} |\varepsilon_i| > t \big) \leq P\Big( \bigcup_{i=1}^n \{|\varepsilon_i| > t\} \Big) \leq \sum_{i=1}^n c \exp(-\sigma^2 t^2) = cn \exp(-\sigma^2 t^2).$$

Consequently, for any $\iota \in (0, 1)$, by taking $t = \sigma^{-1} \sqrt{\log\left(\frac{cn}{\iota}\right)}$, it holds with probability at least $1 - \iota$ that

$$\max_{i=1,\dots,n} |\varepsilon_i| \leq \sigma^{-1} \sqrt{\log\left(\frac{cn}{\iota}\right)} \lesssim \sqrt{\log \frac{1}{\iota}} + \sqrt{\log n}.$$

Further note that by the reproducing kernel property and our assumption that $\sup_{\mathbf{x}, \mathbf{x}'} K(\mathbf{x}, \mathbf{x}') \leq \kappa^2$, we have that for any $\mathbf{x} \in \mathcal{X}$

$$|f^*(\mathbf{x})| = |\langle f^*, K(\mathbf{x}, \cdot) \rangle_K| \leq \|f^*\|_K \|K(\mathbf{x}, \cdot)\|_K \leq \kappa \|f^*\|_K. \tag{39}$$

Therefore, for the sub-Gaussian noise case, we can immediately complete the proof by replacing $U$ with $U_{\iota,n} = \kappa \|f^*\|_K + C(\sqrt{\log \frac{1}{\iota}} + \sqrt{\log n})$. As a result, the upper bound for the sub-Gaussian noise case will align with that for the uniform bounded noise case up to some $\log$ factors.

## E.1 Proof of Lemma E.1

Lemma E.1 states a result concerning metric entropy, and its proof is motivated by that of Lemma 4 in Yang et al. (2017), which only focuses on the just-aligned regime $\gamma = \frac{1}{2}$.

For each $j \in [M]$, let

$$\xi^j = \left( \frac{\delta^{2\gamma}}{\sqrt{2d(\delta)}} w_1^j, \frac{\delta^{2\gamma}}{\sqrt{2d(\delta)}} w_2^j, \dots, \frac{\delta^{2\gamma}}{\sqrt{2d(\delta)}} w_{d(\delta)}^j, 0, \dots, 0 \right)^\top, \tag{40}$$

where

$$\mathbf{w}^1 = (w_1^1, \dots, w_{d(\delta)}^1)^\top, \dots, \mathbf{w}^M = (w_1^M, \dots, w_{d(\delta)}^M)^\top \sim N(0, \mathbf{I}_{d(\delta)})$$

are a collection of independent standard Gaussian vectors. We claim that with a probability greater $0$, we can find a set $\{\xi^1, \dots, \xi^M\}$ generated in the above manner that are $\delta^{2\gamma}$-separated in $\mathcal{E}(\delta)$ and $M \geq e^{\frac{1}{32} d(\delta)}$.

On one hand, to show that $\{\xi^1, \dots, \xi^M\} \subset \mathcal{E}_\gamma(\delta)$, we need to equivalently prove $\|\xi^i\|_{\mathcal{E}_\gamma}^2 \leq u^2$ for each $i \in [M]$. Indeed, for each index $i \in [M]$, since $\delta^2 \leq \mu_j$ for each $j \leq d(\delta)$, we have $\|\xi^i\|_{\mathcal{E}_\gamma}^2 = \frac{\|\mathbf{w}^i\|_2^2}{2d(\delta)}$. Note that $\|\mathbf{w}^i\|_2^2 \sim \chi_{d(\delta)}^2$. Then, by using the tail bound for chi-square distribution (Example 2.11 in Wainwright (2019)), we have

$$\mathbb{P}\left(\|\xi^i\|_{\mathcal{E}_\gamma}^2 \leq u^2\right) = \mathbb{P}\left(\frac{1}{d(\delta)} \|\mathbf{w}^i\|_2^2 - 1 \leq 2u^2 - 1\right)$$

$$\geq \mathbb{P}\left(\frac{1}{d(\delta)} \|\mathbf{w}^i\|_2^2 - 1 \leq 7\right) \geq 1 - e^{-\frac{49d(\delta)}{8}}, \tag{41}$$

where we use the assumption that $u \geq 2$. By applying the union bound, we have

$$\mathbb{P}\left(\|\xi^i\|_{\mathcal{E}_\gamma}^2 \leq u^2 \text{ for all } i \in [M]\right) \geq 1 - M e^{-\frac{49d(\delta)}{8}}. \tag{42}$$

On the other hand, note that $\|\xi^i - \xi^j\|_2^2 = \frac{\delta^{4\gamma}}{2d(\delta)}\|\mathbf{w}^i - \mathbf{w}^j\|_2^2$. Since $\mathbf{w}^i$ and $\mathbf{w}^j$ are independent, we have $(\mathbf{w}^i - \mathbf{w}^j)/\sqrt{2} \sim N(0, \mathbf{I}_{d(\delta)})$. Then, similar to the inequality (41), we also have

$$
\begin{aligned}
\mathbb{P}\left(\|\xi^i - \xi^j\|_2^2 \geq \frac{\delta^{4\gamma}}{4}\right) &= \mathbb{P}\left(\frac{1}{2d(\delta)}\|\mathbf{w}^i - \mathbf{w}^j\|_2^2 \geq \frac{1}{4}\right)\\
&= \mathbb{P}\left(\frac{1}{2d(\delta)}\|\mathbf{w}^i - \mathbf{w}^j\|_2^2 - 1 \geq -\frac{3}{4}\right) \geq 1 - e^{-\frac{9d(\delta)}{128}},
\end{aligned}
$$

and by applying the union bound, we have

$$
\mathbb{P}\left(\|\xi^i - \xi^j\|_2^2 \geq \frac{\delta^{4\gamma}}{4} \text{ for all } i, j \in [M]\right) \geq 1 - M^2 e^{-\frac{9d(\delta)}{128}}. \tag{43}
$$

Combining (42) and (43) yields

$$
\mathbb{P}\left(\|\xi^i\|_{\mathcal{E}_\gamma}^2 \leq u^2 \text{ and } \|\xi^i - \xi^j\|_2^2 \geq \frac{\delta^{4\gamma}}{4} \text{ for all } i, j \in [M]\right) \geq 1 - M e^{-\frac{49d(\delta)}{8}} - M^2 e^{-\frac{9d(\delta)}{128}},
$$

where the left side is positive by setting $\log M = d(\delta)/32$.

We thus conclude the statement in Lemma E.1. $\qquad\square$

# F   More discussions on Assumption 3.1

As discussed in Section 3, Assumption 3.1 is a relatively mild condition for many widely used loss functions. It is clear that the squared loss satisfies Assumption 3.1 with $c_0 = c_0' = 1$. For the Huber loss $L_\tau(y, f(\mathbf{x})) = (y - f(\mathbf{x}))^2$ if $|y - f(\mathbf{x})| \leq \tau$, and $\tau|y - f(\mathbf{x})| - \frac{1}{2}\tau^2$ otherwise, since it is locally equivalent to the squared loss function, thus it satisfies Assumption 3.1 under some mild tail conditions on the noise term $Y - f^*(\mathbf{x})$ (Wainwright, 2019).

For the Hinge loss $L(y, f(\mathbf{x})) = \max\{1 - yf(\mathbf{x}), 0\}$ that is designed for the margin-based classification problem, as mentioned in (Wainwright, 2019), whether Assumption 3.1 holds hinges on the distribution of the covariates $\mathbf{x}$, and the hypothesis function class $\mathcal{F}$. We remark that for the classification problem, the theoretical guarantee for 0-1 loss is also crucial. Once the 0-1 loss is considered, one possible routine for establishing the theoretical results for 0-1 loss is to follow a similar technical treatment as that on Page 17 of Lai et al. (2024) with some slight modifications, where the bridge between the excess risk w.r.t 0-1 loss and mean squared error is established, and based on the result in Lai et al. (2024), the excess risk only gets a slower rate compared to the rates established in our paper.

For other loss functions, including the check loss, Logistic loss, and exponential loss, we provide a more detailed discussion and deduce some sufficient conditions to ensure the satisfaction of Assumption 3.1.

## F.1   Check loss

Let $\rho_\tau(t) = t(\tau - \mathbf{I}_{\{t \leq 0\}})$ and $L_\tau(y, f(\mathbf{x})) = \rho_\tau(y - f(\mathbf{x}))$. We next verify Assumption 3.1 if the following assumption holds.

**Assumption F.1.** Denote $F_{Y|X=\mathbf{x}}$ be the conditional distribution on $\mathcal{Y}$ given $X = \mathbf{x}$. We assume that there exist two constants $c_0, c_0'$ such that

$$
2c_0|y| \leq \left|F_{Y|X=\mathbf{x}_i}(f^*(\mathbf{x}_i) + y) - F_{Y|X=\mathbf{x}_i}(f^*(\mathbf{x}_i))\right| \leq 2c_0'|y|.
$$

**Proposition F.2.** *Under Assumption F.1, for any $b > 0$, both the local $c_0$-strong convexity and the local $c_0'$-smooth condition are satisfied.*

*Proof.* For each $i \in [n]$, denote $w = Y - f^*(\mathbf{x}_i)$ and $v = f(\mathbf{x}_i) - f^*(\mathbf{x}_i)$. By using Knight's identity (Equation B.3 in Belloni & Chernozhukov (2011)) that $\rho_\tau(w - v) - \rho_\tau(w) = -v(\tau -$

$I_{\{w\leq 0\}}) + \int_0^v (I_{\{w\leq t\}} - I_{\{w\leq 0\}})dt$ , we have

$$\mathbb{E}\big[\rho_\tau(Y - f(\mathbf{x}_i)) - \rho_\tau(Y - f^*(\mathbf{x}_i))\big]$$

$$= -\mathbb{E}[(f(\mathbf{x}_i) - f^*(\mathbf{x}_i))(\tau - I_{\{Y\leq f^*(\mathbf{x}_i)\}})] + \mathbb{E}\Big[\int_0^{f(\mathbf{x}_i)-f^*(\mathbf{x}_i)} (I_{\{Y\leq f^*(\mathbf{x}_i)+t\}} - I_{\{Y\leq f^*(\mathbf{x}_i)\}})dt\Big].$$

$$(44)$$

Recall the definition of $f^*$, we have

$$\mathbb{E}[(f(\mathbf{x}_i) - f^*(\mathbf{x}_i))(\tau - I_{\{Y\leq f^*(\mathbf{x}_i)\}})] = (f(\mathbf{x}_i) - f^*(\mathbf{x}_i))\mathbb{E}[(\tau - I_{\{Y\leq f^*(\mathbf{x}_i)\}})] = 0.$$

Now we consider the second term in the right hand of (44). It follows from Fubini's theorem that

$$\mathbb{E}\Big[\int_0^{f(\mathbf{x}_i)-f^*(\mathbf{x}_i)} (I_{\{Y\leq f^*(\mathbf{x}_i)+t\}} - I_{\{Y\leq f^*(\mathbf{x}_i)\}})dt\Big]$$

$$= \int_0^{f(\mathbf{x}_i)-f^*(\mathbf{x}_i)} \mathbb{E}[I_{\{Y\leq f^*(\mathbf{x}_i)+t\}} - I_{\{Y\leq f^*(\mathbf{x}_i)\}}]dt$$

$$= \int_0^{f(\mathbf{x}_i)-f^*(\mathbf{x}_i)} \big(F_{Y|X=\mathbf{x}_i}(f^*(\mathbf{x}_i) + t) - F_{Y|X=\mathbf{x}_i}(f^*(\mathbf{x}_i))\big)dt.$$

According to Assumption F.1, there holds

$$\int_0^{f(\mathbf{x}_i)-f^*(\mathbf{x}_i)} \big(F_{Y|X=\mathbf{x}_i}(f^*(\mathbf{x}_i) + t) - F_{Y|X=\mathbf{x}_i}(f^*(\mathbf{x}_i))\big)dt$$

$$\geq \int_0^{f(\mathbf{x}_i)-f^*(\mathbf{x}_i)} 2c_0|t|dt = c_0(f(\mathbf{x}_i) - f^*(\mathbf{x}_i))^2,$$

and

$$\int_0^{f(\mathbf{x}_i)-f^*(\mathbf{x}_i)} \big(F_{Y|X=\mathbf{x}_i}(f^*(\mathbf{x}_i) + t) - F_{Y|X=\mathbf{x}_i}(f^*(\mathbf{x}_i))\big)dt$$

$$\leq \int_0^{f(\mathbf{x}_i)-f^*(\mathbf{x}_i)} 2c_0'|t|dt = c_0'(f(\mathbf{x}_i) - f^*(\mathbf{x}_i))^2.$$

Then, we have

$$c_0(f(\mathbf{x}_i) - f^*(\mathbf{x}_i))^2 \leq \mathbb{E}\Big[\int_0^{f(\mathbf{x}_i)-f^*(\mathbf{x}_i)} (I_{\{Y\leq f^*(\mathbf{x}_i)+t\}} - I_{\{Y\leq f^*(\mathbf{x}_i)\}})dt\Big]$$

$$\leq c_0'(f(\mathbf{x}_i) - f^*(\mathbf{x}_i))^2.$$

The desired conclusion immediately follows by summing from $1$ to $n$. This completes the proof of Proposition F.2. $\qquad\square$

### F.2 Logistic loss

The Logistic loss $L(y, f(\mathbf{x})) = \log(1 + \exp(-yf(\mathbf{x})))$ is specified for the binary classification problem, where the response $y$ takes values in $\{-1, 1\}$. Simple algebra yields the first and second derivatives of $L(y, \theta)$ in the second argument that

$$\frac{\partial L}{\partial \theta} = \frac{-y\exp(-y\theta)}{1 + \exp(-y\theta)}$$

and

$$\frac{\partial^2 L}{\partial \theta^2} = \frac{y^2}{(\exp(-y\theta/2) + \exp(y\theta)/2)^2}.$$

It is clear that for any $\theta \in \mathcal{R}$, we have

$$\left|\frac{\partial L}{\partial \theta}\right| \leq 1 \quad \text{and} \quad \left|\frac{\partial^2 L}{\partial \theta^2}\right| \leq \frac{1}{4},$$

implying that $L(y, \cdot)$ is 1-Lipschitz continuous and the $c_0'$-local smoothness condition holds with $c_0' = \frac{1}{4}$.

Recall from (39). For $f \in \mathcal{H}_K$ and $\mathbf{x} \in \mathcal{X}$ satisfying $|f(\mathbf{x}) - f^*(\mathbf{x})| \leq D$, denote $B = \kappa \|f^*\|_K + D$, we have

$$\left| \frac{\partial^2 L}{\partial \theta^2} \Big|_{\theta = f(\mathbf{x})} \right| \geq \frac{1}{e^{-B} + e^B + 2},$$

implying the locally strong convexity condition holds with $c_0 = \frac{1}{e^{-B} + e^B + 2}$.

### F.3 Exponential loss

The Exponential loss $L(y, f(\mathbf{x})) = \exp(-yf(\mathbf{x}))$ is used in the AdaBoost algorithm designed for the classification problem, where $y \in \{-1, 1\}$. Note that the first and second derivatives of $L(y, \theta)$ in the second argument is given by

$$\frac{\partial L}{\partial \theta} = -ye^{-y\theta} \quad \text{and} \quad \frac{\partial^2 L}{\partial \theta^2} = e^{-y\theta}.$$

For any $\theta \in \mathcal{R}$, we have $\left| \frac{\partial L}{\partial \theta} \right| \leq 1$, which implies that $L$ is 1-Lipschitz continuous. For the locally strong convexity condition and local smoothness condition, with a similar argument as that for the Logistic loss, we have that

$$e^{-B} \leq \left| \frac{\partial^2 L}{\partial \theta^2} \right| \leq e^B.$$

This ensures that the local strong convexity condition holds with $c_0 = e^{-B}$ and the local smoothness condition holds with $c_0' = e^B$.

## G  Supporting Lemmas.

The following lemma presents Talagrand's concentration inequality for random elements taking values in some space $\mathcal{Z}$ (Bousquet, 2002; Lv et al., 2018). Detailed proofs can be found in Bousquet (2002).

**Lemma G.1** (Talagrand's concentration inequality). *Let $Z_1, \ldots, Z_n$ be independent random elements taking values in some space $\mathcal{Z}$ equipped with norm $\|\cdot\|$. Let $\mathcal{F}$ be a class of real-valued measurable functions acting on $\mathcal{Z}$. If we have*

$$\max_{i \in [n]} \|f(Z_i)\| \leq \eta \quad \text{and} \quad \frac{1}{n} \sum_{i=1}^n \text{Var}(f(Z_i)) \leq \zeta^2, \quad \forall f \in \mathcal{F},$$

*define the empirical process $\boldsymbol{Z} := \sup_{f \in \mathcal{F}} |\frac{1}{n} \sum_{i=1}^n (f(Z_i) - \mathbb{E}f(Z_i))|$, then for any $t > 0$*

$$\mathbb{P}\left( \boldsymbol{Z} \geq \mathbb{E}(\boldsymbol{Z}) + t\sqrt{2\left(\zeta^2 + 2\eta\mathbb{E}(\boldsymbol{Z})\right)} + \frac{2\eta t^2}{3} \right) \leq \exp(-nt^2).$$

The following lemma is known as the symmetrization technique, which provides a fundamental tool to bound from above the expectation of the empirical process (Wainwright, 2019). A typical version of the symmetrization lemma is to consider the Rademacher variables $\{\varepsilon_1, ..., \varepsilon_n\}$, i.e. $\mathbb{P}(\varepsilon_i = 1) = \mathbb{P}(\varepsilon_i = -1) = \frac{1}{2}$ (Proposition 4.11 in Wainwright (2019)). In our proof, we consider a sequence of standard Gaussian variables to utilize the rotation invariance of the Gaussian vector.

**Lemma G.2** (Symmetrization). *Let $X_1, ..., X_n$ be a sequence of random variables and $w_1, ..., w_n \sim N(0, 1)$ denote the standard Gaussian variables independent of $X_1, ..., X_n$. For any measurable function class $\mathcal{F}$, we have*

$$\mathbb{E}\left[ \frac{1}{n} \sup_{f \in \mathcal{F}} \left| \sum_{i=1}^n (f(X_i) - \mathbb{E}[f(X_i)]) \right| \right] \leq \sqrt{2\pi} \mathbb{E}\left[ \frac{1}{n} \sup_{f \in \mathcal{F}} \left| \sum_{i=1}^n w_i f(X_i) \right| \right]$$

*Proof.* By applying Proposition 4.11 in Wainwright (2019), we have

$$\mathbb{E}\Big[\frac{1}{n}\sup_{f\in\mathcal{F}}\big|\sum_{i=1}^{n}(f(X_i)-\mathbb{E}[f(X_i)])\big|\Big] \leq 2\mathbb{E}\Big[\frac{1}{n}\sup_{f\in\mathcal{F}}\big|\sum_{i=1}^{n}\varepsilon_i f(X_i)\big|\Big].$$

Therefore, it remains to prove that

$$\mathbb{E}\Big[\frac{1}{n}\sup_{f\in\mathcal{F}}\big|\sum_{i=1}^{n}\varepsilon_i f(X_i)\big|\Big] \leq \sqrt{\frac{\pi}{2}}\mathbb{E}\Big[\frac{1}{n}\sup_{f\in\mathcal{F}}\big|\sum_{i=1}^{n}w_i f(X_i)\big|\Big]$$

Indeed, we have

$$\mathbb{E}_\varepsilon\Big[\frac{1}{n}\sup_{f\in\mathcal{F}}\big|\sum_{i=1}^{n}\varepsilon_i f(X_i)\big|\Big] \overset{(i)}{=} \sqrt{\frac{\pi}{2}}\mathbb{E}_\varepsilon\Big[\frac{1}{n}\sup_{f\in\mathcal{F}}\big|\mathbb{E}_{\mathbf{w}}\big(\sum_{i=1}^{n}|w_i|\varepsilon_i f(X_i)\big)\big|\Big]$$

$$\overset{(ii)}{\leq} \sqrt{\frac{\pi}{2}}\mathbb{E}_\varepsilon\Big[\frac{1}{n}\mathbb{E}_{\mathbf{w}}\sup_{f\in\mathcal{F}}\big|\sum_{i=1}^{n}|w_i|\varepsilon_i f(X_i)\big|\Big]$$

$$= \sqrt{\frac{\pi}{2}}\mathbb{E}\Big[\frac{1}{n}\sup_{f\in\mathcal{F}}\big|\sum_{i=1}^{n}|w_i|\varepsilon_i f(X_i)\big|\Big]$$

$$\overset{(iii)}{=} \sqrt{\frac{\pi}{2}}\mathbb{E}\Big[\frac{1}{n}\sup_{f\in\mathcal{F}}\big|\sum_{i=1}^{n}w_i f(X_i)\big|\Big],$$

where we use $\mathbb{E}_{\mathbf{w}}$ to denote taking expectation with respect to $w_1, ..., w_n$ and use a similar notation $\mathbb{E}_\varepsilon$ for taking expectation with respect to $\varepsilon_1, ..., \varepsilon_n$, (i) follows from $\mathbb{E}[|w_i|] = \sqrt{\frac{2}{\pi}}$, (ii) uses Jensen's inequality and (iii) is due to the fact that $w_i$ has the same distribution with $\varepsilon_i w_i$ for each $i$. This completes the proof of Lemma G.2. □

The following lemma can be found in Wainwright (2019), which allows us to utilize the symmetrization technique for the Lipschitz function family.

**Lemma G.3** (Gaussian contraction inequality). *For any set $\mathcal{T} \in \mathcal{R}^d$, and let $\{\phi_j : \mathcal{R} \to \mathcal{R}, j = 1, ..., d\}$ be any family of M-Lipschitz functions such that $\phi_j(0) = 0$ for $j \in [d]$, we have*

$$\mathbb{E}\left[\sup_{\theta\in\mathcal{T}}\Big|\sum_{j=1}^{d}w_j\phi_j(\theta_j)\Big|\right] \leq 2M\mathbb{E}\left[\sup_{\theta\in\mathcal{T}}\Big|\sum_{j=1}^{d}w_j\theta_j\Big|\right].$$

# H  Additional Simulations

## H.1  Assessing the Performance of the Truncated Kernel Method

In this section, we conduct a numerical investigation to assess the performance of the truncated kernel method (TKM) and validate our theoretical results in the main text. The experimental design consists of four distinct examples, including kernel quantile regression, kernel ridge regression, kernel support machine, and kernel logistic regression. In each example, we consider both the univariate and multi-dimensional cases. In specific, we apply Sobolev kernel $K(x, x') = \min\{x, x'\}$ for univariate cases and Laplacian kernel $K(\mathbf{x}, \mathbf{x}') = \exp(-\|\mathbf{x} - \mathbf{x}'\|_1)$ for multi-dimension cases. The elements of multi-dimensional covariates are independently sampled from the normal distribution, while in the univariate case, the covariate is sampled from the uniform distribution on $[0, 1]$. Aligned with the previous notation, TKM means truncated kernel-based method, and KM means standard kernel-based method. All the experiments are repeated 50 times and all the tuning parameters are tuned to the best for both methods. All experiments were conducted on the same hardware setup: Intel i9 13900K CPU @ 2.20GHz with 128 GB memory.

**Example 1** (Kernel quantile regression). In this illustrative example, we begin by conducting a comprehensive analysis of multivariate kernel quantile regression and postpone the univariate case to the subsequent discussion. We consider the data-generating scheme that $y = f_0(\mathbf{x}) + \sigma(\epsilon - \Phi^{-1}(\tau))$, where $\sigma = 3$, $\epsilon \sim N(0, 1)$ and $\Phi$ denotes CDF function of standard normal distribution. Here,

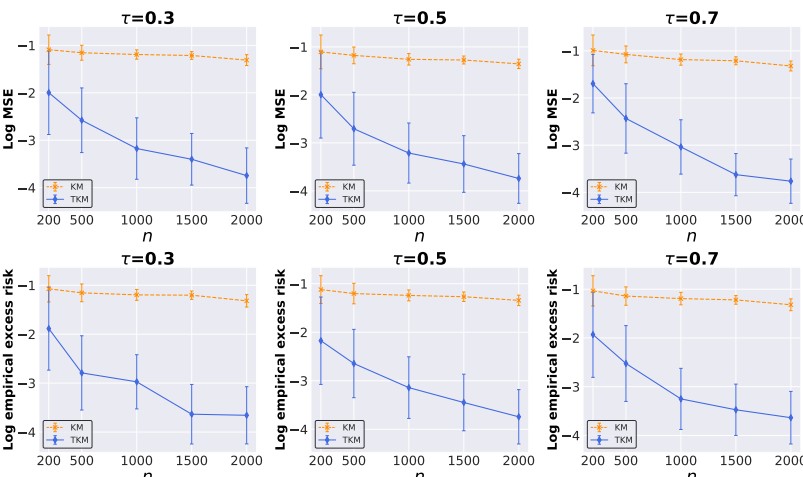

Figure 2: Averaged log MSE and log empirical excess risk for KM and TKM under check loss with varying sample size $n$.

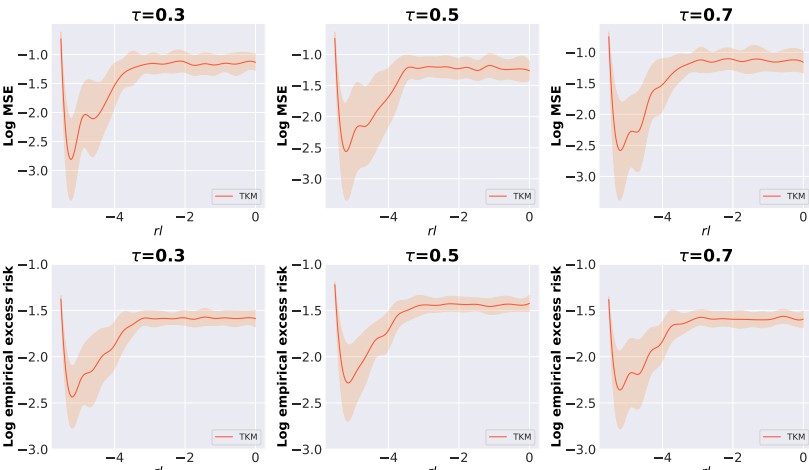

Figure 3: Averaged log MSE and log empirical excess risk for KM and TKM under check loss with varying truncation level $rl = \log(r/n)$.

we set $f_0(\mathbf{x}) = \sin 2(x_1 + x_2 + x_3)$ with $\mathbf{x} = (x_1, x_2, x_3)^\top$ and vary the quantile level $\tau$ from $\{0.3, 0.5, 0.7\}$.

We first compare the numerical performance between TKM and KM in estimating the true function $f_0$ under different sample sizes $n$. The averaged numerical results in terms of logarithmic mean square error (MSE) and empirical excess risk are illustrated in Figure 2.

It is clear that under different quantile levels, TKM always outperforms KM. More interestingly, the decline rate of TKM is significantly faster than that of KM, which validates our theoretical findings that under the over-aligned regime $\gamma > 1$, TKM achieves a faster learning rate than KM as illustrated in Corollary 4.3.

In the following study, we fix the sample size as $n = 500$ to investigate how the numerical performance of estimating $f_0(\mathbf{x})$ is affected by the truncation level $r$ by varying the logarithmic ratio of the truncation level $r$ to the sample size $n$, $rl = \log(r/n)$. The averaged numerical results in terms of logarithmic MSE and empirical excess risk are illustrated in Figure 3.

From Figure 3, we can see that the curves in all the scenarios have a steep decrease at first, then turn to a gradual increase, and finally become stabilizing with little vibration. This confirms our

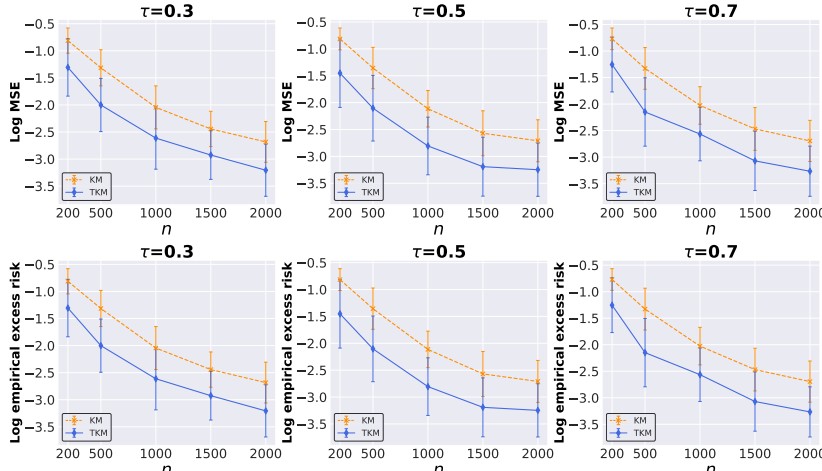

Figure 4: Averaged MSE and empirical excess risk for KQR and truncated KQR in the multivariate case.

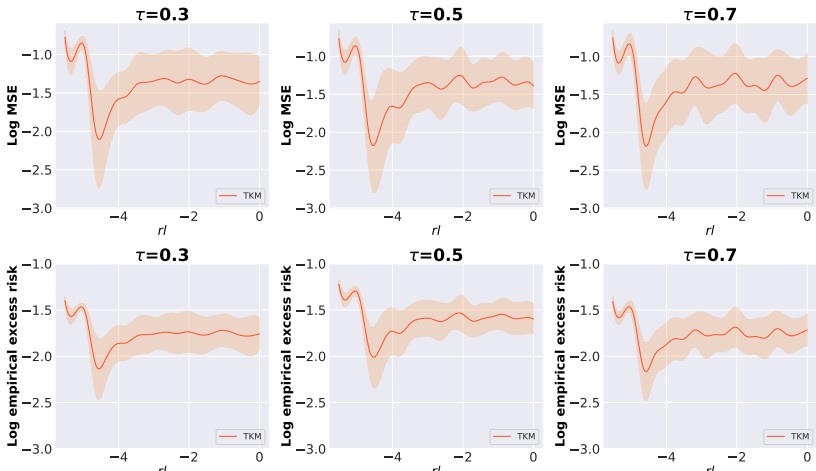

Figure 5: Averaged MSE and empirical excess risk vs $rl = \log(r/n)$; the sample size is set to $n = 500$.

theoretical findings on the truncation level $r$ and illustrates that a properly chosen truncation level is necessary to boost the estimation accuracy of the truncated kernel-based method.

Now we demonstrate univariate simulation for kernel quantile regression. We assume the true function to be $f_0(x) = \sin(10x)$ and underlying model to be $y = f_0(x) + \sigma(\epsilon - \Phi^{-1}(\tau))$, where $\sigma = 3, \epsilon \sim N(0,1)$ and $\Phi$ is CDF function of standard normal distribution.

As shown in Figures 4 and 5, for the univariate kernel quantile regression, we observe that, across different quantiles and various performance measures such as MSE or empirical excess risk, TKM consistently outperforms KM. This observation also confirms the advantages of TKM compared to KM.

**Example 2** (Kernel ridge regression). In the kernel ridge regression, we consider the model $y = f_0(x) + \epsilon$, where $f_0(x) = \sin(x), \epsilon \sim N(0,1)$ for univariate $x$. While for multi-dimension scenario, we assume dimension $p = 3$ and we generate data from $y = \sin(2(x_1 + x_2 + x_3)) + \epsilon$, where $\epsilon \sim N(0, 0.5)$. Another setting is similar to quantile regression.

As shown in Figure 6, it can be observed that in both the univariate and multivariate scenarios, TKM outperforms KM significantly. Furthermore, in multi-dimensional cases, the advantage of TKM is even more pronounced. From the right panel, it can be seen that although the optimal value of $r$

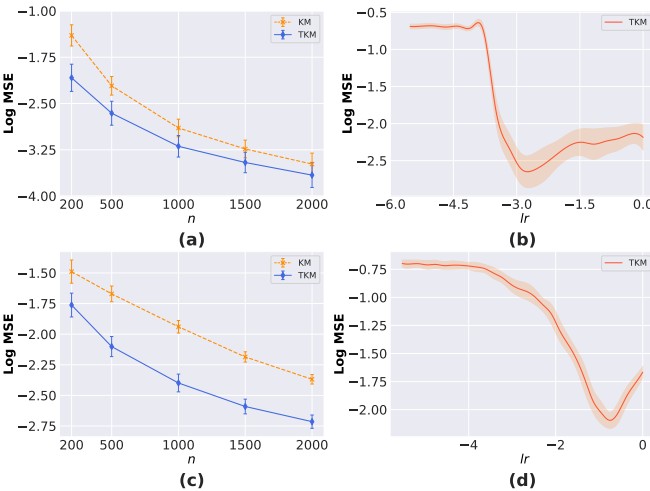

Figure 6: Simulation for kernel ridge regression. (a) univariate covariate case: average accuracy vs $n$. (b) univariate covariate case: average accuracy vs $rl = \log(r/n)$; the sample size is set to $n = 500$. (c) multivariate covariate case: average accuracy vs $n$. (d) multivariate covariate case: average accuracy vs $rl = \log(r/n)$; the sample size is set to $n = 500$.

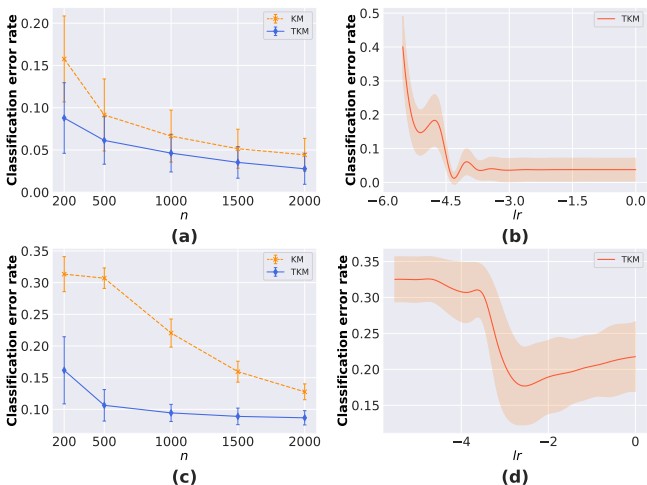

Figure 7: Simulation for kernel support vector machine. (a) univariate covariate case: average accuracy vs $n$. (b) univariate covariate case: average accuracy vs $rl = \log(r/n)$; the sample size is set to 200. (c) multivariate covariate case: average accuracy for vs $n$. (d) multivariate covariate case: average accuracy vs $rl = \log(r/n)$; the sample size is set to 200.

differs, they all reach their minimum at a certain point within the range of $[0, 1]$. This verifies our theoretical conclusion.

**Example 3** (Kernel support vector machine)**.** In the kernel support vector machine, we denote the sign of $x$ as $\text{sign}(x)$ and generate data through the model $y = \text{sign}(f_0(\mathbf{x}) + \epsilon)$. In univariate case $f_0(x) = \sin(10x)$ and for the multi-dimensional counterpart $f_0(\mathbf{x}) = 3\sin(x_1 + x_2 + x_3)$. $\epsilon \sim N(0, 1.5)$ in both case.

As shown in Figure 7, a similar trend can be observed in both methods, where TKM outperforms KM under different sample sizes. Furthermore, the advantage of TKM becomes more pronounced in multi-dimensional cases. As for the error curves against $rl$, it can be seen that both univariate and multivariate scenarios reach their minimum value within the range of $[0,1]$.

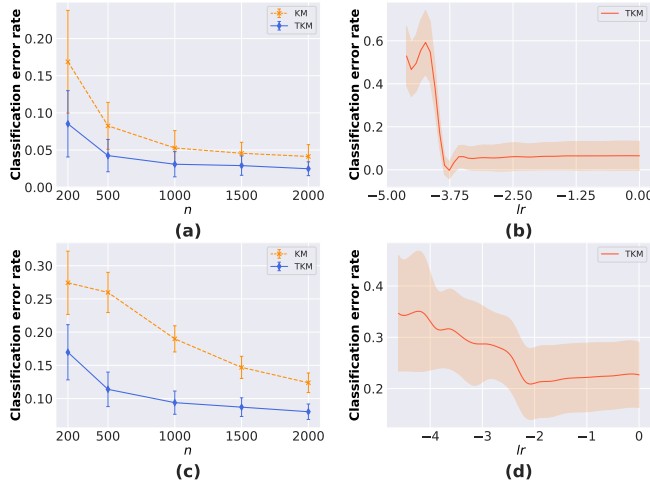

Figure 8: Simulation for kernel logistic regression. (a) univariate covariate case: average accuracy vs $n$. (b) univariate covariate case: average accuracy vs $rl = \log(r/n)$; the sample size is set to $n = 100$. (c) multivariate covariate case: average accuracy vs $n$. (d) multivariate covariate case: average accuracy vs $rl = \log(r/n)$; the sample size is set to $n = 100$.

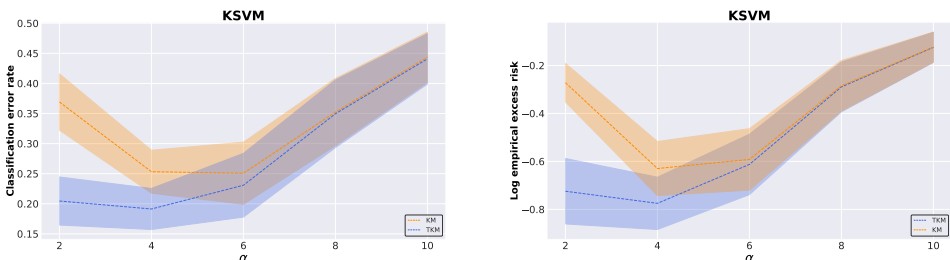

Figure 9: Kernel SVM; averaged classification error rate and log excess risk for KM and TKM versus $\alpha$.

**Example 4** (Kernel logistic regression). In kernel logistic regression, we generate data from $y \sim$ Bernoulli$(p)$, where $p = \frac{1}{1+\exp(-f_0(x))}$, where $f_0(x) = \sin(15x)$ in univariate case and $f_0(\mathbf{x}) = 3\sin(x_1 + x_2 + x_3)$ for the multi-dimensional case.

As shown in Figure 8, its exhibited curve trend is similar to that of kernel support machines. This validates that under different model assumptions, if a specific $r$ is chosen, TKM performs much better than KM.

## H.2   SVM with varying Model Complexities

In this part, we aim to investigate the problem how once the hinge loss is specified (corresponding to SVM), how the RKHS with varying model complexities affect the numerical performance of KM and TKM. Specifically, the experiment setup is the same as that in Section H.1, including the selection of kernel, repeat times, and tuning method for $\lambda$ and $r$ except that the underlying true function is set as $f^*(\mathbf{x}) = \sin(11\mathbf{x})$ and $(\mathbf{x}_i, y_i)_{i=1}^{300}$ is independently drawn from $y_i = \text{sign}(f^*(\mathbf{x}_i) + N(0, 4))$ with $\mathbf{x}_i = \frac{i-1}{300}, i = 1, \ldots, 300$. The obtained numerical results are reported in Figure 9. It is thus clear from Figure 9 that the error curves for the hinge loss align with those for the check loss, which further confirms our theoretical findings and also empirically supports that our theoretical analysis can apply to SVM.

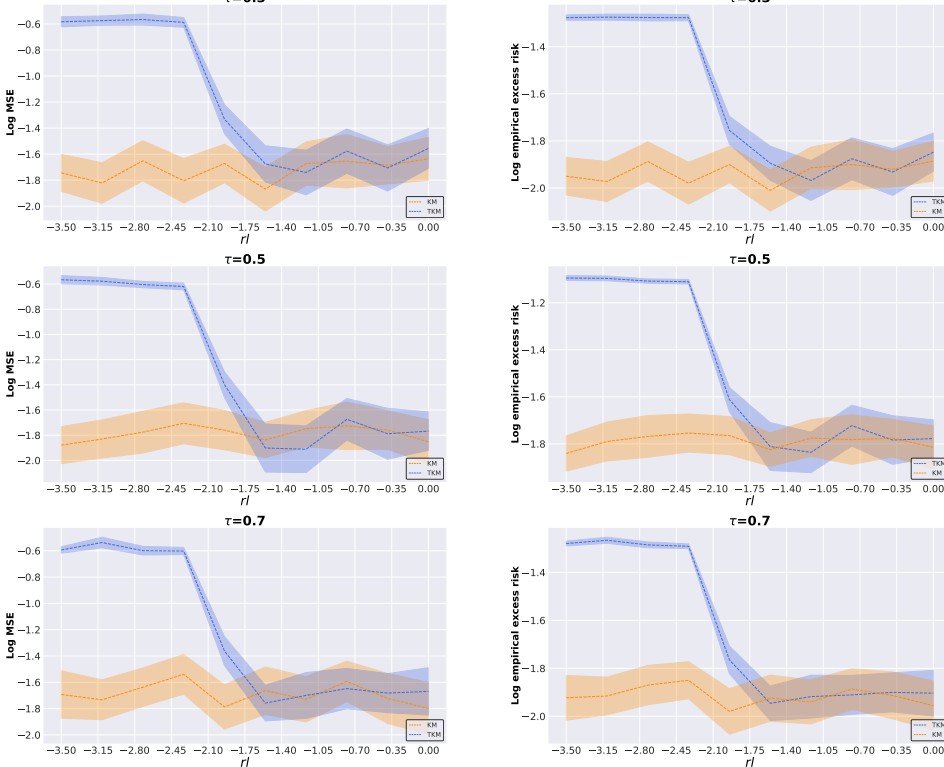

Figure 10: Kernel quantile regression; averaged log MSE and log empirical excess risk for KM and TKM versus log ratios ($rl = \log(r/n)$) of the truncation level $r$ to the sample size $n$ across different quantile levels.

### H.3 Exponential Decay Case

Note that our technical analysis can also cover the exponential decay case that $\mu_j \asymp \exp(-\alpha j)$ and $\xi_j^{*2} \asymp \exp(-(2\gamma\alpha + \beta)j)$ with $\alpha, \beta > 0$. Precisely, under the exponential decay setting, the explicit upper bound of the approximation bias term can be derived by

$$\sum_{j=r+1}^{n} \xi_j^{*2} \le C \int_r^{\infty} \exp(-(2\gamma\alpha + \beta)t)dt = \frac{C}{2\gamma\alpha + \beta} \exp(-(2\gamma\alpha + \beta)r).$$

Note that if $r \ge \frac{\log n}{(2\gamma\alpha+\beta)}$, we always have $\sum_{j=r+1}^{n} \xi_j^{*2} \lesssim \frac{1}{n}$. Consequently, we can also derive the corresponding convergence rates under these scenarios, which suggests that both TKM and KM can attain an optimal rate whatever $\gamma$ is if $r$ is greater than a certain threshold. We also conduct some numerical experiments to verify this finding.

Specifically, the experimental setup is the same as Example 1 in Section H.1 except that we set $f^*(\mathbf{x}) = \sin(6\mathbf{x})$ and the Gaussian kernel is used. The experiment result, presented in Figure 10, shows that TKM initially performs worse than KM for the small value of $r$. Whereas, as $r$ surpasses a threshold, TKM maintains comparable performance to KM. This observation precisely aligns with our theory for the exponential decay scenario.

### H.4 Determining $r$ via Cross-validation

Previously, all the tuning parameters were tuned to the best for both competitors. In this part, we will also provide the numerical experiment with the tuning parameters selected in a data-driven fashion. Specifically, we consider the kernel quantile regression that the data is independently generated from the model $y = f^*(\mathbf{x}) + \sqrt{2}(\varepsilon - \Phi^{-1}(\tau))$ with $f^*(\mathbf{x}) = \sin(6\mathbf{x})$, $\mathbf{x} = 0, \frac{1}{n}, \ldots, \frac{n-1}{n}$, and $\varepsilon \sim N(0, 1)$. In this experiment, we use the Laplacian kernel $K(\mathbf{x}, \mathbf{x}') = \exp(-\|\mathbf{x} - \mathbf{x}'\|_1)$, and

the parameters $r$ and $\lambda$ are tuned by 5-fold cross-validation. The obtained numerical results using the data-driven choice of $r$ are attached in the following tables. Clearly, it can be observed that TKM consistently outperforms KM, which further confirms our theoretical findings that TKM can achieve superior performance across various scenarios.

Table 3: Averaged MSE for different $n$ ($\tau = 0.3$).

| $n$ | 100 | 200 | 300 | 400 |
|---|---|---|---|---|
| KM | $0.583 \pm 0.257$ | $0.220 \pm 0.104$ | $0.165 \pm 0.071$ | $0.121 \pm 0.374$ |
| TKM | $0.367 \pm 0.174$ | $0.188 \pm 0.078$ | $0.140 \pm 0.004$ | $0.099 \pm 0.029$ |

Table 4: Averaged Empirical excess risk for different $n$ ($\tau = 0.3$).

| $n$ | 100 | 200 | 300 | 400 |
|---|---|---|---|---|
| KM | $0.323 \pm 0.039$ | $0.208 \pm 0.040$ | $0.175 \pm 0.036$ | $0.155 \pm 0.059$ |
| TKM | $0.289 \pm 0.066$ | $0.192 \pm 0.060$ | $0.161 \pm 0.021$ | $0.128 \pm 0.018$ |

Table 5: Averaged MSE for different $n$ ($\tau = 0.5$).

| $n$ | 100 | 200 | 300 | 400 |
|---|---|---|---|---|
| KM | $0.246 \pm 0.137$ | $0.177 \pm 0.129$ | $0.096 \pm 0.032$ | $0.114 \pm 0.069$ |
| TKM | $0.195 \pm 0.087$ | $0.153 \pm 0.133$ | $0.075 \pm 0.033$ | $0.079 \pm 0.042$ |

Table 6: Averaged Empirical excess risk for different $n$ ($\tau = 0.5$).

| $n$ | 100 | 200 | 300 | 400 |
|---|---|---|---|---|
| KM | $0.214 \pm 0.062$ | $0.189 \pm 0.052$ | $0.176 \pm 0.047$ | $0.140 \pm 0.028$ |
| TKM | $0.168 \pm 0.039$ | $0.146 \pm 0.048$ | $0.158 \pm 0.055$ | $0.123 \pm 0.027$ |

Table 7: Averaged MSE for different $n$ ($\tau = 0.7$).

| $n$ | 100 | 200 | 300 | 400 |
|---|---|---|---|---|
| KM | $0.434 \pm 0.272$ | $0.273 \pm 0.099$ | $0.163 \pm 0.086$ | $0.121 \pm 0.072$ |
| TKM | $0.325 \pm 0.195$ | $0.200 \pm 0.079$ | $0.134 \pm 0.072$ | $0.098 \pm 0.054$ |

Table 8: Averaged Empirical excess risk for different $n$ ($\tau = 0.7$).

| $n$ | 100 | 200 | 300 | 400 |
|---|---|---|---|---|
| KM | $0.178 \pm 0.053$ | $0.201 \pm 0.050$ | $0.145 \pm 0.032$ | $0.119 \pm 0.026$ |
| TKM | $0.159 \pm 0.053$ | $0.155 \pm 0.053$ | $0.122 \pm 0.018$ | $0.106 \pm 0.023$ |

