# OpenReview forum: "On the Target-kernel Alignment: a Unified Analysis with Kernel Complexity"
_NeurIPS.cc/2024/Conference — NeurIPS 2024 poster_

### Official Review · Reviewer_ikTW · 2024-07-03

**Soundness:** 4
**Presentation:** 4
**Contribution:** 3
**Rating:** 7
**Confidence:** 4

**Summary:**

This paper provides an in-depth error analysis of of kernel ridge regression (KRR) and truncated KRR (TKRR), where one replace the original kernel $K$ with its finite $r$-dimensional approximation $K^T$, such that their regressors $\hat{f}$ and $\hat{f}_r$ agree on the dataset. It is well known that KRR suffers from the so-called *saturation effect*, where increasing the smoothness/alignment of the target function cannot further improve the generalization performance after a certain threshold. TKRR, on the other hand, with correctly selected hyperparameter $r$, can continuously improve the generalization performance as the target function is more and more aligned with the reproducing kernel Hilbert space (RHKS) $\mathcal{H}$ of the kernel $K$.

**Strengths:**

This paper offers a very general theoretical result on kernel performance with various losses, in contrast to the square loss considered in much of the previous literature. Additionally, this paper works under realistic assumptions and presents numerical validations to support its claims. The TKRR method presented in [Amini2021] and this paper offer a simple way to bypass the saturation effect of KRR, which might have significant applications in many realistic settings.

Reference:
Amini, A. A. (2021). Spectrally-truncated kernel ridge regression and its free lunch. Electronic Journal of Statistics, 15, 3743–3761.

**Weaknesses:**

No major weakness is spotted in this paper. However, as a new algorithm, I think that there should be more discussion on the computational complexity TKRR, which is lacking in both [Amini201] and this paper.

**Questions:**

I have several remark/questions regarding the algorithmic perspective on TKRR:

1. Let $\mathbf{K} = \mathbf{U}^\top\mathbf{D}\mathbf{U}$ be the diagonalization of the kernel matrix w.r.t. inputs $X = \\{\mathbf{x}\_i\\}\_{i=1}^n$, and $ S_X : \mathcal{H}\_K \to \mathbb{R}\^n $ be the evaluation operator $f \mapsto (f(\mathbf{x}\_1),...,f(\mathbf{x}\_n))^\top$. By line 239 in the paper (there is a small typo in the definition of $\psi_k$, see below for the correct one), the finite dimensional RKHS $\tilde{\mathcal{H}}\subset \mathcal{H}$ has the basis $\\{ \psi_k \\}\_{k=1}^r$ where $r\leq n$ and $\psi_k := \text{argmin} \\{ \\| \psi \\|_K:\psi\in\mathcal{H}, S_X(\psi) = \mathbf{u}_k \\}$ with $\mathbf{u}_k$ the $k$-th column of the matrix $\mathbf{U}$. This definition of $\psi_k$ is simply the kernel interpolant: $\psi_k(x) = \mathbf{u}^\top\mathbf{K}^{-1}\mathbf{K}_x$. Hence the computational bottleneck is the inversion of $\mathbf{K}$ with complexity $\mathcal{O}(n^3)$, and thus the total complexity of TKRR should not be much larger than KRR. Although it is not difficult to see, I think it is still worth a small section/paragraph to discuss this.

2. The choice of the hyperparameter $r$ is important when the alignment parameter $\gamma$ is at least $1/2$. But how can one determine the optimal choice of $r$ if $\gamma$ is not directly accessible?

**Limitations:**

This is a theoretical paper. All limitations are stated clearly in the statements.

---

> ### Author Rebuttal · Authors · 2024-08-07
>
> We appreciate your acknowledgment and positive feedback on this work. Your valuable suggestions and comments are very helpful and significantly improve this paper. Below are our point-to-point replies.
>
> **Weakness 1 \&  Question 1 (Part I): More discussion on the computational complexity of TKRR.**
>
> **Answer:**  Thank you very much for your precious suggestions on the computational complexity.  The total computational complexity of the truncated kernel method (TKM) is composed of three parts.   In the first part, spectrally decomposing the kernel matrix $\mathbf{K}$ has the computational complexity of $\mathcal{O}(n^3)$. In the second part, as pointed out by you, the basis $(\psi _k) _{1\le k \le r}$  can be simply calculated by  $\psi _k(\mathbf{x})= \mathbf{u}_k^T\mathbf{K} ^{-1} \mathbf{K} _{\mathbf{x}}$,
> which also has  $\mathcal{O}(n^3)$ computational complexity. In the last part, deriving the kernel estimator based on the $r$-dimensional RKHS $\widetilde{\mathcal{H}}$ has computational complexity of  $\mathcal{O}(nr^2)$.  To sum up,  the overall computational complexity of TKM is $\mathcal{O}(n^3)$ and clearly, the truncated method does not impose an additional computational cost compared to the standard kernel method. Following your suggestion, detailed discussions and comparisons on the computational complexities of  TKM and the standard kernel method have been added in Section 5 of the revised manuscript.
>
> **Question 1 (Part II): There is a small typo in the definition of  $\psi_k$.**
>
> **Answer:**
> Thank you very much for pointing out this typo and correcting it. In the revised version, this typo has been fixed and we have carefully proofread the manuscript again and tried our best to correct all the typos.
>
> **Question 2: How can one determine the optimal choice of $r$ if $\gamma$ is not directly accessible?**
>
> **Answer:** Thank you very much for your question.  Indeed, the choice of the hyperparameter $r$ is crucial for the theoretical results of TKM that it balances the estimation error and approximation bias as discussed after Theorem 4.2, and the optimal theoretical choice of $r$ depends on the alignment parameter  $\gamma$ under the setting $\gamma\ge 1/2$.  Note that a similar analysis framework is also considered in  Amini et al. (2022). In practice, since the underlying parameter $\gamma$ is unknown, some data-driven strategies, such as the cross-validation procedure, can be used to determine the possible optimal choice of $r$. In our numerical example in Section 6,   all the tuning parameters are tuned to the best for both competitors and in this revision, we will also add the numerical experiments with the tuning parameters selected in the data-driven fashion, and part of the numerical results are provided below. Specifically, we consider the kernel quantile regression that the data is independently generated from the model $y=f^*(\mathbf{x})+\sqrt{2}(\varepsilon-\Phi^{-1}(\tau))$ with $f^*(\mathbf{x})=\sin(6\mathbf{x})$, $\mathbf{x}=\{0, \frac{1}{n}, \dots, \frac{n-1}{n}\}$, and $\varepsilon\sim N(0, 1)$. In this experiment, we use the Laplacian kernel  $K(\mathbf{x}, {\mathbf{x}}')=\exp(-||\mathbf{x}-{\mathbf{x}}'||_1)$, and the parameters $r$ and $\lambda$ are tuned by  $5$-fold cross-validation.
> Clearly, it can be observed from the obtained numerical results, attached in the following tables,  that using the data-driven choice of $r$, TKM consistently outperforms KM, which further confirms our theoretical findings that TKM can achieve superior performance across various scenarios.  In the revised version, some additional numerical experiments will be added in Appendix and more detailed discussions on this issue will be provided in Section 6 and Appendix.
>
>
>
> **Table 1： Averaged MSE for different $n$ ($\tau=0.3$).**
> | $n$  | $100$           | $200$           | $300$           | $400$           |
> |------|-----------------|-----------------|-----------------|-----------------|
> | KM   | $0.583 \pm 0.257$ | $0.220 \pm 0.104$ | $0.165 \pm 0.071$ | $0.121 \pm 0.374$ |
> | TKM  | $0.367 \pm 0.174$ | $0.188 \pm 0.078$ | $0.140 \pm 0.004$ | $0.099 \pm 0.029$ |
>
> **Table 2： Averaged empirical excess risk for different $n$ ($\tau=0.3$).**
> | $n$   | $100$            | $200$            | $300$            | $400$            |
> |-------|------------------|------------------|------------------|------------------|
> | KM    | $0.323 \pm 0.039$ | $0.208 \pm 0.040$ | $0.175 \pm 0.036$ | $0.155 \pm 0.059$ |
> |       |                  |                  |                  |                  |
> | TKM   | $0.289 \pm 0.066$ | $0.192 \pm 0.060$ | $0.161 \pm 0.021$ | $0.128 \pm 0.018$ |
> |       |                  |                  |                  |                  |

---

> > ### Author Response · Authors · 2024-08-07
> > **Reference**
> >
> > We apologize for any confusion regarding the references cited in our response during the rebuttal phase. Below, we have provided the complete reference information that you may need.
> >
> > [1]  Amini, A., Baumgartner, R., & Feng, D. (2022). Target alignment in truncated kernel ridge regression. *In Advances in Neural Information Processing Systems* (pp. 21948–21960). Curran Associates, Inc. volume 35.

---

> > ### Comment · Reviewer_ikTW · 2024-08-09
> >
> > Thank you for your response and your prompt experimental results. I would keep my score and lean to accept this paper.

---

> > > ### Author Response · Authors · 2024-08-09
> > > **Appreciation for Reviewer ikTW**
> > >
> > > We sincerely appreciate your feedback and the positive evaluation of our paper. Thank you once again for the insightful comments during the review, which greatly contributed to the improvement of our work.

---

### Official Review · Reviewer_Twpu · 2024-07-07

**Soundness:** 4
**Presentation:** 4
**Contribution:** 4
**Rating:** 7
**Confidence:** 3

**Summary:**

This paper investigated the impact of alignment between the target function of interest and kernel matrices.
To overcome the saturation effect, the TKM was introduced and its learning rate is analyzed.

**Strengths:**

This paper is well-written and the theorems are solid.

This paper analyzed different alignment level and the learning rate in these cases.
TKM decreases the kernel complexity to achieve the trade-off between model complexity and approximation error.

**Weaknesses:**

The approximation bias term in Thm 4.2 requires Assumption 3.4 holds. Could you also provide some results for different decay rate?

**Questions:**

None.

**Limitations:**

None.

---

> ### Author Rebuttal · Authors · 2024-08-07
>
> We appreciate your time in reviewing our work and thanks a lot for your question and valuable comments. We have carefully addressed your question below and provided some results for the exponential decay rate.
>
> Indeed, Assumption 3.4 is needed if we want to derive the explicit upper bound of the approximation bias term established in Theorem 4.2. Note that our technical analysis can also be modified to consider the exponential decay case that $\widehat{ \mu} _j  \asymp \exp(-\alpha j)$ and ${\xi_j^*}^2 \asymp \exp(-(2\gamma \alpha + \beta) j)$ with $\alpha, \beta > 0$.
>
> Precisely, under the  exponential decay setting, the explicit upper bound of the approximation bias term can be derived by
>
> $$
> \sum _{j=r+1}^n{\xi^* _j}^2 \le C \int _r ^\infty \exp(-(2\gamma \alpha + \beta) t) dt = \frac{C}{2\gamma \alpha + \beta } \exp(-(2\gamma \alpha + \beta) r).
> $$
>
> Note that if $r\ge
> \frac{\log n }{(2\gamma \alpha + \beta)}$, there holds $\sum_{j=r+1}^n{\xi^*_j}^2\lesssim \frac{1}{n}$. Consequently, we can also derive the corresponding convergence rates under these scenarios, which suggests that both TKM and KM can attain an optimal rate whatever $\gamma$ is if $r$ is greater than a certain threshold (to be added in the revised version). We also conduct some numerical experiments to verify this finding.
> Specifically,
> the experimental setup is the same as Example 1 in Appendix G of the manuscript except that we set $f^*(\mathbf{x})=\sin(6\mathbf{x}) $ and  Gaussian kernel is used. The experiment result, presented in Figure 2  in the total Rebuttal pdf file, shows that  TKM initially performs worse than KM for very small value of $r$. Whereas,  as $r$ surpasses a threshold, TKM maintains comparable performance to KM.
>
> This observation precisely aligns with our theory for the exponential decay scenario.
> In this revision, more detailed results and discussions of this setting will be added in Section 4.1 and Appendix.

---

> > ### Comment · Reviewer_Twpu · 2024-08-08
> >
> > Thank you for your reply. My concern is addressed, and I will raise my score.

---

> > > ### Author Response · Authors · 2024-08-08
> > > **Appreciation for Reviewer Twpu**
> > >
> > > Thank you very much for your response and raising the score!  We are pleased to hear that your concern has been addressed.
> > > Once again, thank you for the positive evaluation of our paper and the constructive comments, which are greatly helpful to us.

---

### Official Review · Reviewer_4evc · 2024-07-07

**Soundness:** 3
**Presentation:** 3
**Contribution:** 2
**Rating:** 5
**Confidence:** 3

**Summary:**

This paper conducts a comprehensive theoretical analysis on the learning rate of kernel-based machine learning methods under a general setting. It establishes the upper bounds for standard kernel-based estimators, and demonstrates that standard kernel-based estimators suffer from saturation effect at high target-kernel alignment levels. It proves that the learning rate of truncated kernel-based estimators, where kernel matrices are spectrally truncated, continues to improve even at high alignment levels, indicating an improvement over standard estimators. It establishes minimax lower bound for both standard and truncated kernel-based estimators, and demonstrates that the standard kernel-based estimator can only attain suboptimality for the strong-aligned regime, while the truncated estimator is minimax-optimal. The author conducts various experiments on regression and classification problems to demonstrate the advantages of the truncated estimator and support the established theory.

**Strengths:**

1) The paper reveals that the saturation effect, i.e. the phenomenon that the learning rate of kernel ridge regression (KRR) no longer improves at high target-kernel alignment levels, also occurs in general kernel-based methods other than KRR. This discovery extends the original focus on KRR’s learning rate to general kernel-based methods, and may spark more interest and research in this direction.
2) The paper demonstrates theoretically and experimentally that truncated kernel-based method (TKM) can overcome the saturation effect in a general loss-function setting. This indicates that the saturation effect can be addressed not only in KRR, but also in general kernel-based methods. It may provide insights for future research on tackling saturation effect and improve learning rate of kernel methods.
3) The theoretical analysis is comprehensive and rigorous. Sufficient details are provided to help understand and verify the paper.

**Weaknesses:**

1) The truncated kernel-based method (TKM) considered in this paper is not a very novel approach. As mentioned in the paper, truncated KRR (Amini, 2021; Amini et al., 2022) also uses spectral-truncated kernel matrix as the underlying method; the main difference between TKM and truncated KRR is that TKM considers a more general loss function family, while truncated KRR specifies a squared loss. In other words, TKM is basically a generalized version of truncated KRR; hence, the methodological contribution of this paper is limited and largely incremental.
2) The work in this paper only considers Lipschitz continuous loss functions. It is a reasonable setting, but there do exist some non-Lipschitz continuous loss functions in practice, e.g. 0-1 loss. The paper doesn’t take such condition into account; this limits its scope of application.
3) The use of symbols and notations in some formulas is somewhat confusing. For example, in Section 1, the symbol r is used to denote target-kernel alignment level, but in Sections 3 and 4 the authors denote target-kernel alignment as gamma, and r becomes the dimensionality of reduced RKHS. It would be better if the authors could make their use of symbols and notations consistent throughout the paper.

**Questions:**

1) If the loss function is not Lipschitz continuous (e.g. 0-1 loss), would any of the results presented in the paper change? For instance, would the learning rate bounds for standard kernel-based estimators established in Section 3 remain the same?
2) Are the choices of r and lambda mentioned in Section 4 the only feasible choices to achieve optimal learning rate for TKM?

**Limitations:**

Yes.

---

> ### Author Rebuttal · Authors · 2024-08-07
>
> Your insightful comments and constructive suggestions are highly valued by us, and greatly contributed to the revision of this work. Below are our point-to-point replies.
>
> **Weakness 1:  TKM considered in this paper is not a very novel approach.**
>
> **Answer:** Thank you very much for your comment.  We agree that TKM considered in this paper is not a very novel approach and we admit that this paper is motivated by the prior work   (Amini et al., 2022)  where only the squared loss function is considered and the investigation on the problem of optimality is lacking.  In sharp contrast, our theoretical results are established for a general loss function by using totally different technical treatments,  which covers many commonly used methods in regression and classification problems, and a minimax lower bound is also established for the squared loss function, which confirms the optimality of TKM.
>
> We want to emphasize that some significant gaps exist between our work and the prior work  (Amini et al., 2022) and we list some of them below.    Amini et al. (2022) only considers the square loss function where the analytical solution exists, and thus their theoretical analysis heavily relies on the closed form of the solution to establish some critical results, including the theoretical bounds similar to Corollaries 3.5 and 4.3 in our paper. As opposed, our work considers a general loss function and such an explicit solution does not exist anymore, which requires different technical treatments to derive the theoretical results.  Specifically, our theoretical analysis adopts an alternative analytic treatment by utilizing kernel complexity and deriving upper bounds associated with the critical radius. Moreover, we rigorously establish the minimax lower bound when the squared loss is specified, which is unsolved in  Amini et al. (2022) and confirms the optimality of the truncated kernel method. Extensive numerical studies are also provided in our paper, which further supports our theoretical findings and verifies the existence of the trade-off between the target-kernel alignment and model complexity.  This also highlights our contributions.
>
> In this revision, various parts have been revised and more detailed discussions have been added to highlight the contributions of this paper, including Sections 1.1, 4 and 5.
>
> **Weakness 2 \& Question 1: Non-Lipschitz continuous loss functions, e.g. 0-1 loss.**
>
> **Answer:** Thank you very much for your comment. It is true that the established results in this paper require the Lipschitz continuity assumption, and can not be directly applied to the non-Lipschitz continuous cases. Yet, we want to emphasize that this type of Lipschitz continuity assumption is commonly considered in the literature of machine learning  (Steinwart & Christmann, 2008; Wei et al., 2017; Li et al., 2019; Farrell et al., 2021),  which covers many commonly used loss functions as illustrated in Table 1 of the manuscript.
>
> When the loss function is non-Lipschitz continuous or even not continuous, extending our established results to the general non-Lipschitz continuous setting is not trivial due to the technical difficulties, such as the application of Talagrand’s concentration inequality requires the Lipschitz continuity of loss function. However, once the $0$-$1$ loss is considered, one possible routine for establishing the theoretical results for $0$-$1$ loss is to follow a similar technical treatment as that on Page 17 of Lai et al. (2024) with some slight modifications, where the bridge between the excess risk w.r.t $0$-$1$ loss and mean squared error is established, and based on the result in Lai et al. (2024), the excess risk only gets a slower rate compared to the rates established in our paper.
>
> It is also interesting to point out that optimizing the $0$-$1$ loss is very difficult. In literature, it is common to replace it with a convex surrogate loss function, such as the logistic loss function, which is covered by our paper. We will leave this very interesting problem for future work, but add more detailed discussions on possible routines for the extension to the non-Lipschitz continuous case in the revised manuscript.
>
> **Weakness 3: Confusion of notation and symbols in some formulas.**
>
> **Answer:** Thank you very much for pointing out this problem.  We apologize for this abuse of notation. In this revision, this problem has been corrected that we consistently use $r$ to denote the dimensionality of the reduced RKHS, and use $\gamma$ to denote the target-kernel alignment level. Moreover, we have proofread this manuscript again and made our best efforts to correct all the typos, confusing notations and symbols.
>
> **Question 2: The choices of $r$ and $\lambda$.**
>
> **Answer:** Thank you very much for your question. Indeed, the theoretical orders of $r$ and $\lambda$ mentioned behind Theorem 4.2 and in Corollary 4.3 of Section 4 are the only feasible choices to achieve optimal learning rate for TKM.  Specifically, under Assumption 3.4, the optimal choice of $\lambda$ is unique that $\lambda\asymp \delta_{n,r}^2 \asymp \big(\frac{(\log \iota^{-1})^2}{n}\big)^{\frac{\max(\gamma,1) \alpha}{2\gamma\alpha+1}}$ to balance the variance and bias, and the unique optimal choice of $r$ is $ r\asymp (\frac{n}{(\log \iota^{-1})^2})^{\frac{1}{2\gamma\alpha+1}}\text{I} _{(\gamma> 1)}+n\text{I} _{(\frac{1}{2}\le \gamma\le 1)}$ to balance the estimation error and approximation bias.  We want to emphasize that in literature, it is common to require the explicit unique orders of the parameters to achieve the fast learning rate (Yang et al., 2017; Wei et al., 2017; Cui et al., 2021; Amini et al., 2022). In practice, the values of $r$ and $\lambda$ can be determined by using some data-driven procedures, such as the cross-validation technique. More detailed discussions on the choices of the two parameters will be added in Section 4 of the revised manuscript.

---

> > ### Author Response · Authors · 2024-08-07
> > **Reference**
> >
> > We apologize for any confusion regarding the references cited in our response during the rebuttal phase. Below, we have provided the complete reference information that you may need.
> >
> > [1]  Amini, A., Baumgartner, R., & Feng, D. (2022). Target alignment in truncated kernel ridge regression. *In Advances in Neural Information Processing Systems* (pp. 21948–21960). Curran Associates, Inc. volume 35.
> >
> > [2]  Steinwart, I., & Christmann, A. (2008). *Support Vector Machines.*  Springer Science & Business Media.
> >
> > [3]  Wei, Y., Yang, F., & Wainwright, M. J. (2017). Early stopping for kernel boosting algorithms: A general analysis with localized complexities. *Advances in Neural Information Processing Systems,* 30.
> >
> > [4]  Li, Z., Ton, J.-F., Oglic, D., & Sejdinovic, D. (2019). Towards a unified analysis of random Fourier features. *In International Conference on Machine Learning* (pp. 3905–3914). PMLR.
> >
> > [5] Farrell, M. H., Liang, T., & Misra, S. (2021). Deep neural networks for estimation and inference. *Econometrica,* 89, 181–213.
> >
> > [6] Lai, J., Huang, D., Lin, Q. et al. (2024). The optimality of kernel classifiers in Sobolev space. *In The Twelfth International Conference on Learning Representations.*
> >
> > [7] Yang, Y., Pilanci, M., & Wainwright, M. J. (2017). Randomized sketches for kernels: Fast and optimal nonparametric regression. *Annals of Statistics,* 45, 991–1023. 14.
> >
> > [8] Cui, H., Loureiro, B., Krzakala, F., & Zdeborová, L. (2021). Generalization error rates in kernel regression: The crossover from the noiseless to noisy regime. *Advances in Neural Information Processing Systems,* 34, 10131–10143.

---

### Official Review · Reviewer_Pn9U · 2024-07-09

**Soundness:** 3
**Presentation:** 3
**Contribution:** 3
**Rating:** 6
**Confidence:** 2

**Summary:**

In this paper the authors consider both truncated kernel-based method (TKM) and standard kernel-base method (KM) and how their performance is affected by the target-kernel alignment (a.k.a. the smoothness of target function in RKHS). The authors show that they have the same effects in weak and just-aligned regime, but TKM is able to tackle the saturation effect in the strongly-aligned regime since it can attain the minimax rate. The theoretical analysis of this paper includes a general class of loss function with Lipschitz continuity assumption.

**Strengths:**

- The paper characterizes the kernel complexity using statistical dimension and uses empirical process technique to derive the result
- The paper discusses an interesting finding that TKM can overcome saturation effect
- Can be any loss function with Lipschitz continuity assumption

**Weaknesses:**

- Some of the findings are also covered in this paper [1]

[1] Arash A. Amini et al., Target alignment in truncated kernel ridge regression

**Questions:**

- As in the paper it is demonstrated that spectrally truncated kernel can overcome the saturation effect, can spectrally transform kernel [1] overcome it too? or improve the minimax rate?

[1] Runtian Zhai et al. , Spectrally Transformed Kernel Regression

**Limitations:**

See weakness

---

> ### Author Rebuttal · Authors · 2024-08-07
>
> Thank you very much for your nice summary and precious comments on this paper. Below are our point-to-point replies.
>
> **Weakness 1: Some of the findings are also covered in  Amini et al. (2022).**
>
> **Answer:**
> Thank you very much for your comment. We agree with you that some findings in this paper contain the results provided in  Amini et al. (2022), where only the squared loss function is considered. In fact, this work is motivated by  Amini et al. (2022)  and part of our established results can be regarded as the extension of those in Amini et al. (2022), where a board of learning tasks is considered in our paper.  We want to emphasize that there exist some significant differences between our work and the prior work (Amini et al., 2022)  from multiple points of view.
>
> Note that  Amini et al. (2022) only considers the square loss function where the analytical solution exists, and thus their theoretical analysis heavily relies on the closed form of the solution to establish some critical results, including the theoretical bounds similar to Corollaries 3.5 and 4.3 in our paper. As opposed, our work considers a general loss function which covers many commonly
> used methods in regression and classification problems, and such explicit solution does not exist anymore, which requires different technical treatments to derive the theoretical results. Specifically, our theoretical analysis adopts an alternative analytic treatment by utilizing kernel complexity and deriving upper bounds associated with the critical radius. Our results successfully capture the trade-off between the model complexity of the truncated RKHS $\widetilde{\mathcal{H}}$ and approximation bias as presented in Theorem 4.2.  Moreover, we also establish the minimax lower bound when the squared loss is specified, and thus rigorously confirm the conjecture in Amini et al. (2022) stating that the truncated kernel ridge estimator attains minimax optimality. Numerically, extensive experiments are also conducted to confirm our theoretical findings and verify the existence of the trade-off between the target-kernel alignment and model complexity, which also highlights our contributions.  In the revised version,  more detailed discussions and comparisons on the differences between our work and the previous work (Amini et al., 2022) will be added in various parts, including Sections 1.1 and 5.
>
>
> **Question 1:  can spectrally transform kernel  (Zhai et al.,  2024)  overcome it too? or improve the minimax rate?**
>
> **Answer:**
>  Thank you very much for your comment and for bringing us this interesting reference.  We notice that the spectrally transformed kernel regression method (SKRR;  Zhai et al.,  2024) aims to use spectrally transformation for constructing a new kernel, which can leverage the information contained in unlabeled data in an explicit way.
>
> We believe that  SKRR may have the ability to overcome the saturation effect as well if the transformation function can be properly chosen.  The possible routine for establishing the theoretical results is discussed below. Recall that by Mercer's theorem, the kernel function admits a decomposition as  $K(\mathbf{x}, \mathbf{x}')= \sum_{ j=1}^\infty\mu_j\phi_j(\mathbf{x}) \phi_j({\mathbf{x}}'),$
> where $\mu_j$'s are the eigenvalues in descending ordering and $\phi_j$'s are the corresponding eigenfunctions of the integral operator (Zhai et al.,  2024), and we can write the target function $f^*$ as $f^* =  \sum_{ j=1}^\infty \alpha^*_j  \phi_j$.
>
> For  SKRR, $K( \mathbf{x}, {\mathbf{x}}')$ is replaced with a new kernel that   $K'( \mathbf{x}, {\mathbf{x}}') = \sum _{j=1}^\infty s(\mu _j) \phi _j ( \mathbf{x})\phi _j ({\mathbf{x}}')$, where  $s$ is the general transformation function.
> The idea of deriving the upper bound is that we can separately bound the estimation error $|| \widehat{f} _\lambda- f^\sharp|| _\mu$ and approximation bias $|| f^{\sharp}- f^*|| _\mu$,  where $|| \cdot || _\mu$ denotes the norm equipped with  $ \mathcal{L}(\mathcal{X}, \mu)$, and  $f^ \sharp=\sum _{j=1}^\infty s(\alpha^* _j) \phi _j$ denotes an immediate function belonging to the RKHS induced by $K'$.
> Following a similar technical treatment in our paper,  the upper bound on estimation error can be established. For the approximation bias,  we notice that
>
> $$
> ||{f}^{\sharp}_r-f^*|| _\mu^2= \sum _{j=1}^\infty  ( s(\alpha _j^*)- \alpha _j^*)^2.
> $$
>
> Clearly, the selection of $s(\cdot)$ is crucial and it is favorable if $s(\cdot)$ is close to the identity function for small $j$ and decays extremely rapidly as $j$ tends to infinity, such as $s(\mu_j)=\mu_j\mathbf{I}_{(j\le r)}$. Then, SKRR with some proper choices of $s(\cdot)$ may achieve similar conclusions about the upper bound as we provided in our paper.  Moreover, we think the minimax lower bound can not be improved since it is independent of any specific learning algorithm developed.
>
> Since the theoretical derivation should be more involved, we decide to leave such a promising topic as potential future work, but add some detailed discussions on the possible route for establishing the theoretical results of SKRR in Appendix A of the revised version.

---

> > ### Author Response · Authors · 2024-08-07
> > **Reference**
> >
> > We apologize for any confusion regarding the references cited in our response during the rebuttal phase. Below, we have provided the complete reference information that you may need.
> >
> > [1]  Amini, A., Baumgartner, R., & Feng, D. (2022). Target alignment in truncated kernel ridge regression. *In Advances in Neural Information Processing Systems* (pp. 21948–21960). Curran Associates, Inc. volume 35.
> >
> > [2]  Zhai, R., Pukdee, R., Jin, R., Balcan, M. F., & Ravikumar, P. K. (2024). Spectrally transformed kernel regression. *In The Twelfth International Conference on Learning Representations.*

---

> > ### Comment · Reviewer_Pn9U · 2024-08-09
> >
> > Thank you for your detailed response! And I will keep my already positive score

---

> > > ### Author Response · Authors · 2024-08-10
> > > **Appreciation for Reviewer Pn9U**
> > >
> > > Thank you for your feedback. We sincerely appreciate your time and effort in reviewing our work and the valuable comments during the review.

---

### Official Review · Reviewer_rK2F · 2024-07-09

**Soundness:** 3
**Presentation:** 3
**Contribution:** 3
**Rating:** 6
**Confidence:** 2

**Summary:**

This paper investigates the impact of target-kernel alignment to mitigate the saturation effect, where the learning rate of kernel ridge regression plateaus when the smoothness of the target function exceeds certain levels. The kernel complexity function is used to establish the upper bounds for both the standard kernel-based estimator and the truncated estimator. Also, the Fano method is employed to establish minimax lower bound when the squared loss is utilized.

**Strengths:**

- The paper is well-written and provides a comprehensive review of related work.
- Detailed theoretical results are given for standard and truncated kernel-based methods.
- Confirming theoretical results by using numerical simulations.

**Weaknesses:**

N/A

**Questions:**

- Can you clarify how the choice of loss function impacts your theoretical results? It appears that in most cases, the squared loss is necessary. Does your analysis apply to SVMs?

**Limitations:**

- A more thorough experimental analysis can be helpful.

---

> ### Author Rebuttal · Authors · 2024-08-07
>
> We appreciate your time and great efforts in reviewing our paper and thanks a lot for your constructive comments and suggestions. Below are our point-to-point replies.
>
> **Question 1 (part I): Can you clarify how the choice of loss function impacts your theoretical results?**
>
> **Answer:**
> Thank you very much for your question. In fact, the theoretical results in Theorems 3.3 and 4.2 and Corollaries  3.4 and 4.3 are established for a general loss function,  belonging to a rich loss function family with Lipschitz continuity and satisfying Assumption 3.1. As discussed after Assumption 3.1 and in Appendix E, many popularly used loss functions satisfy these requirements under some mild conditions. In the revised version, we will add more detailed discussions on the effect of the choice of loss function at the end of Section 4.1.
>
> **Question 1 (part II):  It appears that in most cases, the squared loss is necessary.**
>
> **Answer:** Thank you very much for your comments. Note that the theoretical upper bounds provided in Sections 3 and 4 are established for a general loss function, whereas the minimax lower bound provided in Section 4.2 is established only for the squared loss. We want to emphasize that it is common in literature to establish the upper bound for other loss functions and compare it to the lower bound for the squared loss to check the optimality (Wei et al., 2017; Lv et al., 2018; Li et al., 2019). Moreover, the theoretical results in Sections 3 and 4 are established by using the empirical process techniques, which is in sharp contrast to that in  Amini et al. (2022)  where only the squared loss is considered.  More detailed discussions on the differences will be added in Section 5 of the revised manuscript.
>
> **Question 1 (part III):  Does your analysis apply to SVMs?**
>
> **Answer:**
> Thank you very much for your question. Note that from  Table 1 of the manuscript, the hinge loss satisfies the Lipshcitz continuity requirement with Lipshcitz constant 1. Moreover,   as pointed out by Wainwright (2019) on Page 472, Assumption 3.1 holds for the hinge loss when some conditions on data distribution and function class are satisfied.  Moreover, in this revision, we also added some numerical experiments using the hinge loss to verify our theoretical analysis, and the obtained numerical results as provided in Figure 1 of the total Rebuttal pdf file further support the applicability of our analysis to SVMs. In the revised version, we will add more detailed discussions on the applicability of the established results in Appendix E.
>
> **Limitations: A more thorough experimental analysis can be helpful.**
>
> **Answer:**
> Thank you very much for your suggestion.   In the revised version,   more thorough numerical experiments will be added in Section 6 and Appendix. One added numerical experiment is to investigate the problem that once the hinge loss is specified (corresponding to SVMs), how the RKHS with varying model complexities affect the numerical performance of KM and TKM.  Clearly, this added numerical experiment serves as a complement to that reported in Section 6 of the original submission where the check loss is considered. Specifically, the experiment setup is the same as that of the original submission, including the selection of kernel, repeat times, and tuning method for $\lambda$ and $r$ except that the underlying true function is set as $f^*(\mathbf{x})=\sin(11\mathbf{x})$ and  $(\mathbf{x}_i, y_i) _{i=1}^{300}$ is independently drawn from $y _i=\text{sign}(f^*(\mathbf{x} _i)+N(0, 4))$ with $\mathbf{x}_i=\frac{i-1}{300}, i=1, \dots, 300$.  The obtained numerical results are reported in Figure 1 of the total Rebuttal pdf file. It is thus clear from Figure 1 that the error curves for the hinge loss align with those for the check loss, which further confirms our theoretical findings and also empirically supports that our theoretical analysis can apply to SVMs. Similarly, we will also add the numerical experiments when the logistic loss function is specified, and the numerical results will be added in Appendix.

---

> > ### Author Response · Authors · 2024-08-07
> > **Reference**
> >
> > We apologize for any confusion regarding the references cited in our response during the rebuttal phase. Below, we have provided the complete reference information that you may need.
> >
> > [1]  Wei, Y., Yang, F., & Wainwright, M. J. (2017). Early stopping for kernel boosting algorithms: A general analysis with localized complexities. *Advances in Neural Information Processing Systems,* 30.
> >
> > [2]  Lv, S., Lin, H., Lian, H., & Huang, J. (2018). Oracle inequalities for sparse additive quantile regression in reproducing kernel Hilbert space. *The Annals of Statistics, 46,* 781–813.
> >
> > [3]  Li, Z., Ton, J.-F., Oglic, D., & Sejdinovic, D. (2019). Towards a unified analysis of random Fourier features. *In International Conference on Machine Learning* (pp. 3905–3914). PMLR.
> >
> >
> > [4]  Amini, A., Baumgartner, R., & Feng, D. (2022). Target alignment in truncated kernel ridge regression. *In Advances in Neural Information Processing Systems* (pp. 21948–21960). Curran Associates, Inc. volume 35.
> >
> > [5]  Wainwright, M. J. (2019). *High-dimensional Statistics: A Non-asymptotic Viewpoint* volume 48. Cambridge University Press.

---

### Author Rebuttal · Authors · 2024-08-07

Dear Reviewers,

We sincerely thank you for your insightful comments and for the time you have dedicated to thoroughly reviewing our work.  Your valuable and constructive feedback has significantly contributed to enhancing the quality of our work.
We have carefully considered all comments, concerns, and questions and have provided detailed responses to each review separately. These responses have been meticulously incorporated into the revised paper, mainly covering the following aspects:
- Highlighting the contributions of our paper from both theoretical and practical perspectives;
- Providing  deeper insights into the established results from both theoretical and numerical aspects; discussing the potential future direction, including the extension to non-Lipschitz continuous case;
-  Conducting additional experiments to further validate our theoretical findings, and part of the numerical results are contained in the attached pdf file；
- Correcting all typos and ensuring clarity of the introduced symbols and expressions.

Once again, we extend our sincere gratitude for your time, expertise, and contribution to our work. We would be grateful for your reply to ensure that all your concerns have been adequately addressed,  and we welcome any additional comments or suggestions you might have.

Warm regards,

the Authors

---

> ### Comment · Area_Chair_W6Rn · 2024-08-12
>
> Thank you for your detailed response and dialog with reviewers.
>
> One comment/question I have is around your numerical evaluation: from what I can see, the evaluation is conducted with synthetic data only. Do you know if you also see similar improved MSE on some non-synthetic datasets (as used in previous kernel alignment studies [1], [2])?
>
> [1] Cristianini, Nello, et al. "On kernel-target alignment." Advances in neural information processing systems 14 (2001).
> [2] Cortes, Corinna, et al. "Algorithms for learning kernels based on centered alignment." The Journal of Machine Learning Research 13 (2012): 795-828.

---

> > ### Author Response · Authors · 2024-08-13
> >
> > Thank you very much for your question and for bringing us these references. Following your suggestion, we apply both TKM and KM with check loss to the wine quality dataset, which is available in the UCI Machine Learning Repository. Specifically, we first adopt the random forest method (Breiman, 2001) to rank the feature importance and select the first three influential features: ‘Alcohol’, ‘Sulfates’, and ‘Volatile Acidity’ for analysis. Then, we randomly select 500 samples for training and another 500 samples for testing. The above procedure is repeated 10 times, where the Laplacian kernel is adopted and the parameters $\gamma$ and $r$ are tuned by $5$-fold cross-validation. The averaged MSE  with different $\tau\in (0.3,0.5,0.7)$ is reported in Table 1.
> >
> > It is thus clear that the obtained results in the real application align with the results for synthetic data and our theoretical findings, which further demonstrates the benefits of TKM.  Due to the time limit, a more comprehensive analysis and discussion of these two competitors in real applications will be added to Appendix of the revised manuscript.
> >
> > **Table 1： Averaged MSE for different $\tau$.**
> > | $\tau$  | $0.3$           | $0.5$           | $0.7$           |
> > |------|-----------------|-----------------|-----------------|
> > | KM   | $0.590 \pm 0.027$ | $0.483 \pm 0.035$ | $0.638 \pm 0.073$ |
> > | TKM  | $0.548 \pm 0.026$ | $0.454 \pm 0.045$ | $0.530 \pm 0.046$ |
> >
> > **Reference:**
> >
> > [1] Breiman, L. (2001). Random forests. *Machine Learning,* 45:5–32.

---

### Decision · Program_Chairs · 2024-09-25

**Decision:**

Accept (poster)

**Comment:**

After reviewing the paper, reviewer comments, and author rebuttal, I recommend accepting this submission (poster).  During the rebuttal the authors satisfied several of the reviewer (and AC) comments, clarifying the contribution over existing works (extending TKM analysis to general Lipschitz functions and providing mimimax bounds), extended the analysis of TKM to target alignment with exponential decay rate, and providing some additional numerical verification on synthetic and small real-world datasets.  I expect the authors to update the final draft with these clarifications and additional results.